# Quantum Chemical Approaches to the Calculation of NMR Parameters: From Fundamentals to Recent Advances

**Irina L. Rusakova** 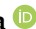

A. E. Favorsky Irkutsk Institute of Chemistry, Siberian Branch of the Russian Academy of Sciences, Favorsky St. 1, 664033 Irkutsk, Russia; i-rusakova@bk.ru

**Abstract:** Quantum chemical methods for the calculation of indirect NMR spin–spin coupling constants and chemical shifts are always in progress. They never stay the same due to permanently developing computational facilities, which open new perspectives and create new challenges every now and then. This review starts from the fundamentals of the nonrelativistic and relativistic theory of nuclear magnetic resonance parameters, and gradually moves towards the discussion of the most popular common and newly developed methodologies for quantum chemical modeling of NMR spectra.

**Keywords:** NMR; NMR spin–spin coupling constants; NMR chemical shifts; NMR spectra modeling; NMR quantum chemical methods

## 1. Introduction

The determination of the structure of compounds using the NMR spectral analysis has now become an integral part of physical-chemical research in organic and inorganic chemistry. In general, establishing the structure of compounds using NMR spectra based only on the empirical or semi-empirical rules is not always completely correct. An erroneous interpretation may occur, for example, when it is necessary to choose from several close presumptive structures, such as diastereomers, or, when analyzing new structures with NMR parameters going beyond the usual ranges. In such cases, high-quality quantum chemical calculations are necessary.

Since the application of the perturbation theory to the NMR properties by Ramsey, over 70 years ago, computational methodology has made a great leap due to both the accelerated progress of computer technique and the development of the electron theory. Now we can witness the flourishing of density functional methods, being successfully applied to biological macromolecules at acceptable computational cost as well as a routine usage of the ab initio wavefunction-based correlated methods such as polarization propagator approaches or coupled-cluster techniques in the calculations of the NMR properties of medium-size molecules.

Here it is relevant to recall that a good deal of reviews regarding the calculation of the indirect nuclear spin–spin coupling constants and NMR chemical shifts have appeared over the past decades. In this respect, quantum chemical methods for the calculations of NMR parameters were reviewed by Gauss et al. [1], Fukui et al. [2–4], Helgaker et al. [5–8], Contreras [9,10], Aucar et al. [11,12], Autschbach et al. [13–16], Sauer et al. [17], Lazzeretti [18], Cremer and Gräfenstein [19], de la Vega and Fabián [20], Rusakov et al. [21–24], Krivdin [25–40], Mulder et al. [41], Pyykkö [42], Facelli [43], and some others [44–49].

This review is aimed at surveying modern ubiquitous methods for modelling the NMR spectra of compounds with a special accent placed on the recent developments. Notes are given on the propensities of various quantum chemical methodologies suitable for the calculations of NMR parameters. Configuring a route to an effective computational protocol, a good number of factors influencing the accuracy, such as the choice of an atomic

basis set, the treatment of relativistic effects, vibrational corrections, and solvent effects, are discussed to a reasonable extent.

## 2. Theoretical Background

### 2.1. Nonrelativistic Representation of NMR Parameters

The original expressions of the NMR parameters were derived by Ramsey [50,51] on the basis of the second-order Rayleigh–Schrodinger perturbation theory without taking into account the relativistic effects. In the calculation of the NMR molecular properties, Schrödinger–Hamiltonian is extended to include hyperfine interactions between nuclei and magnetic field, which can be expressed as follows:

$$\hat{H}(\mathbf{B}, \boldsymbol{\mu}_N) = \hat{H}^{(0)} + \hat{\mathbf{H}}^{(10)}\mathbf{B} + \frac{1}{2}\mathbf{B}\hat{\mathbf{H}}^{(20)}\mathbf{B} + \sum_N \boldsymbol{\mu}_N\hat{\mathbf{H}}_N^{(01)} + \frac{1}{2}\sum_N \boldsymbol{\mu}_N\hat{\mathbf{H}}_N^{(11)}\mathbf{B} + \frac{1}{2}\sum_{MN} \boldsymbol{\mu}_M\hat{\mathbf{H}}_{MN}^{(02)}\boldsymbol{\mu}_N + \dots \tag{1}$$

where $\hat{H}^{(0)}$ is the Schrödinger–Hamiltonian, $\mathbf{B}$ is the external magnetic flux density, $\boldsymbol{\mu}_N$ is the nuclear magnetic moment, $\hat{\mathbf{H}}^{(nl)}$ is the interaction operators containing the $n$-th power of $\mathbf{B}$ and $l$-th power of $\boldsymbol{\mu}_N$. The wave function of a system is represented in the form of power series of $\mathbf{B}$ and $\boldsymbol{\mu}_N$:

$$\Psi(\mathbf{B}, \boldsymbol{\mu}_N) = \Psi^0 + \Psi^{(10)} \cdot \mathbf{B} + \sum_N \Psi_N^{(01)} \cdot \boldsymbol{\mu}_N + \dots, \tag{2}$$

where $\Psi^0$ is the unperturbed ground state wave function and $\mathbf{\Psi}^{(nl)}$ are the expansion coefficients. Substituting $\hat{H}(\mathbf{B}, \boldsymbol{\mu}_N)$ and $\Psi(\mathbf{B}, \boldsymbol{\mu}_N)$ into the expression for electronic energy:

$$E(\mathbf{B}, \boldsymbol{\mu}_N) = \langle \Psi(\mathbf{B}, \boldsymbol{\mu}_N) | \hat{H}(\mathbf{B}, \boldsymbol{\mu}_N) | \Psi(\mathbf{B}, \boldsymbol{\mu}_N) \rangle, \tag{3}$$

gives an infinite power series:

$$E(\mathbf{B}, \boldsymbol{\mu}_N) = E^0 + \mathbf{E}^{(10)} \cdot \mathbf{B} + \sum_N \mathbf{E}_N^{(01)} \cdot \boldsymbol{\mu}_N + \sum_N \boldsymbol{\mu}_N^{\mathrm{T}}\mathbf{E}_N^{(11)}\mathbf{B} + \sum_{NM} \boldsymbol{\mu}_N^{\mathrm{T}}\mathbf{E}_N^{(02)}\boldsymbol{\mu}_M + \dots \tag{4}$$

By definition, the NMR shielding tensor is expressed as the second derivative of the total energy with respect to the Cartesian components of an external magnetic flux density and the nuclear magnetic moment

$$\sigma_{N;\alpha\beta} = \left. \frac{\partial^2 E(\mathbf{B}, \ \boldsymbol{\mu})}{\partial B_\alpha \partial \mu_{N;\beta}} \right|_{\boldsymbol{\mu}_N = 0, \ \mathbf{B} = 0} \tag{5}$$

The tensor $\boldsymbol{\sigma}_N$ can be split apart into two components that are different from a physical point of view, namely the diamagnetic and paramagnetic contributions:

$$\boldsymbol{\sigma}_N = \boldsymbol{\sigma}_N^{dia} + \boldsymbol{\sigma}_N^{para}. \tag{6}$$

These contributions were first deduced by Ramsey [50] within the framework of the common perturbation theory. The diamagnetic contribution has the form of an average of the diamagnetic operator over the ground state of the unperturbed system:

$$\boldsymbol{\sigma}_N^{dia} = C^{dia} \left\langle {}^1\Psi_0^{(0)} \left| \sum_i \left( \mathbf{r}_{i0}^{\mathrm{T}}\mathbf{r}_{iN}\mathbf{I} - \mathbf{r}_{i0}\mathbf{r}_{iN}^{\mathrm{T}} \right) r_{iN}^{-3} \right| {}^1\Psi_0^{(0)} \right\rangle. \tag{7}$$

At the same time, the paramagnetic contribution is determined through the perturbed wave function as a sum over singlet-excited states of the electronic system:

$$\sigma_N^{para} = -C^{para} \sum_{n \neq 0} \left( {}^1E_n^{(0)} - {}^1E_0^{(0)} \right)^{-1} \left\{ \left\langle {}^1\Psi_0^{(0)} \left| \sum_i \hat{\mathbf{L}}_{i0} \right| {}^1\Psi_n^{(0)} \right\rangle \left\langle {}^1\Psi_n^{(0)} \left| \sum_j 2\hat{\mathbf{L}}_{jN}^{\mathrm{T}} r_{jN}^{-3} \right| {}^1\Psi_0^{(0)} \right\rangle + \right.$$

$$\left. + \left\langle {}^1\Psi_0^{(0)} \left| \sum_i 2\hat{\mathbf{L}}_{iN} r_{iN}^{-3} \right| {}^1\Psi_n^{(0)} \right\rangle \left\langle {}^1\Psi_n^{(0)} \left| \sum_j \hat{\mathbf{L}}_{j0}^{\mathrm{T}} \right| {}^1\Psi_0^{(0)} \right\rangle \right\}, \tag{8}$$

where the $\hat{\mathbf{L}}_{i0} = -i(\mathbf{r}_i - \mathbf{R}_0) \times \vec{\nabla}_i$ and $\hat{\mathbf{L}}_{iN} = -i(\mathbf{r}_i - \mathbf{R}_N) \times \vec{\nabla}_i$ are the electron angular momentum in relation to the gauge origin and the position of $N$th nucleus, respectively. From a physical point of view, the diamagnetic contribution (7) is due to the diamagnetic circular electron currents in the orbitals of the atom $N$ induced by an external magnetic field. The expression (7) is an analogue of Lamb's formula [52], which describes the shielding effect for nucleus of an atom, which is proportional to the electron density in the region of the nucleus.

The paramagnetic component is expressed in terms of paramagnetic currents caused by the mixing of the ground and excited states under the action of an external field. The paramagnetic contribution is nonzero only for systems having the electrons with a non-zero angular momentum. The diamagnetic component is a local property, while the paramagnetic part can be roughly divided into local and non-local components, the first of which is expressed in terms of local paramagnetic currents of the atom in question, and the second is due to the electron currents of neighboring atoms or currents circulating over the groups of atoms, as, for example, in the case of aromatic molecules. The paramagnetic and diamagnetic components are of different signs.

In a liquid or gas phase NMR experiment, the rotation of molecules is so fast that the nuclear shielding tensor is isotropically averaged. The isotropic constant of the nuclear magnetic shielding is expressed as the one third of the trace of the corresponding tensor:

$$\sigma_{N;iso} = \frac{1}{3} Tr(\boldsymbol{\sigma}_N) = \frac{1}{3} \sum_{\alpha = x,y,z} \sigma_{N;\alpha\alpha}. \tag{9}$$

In order to obtain the data independent of experimental conditions, the so-called chemical shifts or $\delta$-scale was introduced. The values on $\delta$-scale represent the ratio of chemical shifts measured in Hz and the operating frequency of the spectrometer. As a result, dimensionless quantities $\delta$, measured in points per millionths (ppm), are obtained. The chemical shifts, $\delta$, measured in the NMR experiment, are calculated using the International Union of Pure and Applied Chemistry (IUPAC) formula [53,54], through the isotropic absolute NMR shielding constant of a given nucleus in the reference compound (the standard), $\sigma_{ref}$, and that of the compound under question, $\sigma_{sample}$ [55]:

$$\delta = \left. \frac{\sigma_{ref} - \sigma_{sample}}{1 - \sigma_{ref}} \times 10^6 \right| \tag{10}$$

However, in the units of parts per million, a simplified equation for chemical shifts is adopted:

$$\delta = \sigma_{ref} - \sigma_{sample}. \tag{11}$$

The magnetic vector-potential of the external magnetic field, $\mathbf{A}$, uniquely determines the magnetic field $\mathbf{B} = \nabla \times \mathbf{A}(\mathbf{r})$, however, the otherwise statement is not true, i.e., a given magnetic field $\mathbf{B}$ gives a variety of vector-potentials $\mathbf{A}$. Suppose, one adds the gradient of any scalar function $\nabla f$ to a given vector-potential $\mathbf{A}$. This results in a zero change in the magnetic field of $\mathbf{B}$, because $\nabla \times \nabla f = 0$. This ambiguity emerges in the expressions (7) and (8) as the dependence of the resulting values on the radius-vector of the center of the coordinate system. The multivariance in describing a physical property with different vector potentials leads to a natural requirement to the property to be independent from the selected coordinate center. This requirement is usually referred to as the gauge invariance principle. The gauge invariance is trivially satisfied by the exact solutions of

the Schrödinger equation (for the proof, see, for example, [56]), however, for approximate solutions this is not the case. This is a serious problem for quantum chemistry, which, in fact, is built upon the approximate solutions. The violation of the gauge invariance principle in approximate approaches of quantum chemistry occurs for two reasons: (a) the use of finite basis sets; (b) the fact that some quantum chemical schemes do not obey the virial theorem [57]. The first reason is usually considered the most serious; the latter is mentioned in the literature much less frequently [58]. For methods that do not violate the virial theorem, for example, the Hartree–Fock method, it can be shown that they provide gauge invariance of the observed physical quantities in the complete basis set (CBS) limit [59].

The solution of the gauge origin problem lies in going to the terms of local gauge origins. The main idea of these methods is to avoid using a single coordinate center, which, in principle, does not provide an adequate solution to the calibration problem in the calculations of molecular properties. For atoms, it was shown that the best possible choice is to place the coordinate center at the position of a nucleus. This ensures the fastest convergence with respect to the basis set. Withdrawal of the coordinate origin from the nucleus position leads to a dramatic deterioration of the results. For molecules, the difficulty lies in the fact that there is no optimal unique origin of the reference system, since the molecule is a system of many nuclei.

The introduction of individual gauge origins for various local parts of the wave function alleviates the gauge origin problem. Since the operators of the hyperfine magnetic interaction are of one-electron type, it was proposed to introduce the local origin of the coordinate system for individual one-electron components of wave functions, that is, either for the molecular or atomic orbitals [60]. In the case of choosing the individual gauge origins for the molecular orbitals, a serious problem, connected with their strong delocalization, arises. At the initial stage, this disadvantage was circumvented by using the localized molecular orbitals within the framework of the individual gauge for localized orbitals (IGLO) by Kutzelnigg et al. [61,62], and within the localized orbitals/local origin (LORG) scheme by Hansen and Bowman [63,64].

However, this leads to additional constraints for the wave functions in the electron-correlated methods. Resorting to the individual gauge origins for atomic orbitals presents a more adequate solution to the gauge origin problem in molecules. Atomic orbitals are necessarily localized; therefore, no additional constraint concerning the quantum chemical treatment is required. The use of local (nucleus-centered) origins for atomic orbitals is usually referred to as the gauge-including atomic orbitals (GIAO) approach [59,65–71]. Formally, the local origins of the coordinate system are introduced by the non-canonical transformations of the following form [1]:

$$\Psi \rightarrow \Psi' = \sum_{\mu} \exp(-\Lambda_{\mu}(\mathbf{r})) \hat{P}_{\mu} \Psi \tag{12}$$

$$\hat{H} \rightarrow \hat{H}' = \sum_{\mu} \hat{P}_{\mu} \exp(-\Lambda_{\mu}(\mathbf{r})) \hat{H} \sum_{\nu} \exp(\Lambda_{\nu}(\mathbf{r})) \hat{P}_{\nu} \tag{13}$$

where

$$\Lambda_{\mu}(\mathbf{r}) = \frac{i}{2} \big[ (\mathbf{R}_{\mu} - \mathbf{R}_0) \times \mathbf{B} \big] \mathbf{r} \tag{14}$$

and $\hat{P}_{\mu}$ is the projection operator on the local fragments of the molecule wave function, which shifts the gauge origin for the part denoted by $\mu$-th to the new position $\mathbf{R}_{\mu}$. The projection operator $\hat{P}_{\mu}$ is expressed as follows:

$$\hat{P}_{\mu} = \sum_{\nu} |\chi_{\mu}\rangle S_{\mu\nu}^{-1} \langle \chi_{\nu}|, \tag{15}$$

where $S_{\mu\nu}$ are the elements of the atomic orbital overlap matrix, given that the new origin is placed at the corresponding nuclear position. From the formal point of view, introduction of multiple local gauge origins $\mathbf{R}_{\mu}$ of the reference frame does not solve the problem of

the violation of the gauge invariance principle, since there is not an explicit reformulation of the problem in the gauge invariant form. Instead, the methods of multiple local gauge origins give results that do not depend on the choice of the origin of the global coordinate system. In addition, it should be noted that the approach of local coordinate systems is physically motivated, and in practice demonstrate much better convergence of properties on the basis sets.

In addition to the splitting of nuclear spin levels in the external field, there is an additional splitting due to the interaction of nuclear spins. One part of this interaction is due to the direct magnetic dipole-dipole interaction of nuclear spins. In the liquid or gas phase NMR experiment, the direct tensor gives zero due to the isotropic averaging. The second part of this is due to the polarization of the electron cloud caused by the nuclear spins under consideration. The first part of the coupling tensor is called the direct coupling tensor and its components are measured in the solid-state NMR experiments, while the second part is called the indirect reduced coupling tensor, and it gives a non-zero value under the isotropic averaging in the liquid or gas phase NMR experiments. The components of the indirect reduced coupling tensor are the second partial derivatives of the total perturbed energy of an electronic system in relation to the Cartesian components of magnetic moments of the nuclei:

$$K_{MN;\alpha\beta} = \left.\frac{\partial^2 E(\mathbf{B}, \boldsymbol{\mu})}{\partial \mu_{N,\alpha} \partial \mu_{M,\beta}}\right|_{\boldsymbol{\mu}_N, \boldsymbol{\mu}_M = 0} \tag{16}$$

The tensor $\mathbf{K}_{MN}$ has four contributions:

$$\mathbf{K}_{MN} = \mathbf{K}_{MN}^{DSO} + \mathbf{K}_{MN}^{PSO} + \mathbf{K}_{MN}^{FC} + \mathbf{K}_{MN}^{SD} \tag{17}$$

These contributions were first deduced by Ramsey [51]. The first contribution in the Equation (17) is called as the diamagnetic spin-orbit contribution (DSO). It comes from the diamagnetic operator, which is included in the Hamiltonian of the system as an operator bilinear in the magnetic moments of the nuclei and corresponds to the interaction of the magnetic moments of the nuclei through induced diamagnetic currents. It is very similar to the diamagnetic contribution to the nuclear shielding tensor, and represents the average of the diamagnetic interaction operator over the ground state:

$$\mathbf{K}_{MN}^{DSO} = C^{DSO} \left\langle {}^1\Psi_0^{(0)} \left| \sum_i \frac{\mathbf{r}_{iM}^{\mathrm{T}} \mathbf{r}_{iN} \mathbf{I} - \mathbf{r}_{iM} \mathbf{r}_{iN}^{\mathrm{T}}}{r_{iM}^3 r_{iN}^3} \right| {}^1\Psi_0^{(0)} \right\rangle \tag{18}$$

The second contribution in the Equation (17) is the paramagnetic spin-orbit contribution (PSO). The PSO contribution is very close to the paramagnetic contribution to the nuclear magnetic shielding tensor. It is also expressed in terms of the sum of excited singlet states and also contains the angular momentum operators of electrons:

$$\mathbf{K}_{MN}^{PSO} = C^{PSO} \sum_{n\neq0} \left( {}^1E_n^{(0)} - {}^1E_0^{(0)} \right)^{-1} \left\langle {}^1\Psi_0^{(0)} \left| \sum_i \frac{\hat{\mathbf{L}}_{iM}}{r_{iM}^3} \right| {}^1\Psi_n^{(0)} \right\rangle \left\langle {}^1\Psi_n^{(0)} \left| \sum_j \frac{\hat{\mathbf{L}}_{jN}^{\mathrm{T}}}{r_{jN}^3} \right| {}^1\Psi_0^{(0)} \right\rangle \tag{19}$$

Its physical meaning consists in the transfer of the nuclear spin–spin interaction through the induction of the paramagnetic orbital electron currents.

The third and fourth contributions in Equation (17) correspond to the Fermi-contact (FC) and spin-dipole (SD) contributions. These contributions come from the hyperfine dipole interaction of the magnetic moment of one of the nuclei under consideration with the electron spins, resulting in the polarization of the electron spins of a system, which is transmitted to the region of the second nucleus and leads to the appearance of an additional magnetic field in that region. In fact, the FC and SD terms originate from the same hyperfine interaction operator and describe the same physical process. However, the FC contribution

is due to the electrons of orbitals intersecting the nuclear centers. The corresponding equations for the FC and SD contributions are as follows:

$$\mathbf{K}_{MN}^{FC/SD} = C^{FC/SD} \sum_{n \neq 0} \left( {}^3E_n^{(0)} - {}^1E_0^{(0)} \right)^{-1} \left\langle {}^1\mathbf{\Psi}_0^{(0)} \middle| \hat{\mathbf{H}}_M^{SD/FC} \middle| {}^3\mathbf{\Psi}_n^{(0)} \right\rangle \left\langle {}^3\mathbf{\Psi}_n^{(0)} \middle| \hat{\mathbf{H}}_N^{SD/FC} \middle| {}^1\mathbf{\Psi}_0^{(0)} \right\rangle \tag{20}$$

$$\hat{\mathbf{H}}_M^{FC} = \sum_i \delta^3(\mathbf{r}_{iM}) \hat{\mathbf{s}}_i \tag{21}$$

$$\hat{\mathbf{H}}_M^{SD} = \sum_i \frac{3\mathbf{r}_{iM}\mathbf{r}_{iM}^{\mathrm{T}} - \mathbf{r}_{iM}^2\mathbf{I}}{r_{iM}^5} \hat{\mathbf{s}}_i \tag{22}$$

The isotropic spin–spin coupling constant (measured in Hz) is obtained from the reduced tensor **K** as one third of its trace multiplied by a coefficient containing the product of the gyromagnetic ratios of the nuclei under consideration:

$$J_{MN} = \frac{1}{3} \cdot \left( \frac{\hbar}{2\pi} \right) \cdot \gamma_M \gamma_N \cdot Tr(\mathbf{K}_{MN}) \tag{23}$$

The most part of the modern nonrelativistic quantum chemical methods for calculating the NMR parameters are based on Equations (6)–(9), (17)–(23). However, it is impossible to use them in practice in their original form, since they include exact wave functions and energies. A route to make them useful for computational quantum chemistry is to express them in terms of molecular orbitals with the aid of the second quantization technique [72]. In the simplest approximation, where the excited wave functions are built by the action of the singlet and triplet excitation operators on the ground state wave function, described by the Hartee–Fock single-determinant many-body wave function, the nonrelativistic contributions to nuclear shielding and spin–spin coupling constants are as follows:

$$\boldsymbol{\sigma}_N^{dia} = C^{dia} \sum_i^{occ} \langle \varphi_i | \left( \mathbf{r}_0^{\mathrm{T}} \cdot \mathbf{r}_N \mathbf{I} - \mathbf{r}_0 \mathbf{r}_N^{\mathrm{T}} \right) r_N^{-3} | \varphi_i \rangle \tag{24}$$

$$\boldsymbol{\sigma}_N^{para} = -C^{para} \sum_i^{occ} \sum_a^{vac} (\varepsilon_i - \varepsilon_a)^{-1} \left\{ \langle \varphi_i | \hat{\mathbf{L}}_0 | \varphi_a \rangle \langle \varphi_a | 2\hat{\mathbf{L}}_N^{\mathrm{T}} r_N^{-3} | \varphi_i \rangle + perm. \right\} \tag{25}$$

$$\mathbf{J}_{MN}^{DSO} = C^{DSO} \sum_i^{occ} \langle \varphi_i | \frac{\mathbf{r}_M^{\mathrm{T}} \mathbf{r}_N \mathbf{I} - \mathbf{r}_M \mathbf{r}_N^{\mathrm{T}}}{r_M^3 r_N^3} | \varphi_i \rangle \tag{26}$$

$$\mathbf{J}_{MN}^{X} = C^{X} \sum_i^{occ} \sum_a^{vac} (\varepsilon_i - \varepsilon_a)^{-1} \langle \varphi_i | \hat{\mathbf{H}}^X | \varphi_a \rangle \langle \varphi_a | \left( \hat{\mathbf{H}}^X \right)^{\mathrm{T}} | \varphi_i \rangle \tag{27}$$

In Equation (27), the operator $\hat{\mathbf{H}}^X$ designates different types of hyperfine operators, namely X = FC, SD and PSO. In these equations $|\varphi_i\rangle$ and $|\varphi_a\rangle$ correspond to the occupied and vacant molecular orbitals with the energies $\varepsilon_i$ and $\varepsilon_a$, respectively.

## 2.2. Relativistic Representation of NMR Parameters

For heavy elements, the available methods for prediction of NMR parameters based on the Schrödinger equation often become insufficient. In these cases, the average orbital velocities of electrons in the vicinity to nuclei are close to the speed of light, giving rise to relativistic effects such as spin-orbit coupling, the Darwin term, and the mass-velocity correction, which can all substantially affect the NMR spectroscopic parameters. Relativistic effects on the NMR parameters can already take place for the compounds bearing the atoms of the third period of the periodic table. The magnitude of the relativistic effects on the NMR parameters cannot be estimated simply from the reasoning in the atomic terms, using, for instance, the well-known Lorentz factor $\gamma = (1 - v^2/c^2)^{-1/2}$, which allows to determine the "relativistic" contraction of the inner $1s$ shells ($[(\gamma - 1)/\gamma] \times 100\%$). That is not enough for the NMR parameters, since the relativistic corrections to these are rather non-local

properties, which are determined by the electronic structure of the entire electron system of the molecule.

The study of the relativistic effects [73–87] in the NMR parameters was commenced by the scientific groups of Pyper [88–90], Zhang [91] and Pyykkö [92,93] in the 1980s. The pioneering ideas, proposed in these works, were gradually developed to coherent relativistic theories by the scientific groups of Nakatsuji [94–99], Fukui [100,101], Liu [102–107], Aucar [108–115], Vaara [116–119], Autschbach [120–123], Manninen [124–126], Sauer [127], and some others [128–133].

The transition to the relativistic representation of NMR parameters is based on the stationary Dirac equation for a particle in the external electrostatic potential of nuclei. This equation can be represented as a matrix equation with a $4 \times 4$ Hamiltonian:

$$\hat{\mathbf{h}}^{\mathrm{D}}_{4\times4}\boldsymbol{\psi}_{4\times1} = \left(c\vec{\boldsymbol{\alpha}}\vec{\hat{p}} + mc^2(\boldsymbol{\beta}-\mathbf{I}) + \hat{V}\mathbf{I}\right)_{4\times4}\boldsymbol{\psi}_{4\times1} = \begin{pmatrix} \hat{V}\mathbf{I}_{2\times2} & c\vec{\boldsymbol{\sigma}}_{2\times2}\vec{\hat{p}} \\ c\vec{\boldsymbol{\sigma}}_{2\times2}\vec{\hat{p}} & (\hat{V}-2mc^2)\mathbf{I}_{2\times2} \end{pmatrix}\boldsymbol{\psi}_{4\times1} = E\boldsymbol{\psi}_{4\times1}. \tag{28}$$

In Equation (28), the energy level is shifted down by the energy of rest, $mc^2$, and the moieties $\boldsymbol{\beta}$ and $\vec{\boldsymbol{\alpha}}$ are the $4 \times 4$ matrices, called the Dirac matrices:

$$\vec{\boldsymbol{\alpha}} = \begin{bmatrix} \mathbf{0}_{2\times2} & \vec{\boldsymbol{\sigma}} \\ \vec{\boldsymbol{\sigma}} & \mathbf{0}_{2\times2} \end{bmatrix}, \; \boldsymbol{\beta} = \begin{bmatrix} \mathbf{1}_{2\times2} & \mathbf{0}_{2\times2} \\ \mathbf{0}_{2\times2} & -\mathbf{1}_{2\times2} \end{bmatrix}. \tag{29}$$

The matrices $\boldsymbol{\beta}$ and $\vec{\boldsymbol{\alpha}}$ consist of $2 \times 2$ Pauli matrices $\vec{\boldsymbol{\sigma}} = \{\sigma_x, \sigma_y, \sigma_z\}$, which represent the elements of the electron spin operator matrix, $\vec{\boldsymbol{\sigma}} = 2\vec{\hat{s}}$:

$$\boldsymbol{\sigma}_x = \begin{pmatrix} 0 & 1 \\ 1 & 0 \end{pmatrix}, \; \boldsymbol{\sigma}_y = \begin{pmatrix} 0 & -i \\ i & 0 \end{pmatrix}, \; \boldsymbol{\sigma}_z = \begin{pmatrix} 1 & 0 \\ 0 & -1 \end{pmatrix}. \tag{30}$$

The Dirac Equation (28) satisfies all the necessary requirements, in particular, it is Lorentz-covariant, provides a positive probability density, and, in addition, it resorts to the notion of spin as an additional degree of freedom. The solutions of Equation (28) are stationary 4-spinors, $\boldsymbol{\psi}_{4\times1}$ that can be expressed as bispinors, consisting of "large" and "small" components:

$$\boldsymbol{\psi}^{\mathrm{D}}_{4\times1} = \begin{pmatrix} \psi_1 \\ \psi_2 \\ \psi_3 \\ \psi_4 \end{pmatrix} = \begin{pmatrix} \boldsymbol{\psi}_{\mathrm{L}2\times1} \\ \boldsymbol{\psi}_{\mathrm{S}2\times1} \end{pmatrix}. \tag{31}$$

Free-particle Dirac equation (without external potential) has two types of solutions: with positive and with negative energies. Solutions with the positive energies correspond to the electronic continuum, unlimited from above, and those possessing the negative energy belong to the positronic continuum, unlimited from below. The introduction of the external potential leads to the appearance of bound electronic states with a discrete spectrum $-2mc^2 < E < 0$.

Generalization to the case of a many-particle system can be performed by means of introduction of the interelectronic interaction operator into the Hamiltonian and by going to a multidimensional Hilbert space, which is the direct product of one-partial Hilbert spaces. This leads to a Hamiltonian of the dimension $4^{Ne} \times 4^{Ne}$, where $N_e$ is the number of electrons. However, the Hamiltonian of this large dimension is not applicable in practice. As a rule, all standard relativistic methods reduce the multi-electron problem to single-particle equations, so that the resulting equations are very similar to Equation (28). The construction of any single-particle approximations relies upon the four-component many-particle Dirac–Coulomb–Breit Hamiltonian (DCB) [134]:

$$\hat{\mathbf{H}}_{DCB} = \sum_i^{Ne} \hat{\mathbf{h}}_D(i) + \frac{1}{2}\sum_{i\neq j}^{Ne} \hat{\mathbf{g}}(i,j) \tag{32}$$

where $\hat{\mathbf{h}}_D(i)$ is the single-particle Dirac operator, presented by Equation (28) with operator $\hat{V}$ describing the interaction of the electrons with the fixed nuclear framework, whereas the operator $\hat{\mathbf{g}}(i,j)$ represents the interelectronic interaction and is called the Coulomb–Breit operator, which is the sum of the usual Coulomb operator and the leading relativistic Breit correction to the energy of the two-electron interaction, consisting of the Gaunt operator and the gauge term:

$$\hat{\mathbf{g}}(i,j) = r_{ij}^{-1}\mathbf{1}_{4\times4} + \hat{\mathbf{g}}_B(i,j) \tag{33}$$

$$\hat{\mathbf{g}}_B(i,j) = \hat{\mathbf{g}}_{Gaunt}(i,j) + \hat{\mathbf{g}}_{gauge}(i,j) = -\frac{1}{2r_{ij}}\left\{\vec{\alpha}_i\vec{\alpha}_j + \frac{(\vec{\alpha}_i\,\vec{r}_{ij})(\vec{\alpha}_j\,\vec{r}_{ij})}{r_{ij}^2}\right\} \tag{34}$$

The equations for the NMR parameters in the relativistic representation are derived from the energy obtained from the four-component DCB Hamiltonian on the basis of common perturbation theory with the external and nuclei magnetic fields considered as perturbations. The kinetic term in the presence of magnetic fields is obtained by the so-called minimal substitution of the kinetic operator $c\vec{\alpha}\hat{\vec{p}}$ for the prorogated kinetic operator $c\vec{\alpha}\hat{\vec{\pi}}$, where $\hat{\vec{\pi}} = \hat{\vec{p}} + e\vec{A}$, while $\vec{A}$ is the total vector potential of all magnetic fields. Overall, the vector potential $\vec{A}$ is the sum of the vector potentials of the external magnetic field $(\vec{A}_0)$ and the magnetic fields induced by the nuclei $(\vec{A}_N)$:

$$\vec{A} = \vec{A}_0 + \sum_N^{N_{nuc}} \vec{A}_N, \tag{35}$$

$$\vec{A}_0 = \frac{1}{2}\vec{B} \times \vec{r}_0, \ \vec{r}_0 = \vec{r} - \vec{R}_0, \tag{36}$$

$$\vec{A}_N = \frac{\mu_0}{4\pi}\frac{\vec{\mu}_N \times \vec{r}_N}{r_N^3}, \ \vec{r}_N = \vec{r} - \vec{R}_N. \tag{37}$$

In these equations, $\vec{r}$, $\vec{r}_0$, and $\vec{R}_N$ refer to the coordinates of an electron, the origin of the coordinate system and that of the *N*th nucleus, respectively. Thus, the magnetic perturbation describing the interaction of electrons with a magnetic field has the form:

$$\hat{H}_{NMR}^{rel}(\vec{B}, \vec{\mu}_N) = \frac{ce}{2}\vec{B}\cdot[\vec{\alpha}\times\vec{r}_0] + \frac{ce\mu_0}{4\pi}\sum_N\vec{\mu}_N\cdot\frac{[\vec{\alpha}\times\vec{r}_N]}{r_N^3}. \tag{38}$$

Taking into account the relativistic magnetic Hamiltonian (38), the correction to the energy up to the second order according to the standard perturbation theory can be expressed as follows:

$$\Delta E = \left\langle \mathbf{\Psi}^{(0)}\left|\hat{\mathbf{H}}_{11}^{pert}\right|\mathbf{\Psi}^{(0)}\right\rangle + \sum_{n\neq0}\frac{\left\langle\mathbf{\Psi}^{(0)}\left|\hat{\mathbf{H}}_{01}^{pert}\right|\mathbf{\Phi}^{(n)}\right\rangle\left\langle\mathbf{\Phi}^{(n)}\left|\hat{\mathbf{H}}_{10}^{pert}\right|\mathbf{\Psi}^{(0)}\right\rangle}{E^{(0)} - E^{(n)}} + (0 \rightleftarrows 1). \tag{39}$$

where the magnetic perturbation is represented as the sum $\hat{\mathbf{H}}^{pert} = ec\vec{\alpha}\vec{A}_1 + ec\vec{\alpha}\vec{A}_2$, where $\vec{A}_1$, $\vec{A}_2$ are the $\vec{A}_0$, $\vec{A}_N$ and $\vec{A}_M$, $\vec{A}_N$ in the case of $\sigma_{N;\alpha\beta}$ and $K_{MN;\alpha\beta}$, respectively. In Equation (39), $\hat{\mathbf{H}}_{11}^{pert}$, $\hat{\mathbf{H}}_{01}^{pert}$ and $\hat{\mathbf{H}}_{10}^{pert}$ correspond to various combinations of perturbations in the general operator $\hat{\mathbf{H}}^{pert}$, namely, the first operator is a term bilinear by $\vec{A}_1$ and $\vec{A}_2$, while the other two are linear perturbations containing either $\vec{A}_1$ or $\vec{A}_2$. $\left|\mathbf{\Psi}^{(0)}\right>$ represents the unperturbed ground state with the energy $E^{(0)}$. $\left|\mathbf{\Psi}^{(0)}\right>$ are the excited states of an undisturbed system with the energies $E^{(n)}$. The first term in Equation (39)

turns to zero, since there are no terms containing $\vec{A}_1$ and $\vec{A}_2$ at the same time. This means that the diamagnetic contributions to the tensors $\sigma_{N;\alpha\beta}$ and $K_{MN;\alpha\beta}$ are not explicitly present, as opposed to what is observed in the nonrelativistic picture. Nevertheless, despite the compactness of the Hamiltonian (38), all nonrelativistic operators involved in the description of the NMR phenomenon become explicit when going to the nonrelativistic limit [121,122]. The relativistic expressions for the NMR parameters in the four-component representation are derived similarly to the nonrelativistic case, i.e., by double differentiation of the energy correction (39) by the corresponding magnetic perturbations. The final expressions for the shielding and spin–spin coupling tensors have the form of the relativistic four-component polarization propagators in the static approximation:

$$\sigma_{N;\alpha\beta} = \frac{\mu_0}{4\pi}e^2c^2\left\langle\!\left\langle\left(\frac{\vec{\alpha}_N \times \vec{r}_N}{r_N^3}\right)_\alpha ; \left(\vec{\alpha}_N \times \vec{r}_0\right)_\beta\right\rangle\!\right\rangle_0, \tag{40}$$

$$J_{MN;\alpha\beta} = \left(\frac{\mu_0}{4\pi}ec\hbar\right)^2\frac{\gamma_M\gamma_N}{h}\left\langle\!\left\langle\left(\frac{\vec{\alpha}_M \times \vec{r}_M}{r_M^3}\right)_\alpha ; \left(\frac{\vec{\alpha}_N \times \vec{r}_N}{r_N^3}\right)_\beta\right\rangle\!\right\rangle_0. \tag{41}$$

Here $\langle\!\langle\hat{P};\hat{Q}\rangle\!\rangle_0$ is the static polarization propagator or the static linear response function of the operators $\hat{P}$ and $\hat{Q}$:

$$\langle\!\langle\hat{P};\hat{Q}\rangle\!\rangle_0 = \sum_{n\neq 0}\frac{\left\langle\boldsymbol{\Psi}^{(0)}\left|\hat{\mathbf{P}}\right|\boldsymbol{\Phi}^{(n)}\right\rangle\left\langle\boldsymbol{\Phi}^{(n)}\left|\hat{\mathbf{Q}}\right|\boldsymbol{\Psi}^{(0)}\right\rangle}{E^{(0)} - E^{(n)}}. \tag{42}$$

In fact, the polarization propagator describes the response of a molecule to an external perturbation. Such a response function represents a first-order change in the mean value of the quantum operator $\hat{P}$ over the ground state under the action of a static perturbation $\hat{Q}$. From now on, the notations $\langle\!\langle\mathrm{P};\mathrm{Q}\rangle\!\rangle_0^{S/T}$ and $\langle\!\langle\mathrm{P},\mathrm{Q},\mathrm{R}\rangle\!\rangle_0^{S/T}$ are referred to, respectively, as the linear and quadratic response functions of singlet ($S$) or triplet ($T$) types. The dimension of $\sigma_{N;\alpha\beta}$ and $K_{MN;\alpha\beta}$ is determined by the dimension of the characteristic matrix elements such as $\left\langle\boldsymbol{\Psi}^{(0)}{}_{1\times 4}\left|\hat{\mathbf{P}}_{4\times 4}\right|\boldsymbol{\Phi}^{(n)}{}_{4\times 1}\right\rangle$ that give scalars of $1 \times 1$. From the general structure of the tensors $\sigma_{N;\alpha\beta}$ and $K_{MN;\alpha\beta}$, it can be noted that they are of the "paramagnetic type", and the main contributions, which are distinguishable at the nonrelativistic level, seem to be inseparable in the relativistic domain. The only thing that has been done so far is the splitting of the diamagnetic- and paramagnetic-type terms apart in both cases. This was done by Aucar [115], who showed that the polarization propagator can be divided into two parts:

$$\langle\!\langle\hat{P};\hat{Q}\rangle\!\rangle_0 = \langle\!\langle\hat{P};\hat{Q}\rangle\!\rangle_{0;ee} + \langle\!\langle\hat{P};\hat{Q}\rangle\!\rangle_{0;pp}. \tag{43}$$

The first and second terms of the polarization propagator (43) involve the orbital rotations between the orbitals with positive and negative energies, respectively. The first term (*ee*) gives all contributions to the paramagnetic-type parts of the shielding and spin–spin coupling tensors, while the second term (*pp*) gives the corresponding diamagnetic counterparts.

The operators corresponding to various types of relativistic effects on NMR parameters can be expressed explicitly only when going from a four-component to a two-component representation. The transition to the two-component level is carried out through the block-diagonalization of the Dirac Hamiltonian by means of a unitary transformation $\hat{\mathbf{U}}$, which transforms the four-component spinors with positive and negative energies, $\boldsymbol{\Psi}^{(+)}$ and $\boldsymbol{\Psi}^{(-)}$ into the spinors with zero small and large components, respectively:

$$\hat{\mathbf{U}}^+\begin{bmatrix}\hat{\mathbf{h}}_{LL} & \hat{\mathbf{h}}_{LS} \\ \hat{\mathbf{h}}_{SL} & \hat{\mathbf{h}}_{SS}\end{bmatrix}\hat{\mathbf{U}} = \begin{pmatrix}\hat{\tilde{\mathbf{h}}}_+ & \mathbf{0}_{2\times 2} \\ \mathbf{0}_{2\times 2} & \hat{\tilde{\mathbf{h}}}_-\end{pmatrix}, \quad \hat{\mathbf{U}}^+\hat{\mathbf{U}} = \mathbf{I}_{4\times 4}, \tag{44}$$

$$\hat{\mathbf{U}}^{+}\left(\begin{array}{c} \boldsymbol{\psi}_{L}^{(+)} \\ \boldsymbol{\psi}_{S}^{(+)} \end{array}\right)=\left(\begin{array}{c} \widetilde{\boldsymbol{\psi}}^{(+)} \\ \mathbf{0}_{2\times 1} \end{array}\right), \ \hat{\mathbf{U}}^{+}\left(\begin{array}{c} \boldsymbol{\psi}_{L}^{(-)} \\ \boldsymbol{\psi}_{S}^{(-)} \end{array}\right)=\left(\begin{array}{c} \mathbf{0}_{2\times 1} \\ \widetilde{\boldsymbol{\psi}}^{(-)} \end{array}\right). \tag{45}$$

The original form of such a transformation was proposed by Foldy and Wouthuysen (FW) [135]. In the case of free particles, the FW transformation converts the system of Equations (28) into a pair of independent equations for large and small components. However, in the presence of the Coulomb potential, FW constructs singular operators that are not applicable in variational calculations. The most general form for that case was proposed by Heully [136] in terms of kinetic balance operator. It projects small components onto the space of large components and can easily be expressed from the second equation of the system (28):

$$\boldsymbol{\psi}_{S}=\hat{\mathbf{R}}\boldsymbol{\psi}_{L}, \ \hat{\mathbf{R}}=\frac{1}{2mc}\left(1+\frac{E-\hat{V}}{2mc^{2}}\right)^{-1}\vec{\boldsymbol{\sigma}}\hat{\vec{\pi}}. \tag{46}$$

Heully's transformation consists of two matrix operators, one of which directly diagonalizes the Hamiltonian, a second one which is responsible for the renormalization of the spinor. Thus, in general, the transformation operator $\hat{\mathbf{U}}$ has the form:

$$\hat{\mathbf{U}}=\hat{\mathbf{W}}_{1}\hat{\mathbf{W}}_{2}, \ \hat{\mathbf{W}}_{1}=\left(\begin{array}{cc} \mathbf{I} & -\hat{\mathbf{R}}^{+} \\ \hat{\mathbf{R}} & \mathbf{I} \end{array}\right), \ \hat{\mathbf{W}}_{2}=\frac{1}{\sqrt{\mathbf{I}+\hat{\mathbf{R}}^{+}\hat{\mathbf{R}}}}\left(\begin{array}{cc} \mathbf{I} & \mathbf{0} \\ \mathbf{0} & \mathbf{I} \end{array}\right). \tag{47}$$

If the exact operator $\hat{\mathbf{R}}$ is known, then the application of the transformation (47) to the Dirac Hamiltonian leads to its exact block diagonalization in one step, and the solution of the equation with the modified upper left Hamiltonian $\hat{\tilde{\mathbf{h}}}_{+}$ exactly reproduces the solution of the original four-component Dirac equation. Finding the exact operator via the Equation (46) is not possible as it contains the energy in an explicit way. The equations for the operator $\hat{\mathbf{R}}$ and its Hermitian conjugation $\hat{\mathbf{R}}^{+}$ are derived from the requirement that the non-diagonal elements of the transformed Hamiltonian are equal to zero. Thus, a system of two-component nonlinear operator equations containing all elements of the original Dirac Hamiltonian as "coefficients" is obtained. Reformulating this system of equations in the matrix form and finding its solutions is a central idea of modern exact two-component quasi-relativistic methods, generally called X2C. Equation (46) was used in an approximate form by Fukui [100] to obtain the positive-energy two-component Breit–Pauli Hamiltonian $\hat{\mathbf{H}}_{+}^{BP}$ for the case of many particles for the purpose of deducing the leading relativistic contributions to the nuclear shielding tensor. In Fukui's work, the operator $\hat{\mathbf{R}}$ from Equation (48) was reduced to the following expression:

$$\hat{\mathbf{R}}=\frac{1}{2mc}\vec{\boldsymbol{\sigma}}\hat{\vec{\pi}}, \tag{48}$$

where $\hat{\vec{\pi}}=\hat{\vec{p}}+e\vec{A}$ is the momentum operator extended for the presence of the magnetic field. In this form, the operator $\hat{\mathbf{R}}$ is called the magnetic restricted kinetic balance (MRKB). Substituting (48) into Equation (47) yields an approximate unitary transformation that converts the four-component Hamiltonian into the block-diagonal form. The resulting Hamiltonian $\hat{\mathbf{H}}_{+}^{BP}$ contains all the relativistic hyperfine interactions that define the nuclear magnetic shielding tensors and the SSCCs at the non-relativistic level and a large number of additional terms representing the relativistic corrections to NMR parameters. The expressions for tensors $\sigma_{N;\alpha\beta}$ and $K_{MN;\alpha\beta}$ in the two-component representation, in fact, do not differ from the classical non-relativistic definitions, except for the energy, which is expressed in terms of an average value of the Hamiltonian $\hat{\mathbf{H}}_{+}^{BP}$ over the perturbed positive-energy many-body ground state $\mathbf{\Psi}_{+}$:

$$\sigma_{N;\alpha\beta}=\left[\frac{\partial^{2}}{\partial B_{\alpha}\partial\mu_{N;\beta}}\left\langle\mathbf{\Psi}_{+}(\vec{B},\vec{\mu}_{N})\left|\hat{\mathbf{H}}_{+}^{BP}(\vec{B},\vec{\mu}_{N})\right|\mathbf{\Psi}_{+}(\vec{B},\vec{\mu}_{N})\right\rangle\right]_{\vec{B}=\vec{\mu}_{N}=\vec{0}}, \tag{49}$$

$$K_{MN;\alpha\beta} = \left[ \frac{\partial^2}{\partial\mu_{M;\alpha}\partial\mu_{N;\beta}} \left\langle \mathbf{\Psi}_+(\vec{\mu}_M, \vec{\mu}_N) \middle| \hat{\mathbf{H}}_+^{BP}(\vec{\mu}_M, \vec{\mu}_N) \middle| \mathbf{\Psi}_+(\vec{\mu}_M, \vec{\mu}_N) \right\rangle \right]_{\vec{\mu}_M = \vec{\mu}_N = \vec{0}}. \tag{50}$$

Perturbed positive-energy ground state wavefunction $\mathbf{\Psi}_+(\vec{B}, \vec{\mu}_N)$ is expanded into the Taylor series by the powers of magnetic moments of the nuclei $\vec{\mu}_N$ and the external magnetic field $\vec{B}$. The substitution of the perturbed $\mathbf{\Psi}_+(\vec{B}, \vec{\mu}_N)$ as a power series to an average value of $\hat{\mathbf{H}}_+^{BP}(\vec{B}, \vec{\mu}_N)$ in Equations (49) and (50) gives the positive energy $E_+(\vec{B}, \vec{\mu}_N)$ in a form of infinite power series of magnetic moments of the nuclei and the external magnetic field. The second derivatives of $E_+(\vec{B}, \vec{\mu}_N)$ relative to $\vec{B}$ and $\vec{\mu}_N$ at $\vec{B} = \vec{\mu}_N = \vec{0}$ in Equation (49) and to $\vec{\mu}_M$ and $\vec{\mu}_N$ at $\vec{\mu}_M = \vec{\mu}_N = \vec{0}$ in Equation (50) give the coefficients, which are bilinear on both perturbations. According to Fukui's mathematical deductions, the diamagnetic component of the nuclear shielding tensor at the two-component level can be represented as a sum of three terms:

$$\sigma_{N;\alpha\beta}^{dia} = \sigma_{N;\alpha\beta}^{dia}(\text{DS}) + \sigma_{N;\alpha\beta}^{dia}(\text{DS}, \text{ROO}) + \sigma_{N;\alpha\beta}^{dia}(\text{ROO}). \tag{51}$$

The first term corresponds to the standard nonrelativistic diamagnetic contribution, which is an average value of the diamagnetic operator DS over the undisturbed ground state. The other two are the relativistic corrections. These corrections involve both the standard DS operator and different parts of the retarded orbit-orbit interaction operator (ROO), which, in particular, contains Darwin's operator (Dar). It is worth noting that the correction $\sigma_{N;\alpha\beta}^{dia}(\text{DS}, \text{ROO})$ is a singlet linear response function $\langle\langle \text{DS}; \text{ROO} \rangle\rangle_0^S$.

The paramagnetic component is divided into a large number of contributions, which can be formally divided into four types of terms:

$$\sigma_{N;\alpha\beta}^{para} = \sigma_{N;\alpha\beta}^{para}(\text{OP}) + \sigma_{N;\alpha\beta}^{para}(\text{FC}) + \sigma_{N;\alpha\beta}^{para}(\text{SD}) + \sigma_{N;\alpha\beta}^{para}(\text{ROO}). \tag{52}$$

All contributions of the first type $\sigma_{N;\alpha\beta}^{para}(\text{OP})$ necessarily include the classical orbital paramagnetic operator (OP), $\sum_i r_{iN}^{-3}\hat{L}_{iN\alpha}$. One of the contributions of this type is the non-relativistic paramagnetic contribution $\langle\langle \text{OP}; \text{OZ} \rangle\rangle_0^S$, the rest are the relativistic corrections of two types, namely $\langle\langle \text{OP}; \text{ROO} \rangle\rangle_0^S$ and $\langle\langle \text{OP}; \text{OZ}; \text{ROO} \rangle\rangle_0^S$. The contributions of the types $\sigma_{N;\alpha\beta}^{para}(\text{FC})$ and $\sigma_{N;\alpha\beta}^{para}(\text{SD})$ contain Fermi-contact and spin-dipole interaction operators and share a similar general structure. They can be represented as the sum of the contributions of four types: $\langle\langle \text{FC/SD}; \text{SO}; \text{OZ} \rangle\rangle_0^T$, $\langle\langle \text{FC/SD}; \text{SO} \rangle\rangle_0^T$, $\langle\langle \text{FC/SD}; \text{SZ}; \text{SO} \rangle\rangle_0^T$, and $\langle\langle \text{FC/SD}; \text{MV} \rangle\rangle_0^T$. These are the triplet response functions containing not only the FC or SD triplet operators, but also the spin-orbit interaction operator (SO), orbital Zeeman operator (OZ), spin Zeeman operator (SZ), and mass-velocity operator (MV). It is worth noting that the relativistic correction of the type $\langle\langle \text{FC/SD}; \text{SO}; \text{OZ} \rangle\rangle_0^T$ is of particular importance, since the relativistic effect of a heavy atom on the shielding constant of a light atom, the so-called heavy atom on light atom effect (HALA) [122,133,137–140], is almost completely described by this term. The last term in (52) is a singlet-type response function $\langle\langle \text{ROO}; \text{OZ} \rangle\rangle_0^S$. Thus, within the two-component formalism, the shielding tensor $\sigma_{N;\alpha\beta}$ includes the nonrelativistic paramagnetic and diamagnetic contributions and a great number of relativistic corrections in the form of both linear and quadratic response functions containing various combinations of NMR operators (DS, OP, OZ, FC, SD) as well as the standard relativistic operators such as SO, MV, ROO (Dar).

For the SSCCs tensor $K_{MN;\alpha\beta}$, there are many more types of relativistic corrections than that to the shielding tensor $\sigma_{N;\alpha\beta}$. For the most part, these are either triplet linear response functions or singlet-triplet quadratic response functions. For example, nine types of third-order relativistic corrections to $K_{MN;\alpha\beta}$ are determined by the following

response functions: $\langle\langle\mathrm{PSO}_M;\mathrm{PSO}_N;\mathrm{MV}\rangle\rangle_0^S$, $\langle\langle\mathrm{PSO}_M;\mathrm{PSO}_N;\mathrm{Dar}\rangle\rangle_0^S$, $\langle\langle\mathrm{FC}_M;\mathrm{FC}_N;\mathrm{MV}\rangle\rangle_0^{ST}$, $\langle\langle\mathrm{SD}_M;\mathrm{SD}_N;\mathrm{MV}\rangle\rangle_0^{ST}$, $\langle\langle\mathrm{FC}_M;\mathrm{FC}_N;\mathrm{Dar}\rangle\rangle_0^{ST}$, $\langle\langle\mathrm{SD}_M;\mathrm{SD}_N;\mathrm{Dar}\rangle\rangle_0^{ST}$, $\langle\langle\mathrm{SD}_M;\mathrm{FC}_N;\mathrm{MV}\rangle\rangle_0^{ST}$ $+M\rightleftarrows N$, $\langle\langle\mathrm{FC}_M;\mathrm{PSO}_N;\mathrm{SO}\rangle\rangle_0^{ST}+M\rightleftarrows N$, $\langle\langle\mathrm{SD}_M;\mathrm{PSO}_N;\mathrm{SO}\rangle\rangle_0^{ST}+M\rightleftarrows N$, among which only the first two are the singlet response functions with spinless operators. The rest are the singlet-triplet response functions, including both singlet and triplet operators. All currently known relativistic corrections to the tensor obtained from the positive-energy Breit–Pauli Hamiltonian are presented in Manninen's works [124–126].

Based on the operator structure of relativistic corrections to NMR parameters, they can be divided into two categories: scalar or spin-free and spin-dependent (in particular, spin-orbital). Scalar relativistic effects arise from the corrections to kinetic energy caused by the relativistic increase of the mass of electrons at high velocities (mass-velocity operator) and from the corrections to the centrifugal Coulomb potential (Darwin operator), which occur due to the spontaneous creation and annihilation of the electron-positron pairs, resulting in small irregular fluctuations of the electrons around their average positions. The latter phenomenon was called as "Zitterbewegung" [141]. Spin-orbit effects reflect the influence on the NMR parameters of the interaction of spins of electrons with their angular momenta.

Relativistic effects can also be divided into the direct and indirect effects. Direct relativistic effects are due to the incompleteness of the non-relativistic representation of the physical operators in the hyperfine NMR Hamiltonian. Accordingly, direct relativistic effects can be estimated from the difference in NMR parameters calculated using the relativistic NMR Hamiltonian, which explicitly includes the speed of light, and those obtained in the nonrelativistic limit obtained by the increasing the speed of light to infinity, which, in practice, is very well reached by increasing the speed of light by several times. Indirect relativistic effects on NMR parameters include all other types of relativistic effects that can affect their values. For example, for the systems containing heavy elements, the NMR parameters calculated at the equilibrium geometry, which were optimized at the relativistic level of theory, will noticeably differ from those obtained at the nonrelativistic geometry.

## 3. Quantum Chemical Methods for Calculating NMR Parameters

### 3.1. Configuration Interaction Methods

The configuration interaction (CI) method [142–145] is most likely one of the simplest nonempirical methods for solving the Schrödinger equation, which takes into account the effects of electronic correlation. In CI theory, the wave function is expressed as a linear combination of $N$-electron Slater determinants constructed from RHF orbitals:

$$\Psi_{\mathrm{CI}} = a_0\Phi_{\mathrm{HF}} + \sum_S a_S\Phi_S + \sum_D a_D\Phi_D + \sum_T a_T\Phi_T + \ldots = \sum_{i=0} a_i\Phi_i. \tag{53}$$

All configurations are constructed on the basis of the ground state wave function obtained within the Hartree–Fock approximation, $\Phi_{\mathrm{HF}}$. The excited determinants are obtained by the application of the singlet, doublet, triplet, etc., excitation operators to the wave function $\Phi_{\mathrm{HF}}$. This is equivalent to replacing one, two, three, etc., occupied spin-orbitals with the same number of vacant spin-orbitals. The CI-coefficients $a_0$, $a_S$, $a_D$, ... are found from the Schrödinger equation in matrix form:

$$\mathbf{Hc} = E\mathbf{Sc}, \tag{54}$$

where $H_{ij} = \langle\Phi_i|\hat{H}|\Phi_j\rangle$ is the Hamiltonian matrix within the basis of electron configurations, $S_{ij} = \langle\Phi_i|\Phi_j\rangle$ is the overlap matrix, and $\mathbf{c}$ is the vector-column of the CI-coefficients.

If all possible $N$-electron functions are included in the CI procedure (subject to spatial and spin symmetry restrictions), then the Schrödinger equation is solved exactly within the space spanned by the one-particle basis functions. In that case the method is called Full Configuration Interaction (FCI). The FCI method can successfully be applied to difficult cases where the ground state wave function cannot be adequately described within a single

electronic configuration, or to the calculation of the excited state properties, properties of the open-shell systems or those of the systems far from their equilibrium geometries. The FCI method provides exact results within a given finite one-electron basis set. Moreover, it is size-consistent and size-extensive, which results in the fact that the FCI method ensures the independence of the accuracy of correlation energy on the system size [146–150]. However, apart from these advantages, FCI have one great drawback. The dimension of full CI procedure grows factorially with the system size, so it is necessary to select only the most important *N*-electron determinants. Due to its extremely severe computer requirements, the FCI method is rarely used in the calculations of the NMR parameters, however, in some cases it serves as a calibration method for other quantum chemical approaches. In particular, the FCI method was used to calculate the helium–helium SSCC in the helium dimer [151], boron–proton SSCC in the hypothetical BH molecule [152] and deuteron–proton SSCC in the HD molecule [153], as well as the proton shielding constant in $H_2$ molecule within the GIAO approach [154].

A common method of reducing the computer requirements of the CI method consists of restricting its configurational space. The simplest and the most inaccurate approximation is the configuration interaction singles (CIS) [155,156]. In the CIS wave function, only the Hartree–Fock ground state wave function and the linear combination of singly excited configurations are included. CIS is a size-consistent method, which does not include electron correlation and, thus, is a rough excited-state analog of ground-state Hartree–Fock theory. CIS is a candidate for the simplest level of theory in the excited-state hierarchy of methods, although it fails to describe states that have important contributions from double (or higher) excitations [157]. The unrestricted (UCIS) or restricted open-shell (ROCIS) variant of the CIS method was presented by Maurice et al. [157].

The next approximation includes both singly and doubly excited configurations and is referred to as the configuration interaction singles and doubles (CISD) [158,159]. In particular, it should be noted that the CISD method represents an accurate approximation that takes into account 95–96% of the correlation energy [160]. However, there is a considerable drawback of the CISD method, which consists in the lack of size consistency [161] (as opposed to FCI), i.e., this method does not provide a correct scaling of the correlation energy with the increase of the number of electrons, which, in turn, may bring about considerable errors in the calculation of SSCCs and chemical shifts of large molecules. By a common definition, size consistency means that the calculated energy of two noninteracting subsystems is identical to the sum of the calculated energy of the two subsystems separately, as it should be. Specifically, for the CISD model, which is restricted by doubly excited configurations, the size-consistency error stems from the approximating higher excitations by the products of minor excitations [162]. For example, the quadruple excited coefficients are approximated by using the cluster condition, $c_{ijkl}^{abcd} \approx c_{ij}^{ab} c_{kl}^{cd}$. This results in the modified equation on the double-excited amplitudes, which can, therefore, be treated by approximating the "higher excitation terms" using a variety of schemes which introduce different size-consistency corrections. The simplest way of correcting the size-consistency error of the CISD method is to neglect the so-called exclusion principle violating (EPV) terms in the equation on the double-excited amplitudes [162]. This results in the scheme, which has a number of different names; some of them are the coupled electron pair approximation (CEPA(0)) [163] and linearized coupled pair many electron theory (L-CPMET) [164]. As a consequence of a complete exclusion of the EPV terms, the application of the CEPA(0) approximation usually leads to the overestimation of the effect of higher excitations by the corrected CISD method. An advanced approach was introduced by Kelly [165,166], who first proposed the idea to approximate the effect of EPV terms using the orbital energies. Based on the idea of representing the EPV contributions in terms of the orbital energies, Meyer [167,168] suggested several variants of the CEPA approximation, namely, CEPA(1) and CEPA(2). Different types of CEPA approximation were systematically compared by Koch and Kutzelnigg [169]. Another way of correcting for the size consistency error of truncated CI has been worked out by Ahlrichs and co-

workers [170], who related the CISD approach to the coupled electron pair approximation CEPA(1) and developed the coupled pair functional (CPF) approach. The CPF is based on a correct description of separated electron pairs and uses invariance requirements with regard to unitary transformations of equivalent orbitals of identical subsystems [159]. A comprehensive review on the corrections for the size-consistency to the CI methods has been published by Szalay [162].

As a matter of fact, in the calculations of NMR parameters, the CCSD model (see Section 3.2) is far more preferable than the CISD method or its corrected versions, because the CCSD method is genuinely size-consistent and has a similar computational cost ($\sim N^6$, with $N$ designating the number of basis set functions).

After singles and doubles, the most important determinants are triples and quadruples. Sequential inclusion of these into the CI formalism results in CISDT and CISDTQ models [171–174]. Harrison and Handy [160] found that the addition of triple substitutions (CISDT) recovers approximately an additional 1% of the basis set correlation energy compared to CISD (total of 95–96%), whereas the addition of all quadruples (CISDTQ) recovers more than 99%.

If it is not possible to do without the multiconfigurational approach and the size of the system does not allow resorting to super-expensive CI methods, there is a flexible way to circumvent the problem. This implies using the multiconfigurational self-consistent field (MCSCF) method [175,176], where the wavefunction is constructed in the same manner but the orbitals are variationally optimized simultaneously with the expansion coefficients of the determinants. This simultaneous optimization makes the MCSCF model well suited to treat the static correlation that arises from the near degeneracy of several configurations. In this case, the configuration coefficients depend on the molecular coefficients, which leads to an interconnected system of equations, which is resolved by an iterative procedure. Due to the fact that the number of possible configuration states is very large, even for the simplest diatomic molecules, various approximations are used to simplify the calculations. Within these, the configuration space is usually restricted by choosing the sets of upper occupied and lower vacant orbitals between which the electronic transitions are allowed. The main problem in this case is the correct choice of the necessary configurations. One of the most popular approaches is the complete active space self-consistent field (CASSCF) [175,176]. In this method, all active orbitals participate in the FCI wave function construction, and the resulting configurations are included in the MCSCF optimization procedure. Due to taking into account all of the possible excitations within the active space, size consistency of the CASSCF method is achieved [177]. The CASSCF is often used to generate reference states for other, improved multi-reference methods; for example, multi-reference configuration interaction or multi-reference perturbation theories, by which dynamical correlation can be included. Examples of the latter are complete active space perturbation theory (CASPT2) [178,179] or *N*-electron valence state perturbation theory (NEVPT2) [180–186]. The original formulations of multi-reference perturbation theory were not size consistent [187], including the CASPT2 method. However, with the right choice of the zeroth-order Hamiltonian [188], the problem of size inconsistency can be resolved, like for the NEVPT2 method, which can be formally regarded as the size consistent method [189].

A less costly version of the CASSCF method is the so-called restricted active space self-consistent field (RASSCF) method [190]. The RASSCF method requires manual selection of three separate regions of active orbital space, RAS1, RAS2, and RAS3. In practice, this made RASSCF an unpopular approach in the calculations of the NMR parameters. Moreover, the RASSCF method is not size consistent [191].

In general, it should be noted that, despite the lower computer resource requirements, the MCSCF methods have not gained the popularity that was expected in quantum chemical calculations of nuclear shielding constants and SSCCs. Only a limited number of MCSCF calculations of nuclear shielding constants/chemical shifts [192–195] and SSCCs [196–207] have been presented so far.

A successful attempt to incorporate the description of the relativistic effects into the CI methods has been made by Visscher et al. [208], who proposed fully relativistic all-electron self-consistent field calculations based on the Dirac–Coulomb Hamiltonian, performed on the three lowest lying states of the PtH molecule. The resulting four-component Dirac–Hartree–Fock molecular spinors were subsequently used in relativistic configuration interaction calculations on the five lower states of PtH.

A formalism for relativistic four-component MCSCF calculations on molecules was presented by Jensen [209]. The introduced formalism parallels a direct second-order restricted-step algorithm developed for nonrelativistic molecular calculations. It was found that the proposed efficient algorithm requires only twice the memory used by the largest nonrelativistic calculation in the equivalent basis, due to the complex arithmetic.

The attempts to take into account the relativistic effects on NMR properties in MCSCF calculations are very few. An interesting example was presented by Vaara et al. [118], who performed ab initio calculations at the SCF and MCSCF levels for the $^1$H and $^{13}$C shielding tensors in the hydrogen and methyl halides, considering relativistic spin-orbit (SO) effects. The SO corrections were calculated analytically from the quadratic response functions using self-consistent field and multiconfiguration self-consistent field reference wave functions.

### 3.2. Coupled Clusters Methods

The coupled clusters (CC) method [210–230] is one of the most accurate and reliable ab initio approaches, allowing for the effects of electron correlation. The CC theory takes a special place in quantum chemistry, because it results in a hierarchy of approximate models, which provide a systematical convergence towards the FCI results, while maintaining the dimensional extensivity at each hierarchical level. The CC method differs from the CI method in the way of constructing the wave function. Within the CC framework, the wave function is represented in an exponential form, which can also be expressed as an infinite series of the excited determinants [231,232]:

$$\Psi_{\text{CC}} = e^{\hat{T}}\Phi_0 = (1 + \hat{T} + \frac{1}{2!}\hat{T}^2 + \frac{1}{3!}\hat{T}^3 + \ldots)\Phi_0. \tag{55}$$

The operator $\hat{T}$ in Equation (55) consists of the sum of the operators of different excitation classes:

$$\hat{T} = \hat{T}_1 + \hat{T}_2 + \hat{T}_3 + \ldots, \tag{56}$$

$$\hat{T}_1 = \sum_{i,a} t_i^a \hat{\tau}_i^a, \ \hat{T}_2 = \sum_{a>b}\sum_{i>j} t_{ij}^{ab} \hat{\tau}_{ij}^{ab}, \ \ldots \tag{57}$$

The operators $\hat{\tau}_i^a$, $\hat{\tau}_{ij}^{ab} \ldots$ are the operators of single, double, etc., excitations, and and $t_i^a$, $t_{ij}^{ab}$, $\ldots$ are the coupled cluster amplitudes. In order to find the cluster amplitudes, a system of equations is constructed by multiplying the Schrödinger equation from the left by the excited configurations of different classes:

$$\langle \Phi_{\text{S}} | e^{-\hat{T}} \hat{H} e^{\hat{T}} | \Phi_0 \rangle = 0, \tag{58}$$

$$\langle \Phi_{\text{D}} | e^{-\hat{T}} \hat{H} e^{\hat{T}} | \Phi_0 \rangle = 0, \tag{59}$$

$$\langle \Phi_{\text{T}} | e^{-\hat{T}} \hat{H} e^{\hat{T}} | \Phi_0 \rangle = 0. \tag{60}$$

These equations are solved iteratively until the desired accuracy is reached. Once the cluster amplitudes are determined, one can calculate the ground state energy using the following equation:

$$E_{\text{CC}} = \langle \Phi_0 | e^{-\hat{T}} \hat{H} e^{\hat{T}} | \Phi_0 \rangle. \tag{61}$$

The classification of the CC models is based on the excitation classes, taken into account within the main cluster $\hat{T}$. Due to the cross-terms, which arise when expanding the exponential operator $e^{\hat{T}}$, the wave function $\Psi_{\text{CC}}$ contains the configurations that correspond

to higher excitations classes than those included in operator $\hat{T}$. Thus, reducing the operator $\hat{T}$ down to a single excitation class, $\hat{T} = \hat{T}_1$, yields the so-called coupled clusters singles (CCS) scheme. This scheme is not equivalent to the Hartree–Fock method, but to the configuration interaction singles (CIS). Within CCS approximation, the excitation energies for states, which are dominated by single replacements of one spin-orbital in the Hartree–Fock reference determinant, are obtained correctly through the first order in the electron-electron interaction. A sequential expansion of the operator $\hat{T}$ with double, triple, and quadruple excitation classes ($\hat{T} = \hat{T}_1 + \hat{T}_2$, $\hat{T} = \hat{T}_1 + \hat{T}_2 + \hat{T}_3$ and $\hat{T} = \hat{T}_1 + \hat{T}_2 + \hat{T}_3 + \hat{T}_4$) leads to the coupled clusters singles and doubles (CCSD) [149], the coupled clusters singles, doubles, and triples (CCSDT) [230,233] and the coupled clusters singles, doubles, triples, and quadruples (CCSDTQ) [234–236] models, respectively. The formal computational scaling for the CCS, CCSD, CCSDT, and CCSDTQ schemes with respect to number of basis set functions $N$ is as follows [217,237]: $N^4$, $N^6$, $N^8$, $N^9$. The CCSD scheme has received a significant attention, but is often too expensive to be useful for molecules with more than 10 atoms. The CCSD provides high accuracy for many challenging response properties and is usually considered as a very accurate method for calibration of the other inferior computational methodologies. The CCSDT and higher-ranking pure CC schemes are out of routine use for today due to their dramatic scaling.

To reduce the computational costs of the highly demanding coupled cluster schemes, intermediate mixed schemes were introduced. One of the most popular approximate schemes is the CC2 [222]. Within this model, the equations for the amplitudes for single excitations are the same as in the CCSD method, but the equations for the amplitudes for double excitations are approximated so that they are accurate only up to the first order according to the perturbation theory in the fluctuation potential. Thus, the CC2 model is intermediate between the CCS and CCSD models. The computational cost of CC2 model can be expressed as $N^5$ with respect to the number of basis set functions $N$ [141]. Another approximate model, which was built on the similar concept, is the CC3 model [238]. The CC3 represents an intermediate model between the CCSD and CCSDT schemes, so that the computational scalability of CC3 is $N^7$ [217]. In general, the main principle of building the approximate $CC_n$ ($n > 1$) models is based on reducing the cluster equations for the $n$-fold cluster excitation amplitudes to the lowest non-vanishing order in the perturbation theory [217].

Though the effects of triple excitations on the NMR parameters are in most cases not particularly pronounced, for the special cases it was very important to develop a flexible scheme which takes into account triple excitations at the lowest possible computational costs. Apart from the CC3 scheme, which handles the triples in an iterative manner, the so-called noniterative perturbative CCSD(T) approach for the treatment of triple excitations was introduced. In the CCSD(T) scheme, the triples amplitudes are estimated from the triples excitation equations as they occur in the lowest non-vanishing order for the Möller–Plesset perturbational theory [217]. Overall, the CCSD(T) model has been proven successful and is used for the accurate prediction of many properties nowadays. The scaling of the CCSD(T) scheme with respect to the number of basis sets functions is the same as that of the CC3 model, namely $N^7$. However, the CCSD(T) has not been adopted for the calculations of SSCCs, because it has the triplet instability issue [239,240], which occurs when calculating the triplet FC and SD contributions to SSCCs. As opposed to the CCSD(T) model, the second derivatives within the CC3 scheme can be computed in two different ways, namely either with orbital relaxation effects explicitly included (the so-called "relaxed" CC3) or with the orbital relaxation effects excluded (the so-called "unrelaxed" CC3). The "unrelaxed" CC3 scheme circumvents possible problems with the triplet instabilities [224] and can be successfully applied to the calculation of the triplet properties such as FC and SD terms in SSCCs. To reduce the computational costs of CC methods, the resolution of the identity (RI) approximation [241,242] for two-electron integrals was applied [243,244], however, in practice, this is relevant only for the calculations of equilibrium geometries, harmonic frequencies, energy gradients, and some other first-order properties for now.

To calculate the second-order molecular properties, which require the summation over the excited states within the framework of a linear response function, either the coupled clusters linear response (CCLR) [210–214,245,246] or equation-of-motion coupled clusters (EOM-CC) [247–249] are used.

What is currently missing from the perspective of accurate NMR studies is the relativistic coupled cluster models for calculating NMR parameters at the two- or four-component levels of theory. However, work in this direction is being pursued by Gauss and co-workers [250,251].

The coupled cluster methods are widely used nowadays to calculate the parameters of NMR spectra of various small and medium-sized molecular systems. Thus, of all the models, the CCSD scheme turned out to be the most popular for calculating various types of SSCCs [205,252–254]. The introduction of the GIAO formalism into the coupled cluster theory was originally presented by Gauss et al. [255,256] at the CCSD level. The implementation of the GIAO formalism to the CCSD(T) and CCSDT models was proposed by Gauss and Stanton in works [219,257] and [258], respectively. The GIAO-CCSD(T) scheme proved to be particularly effective and has been widely used in the calibration calculations, in particular, for the purpose of establishing the accurate absolute scales for the NMR shielding constants [259], as well as for resolving entangled structural problems [260–262].

Among the recent works reporting on the CC calculations of the NMR properties is the work by Faber and Sauer [263]. They investigated the basis set convergence of nuclear SSCCs at the CCSD level of theory for 10 difficult molecules. Test molecules were chosen as each molecule has fluorine atom(s) and/or double or triple bonds, which are typically associated with large non-contact contributions. The results were obtained using Benedikt's hierarchy of basis sets (aug-)ccJ-pVXZ, X = D, T, Q, 5 [264] and Jensen's (aug-)pcJ-$n$, $n$ = 1, 2, 3 basis sets [265]. The aug-ccJ-pVXZ basis sets were constructed manually in that work, by adding the diffuse functions taken from the aug-cc-pVXZ basis set [266] to the original ccJ-pVXZ basis set. The accuracies of commensurate basis sets were compared to each other in the CCSD calculations of the one-, two-, and three-bond SSCCs of different types, involving $^1$H, $^{13}$C, $^{15}$N, $^{17}$O, and $^{19}$F nuclei. The CCSD method applied in conjunction with the uncontracted aug-ccJ-pVTZ basis set was found to be very accurate for calculating the $^1J_{CF}$. This follows from the fact that the estimated errors for the spin–spin coupling constants $^1J_{CF}$ turned out to be about 2.0 Hz, given that the values of this type of coupling constants usually exceed 200 Hz. For the two- and three-bond couplings involving 1–2 row elements, it was found that it is quite important to add diffuse functions. With the diffuse functions added to the ccJ-pVXZ and pcJ-$n$ basis sets, the CCSD method gives very good results. In particular, for the $^2J_{CF}$ SSCCs, calculated at the CCSD/aug-ccJ-pVTZ level, the typical error was found to be only 0.6 Hz. It was also shown that if the higher accuracy is needed within the CCSD framework, the basis set error can be reduced by roughly a factor of two by going to the quadruple zeta basis set.

The molecules discussed in [263] have previously been investigated by Del Bene et al. [253,267], using the EOM-CCSD method in conjunction with the qzp [268] basis set. The results presented by Faber and Sauer occurred to be similar to those obtained by Del Bene et al., with the exception of the $^1J_{CN}$ in HCN and the SSCCs of F$_2$CO, where the results of Del Bene deviated significantly from those of Faber and Sauer and the experimental data.

The importance of triples contributions to NMR spin–spin coupling constants computed at the CC3 and CCSDT levels was investigated by Faber, Sauer, and Gauss [269]. The analytical implementation of CC3 second derivatives method using the spin-unrestricted approach was presented at the first time. This allowed for calculation of the SSCCs at the CC3 level of theory in a fully analytical manner. The calculations of one-, two-, and three-bond SSCCs in a number of small molecules and their fluorine substituted derivatives were carried out at the CCSD and CC3 levels using the aug-ccJ-pVTZ basis set. To study the triples effects beyond the CC3 level, the calculations of different types of SSCCs involving 1–2 row atoms were performed using the CC3 and CCSDT methods using the ccJ-pVDZ

basis set. The CC3 triples correction ($J_{CC3}$-$J_{CCSD}$) and the residual triples correction ($J_{CCSDT}$-$J_{CC3}$) to various one-bond nuclear spin–spin coupling constants are illustrated in Figure 1.

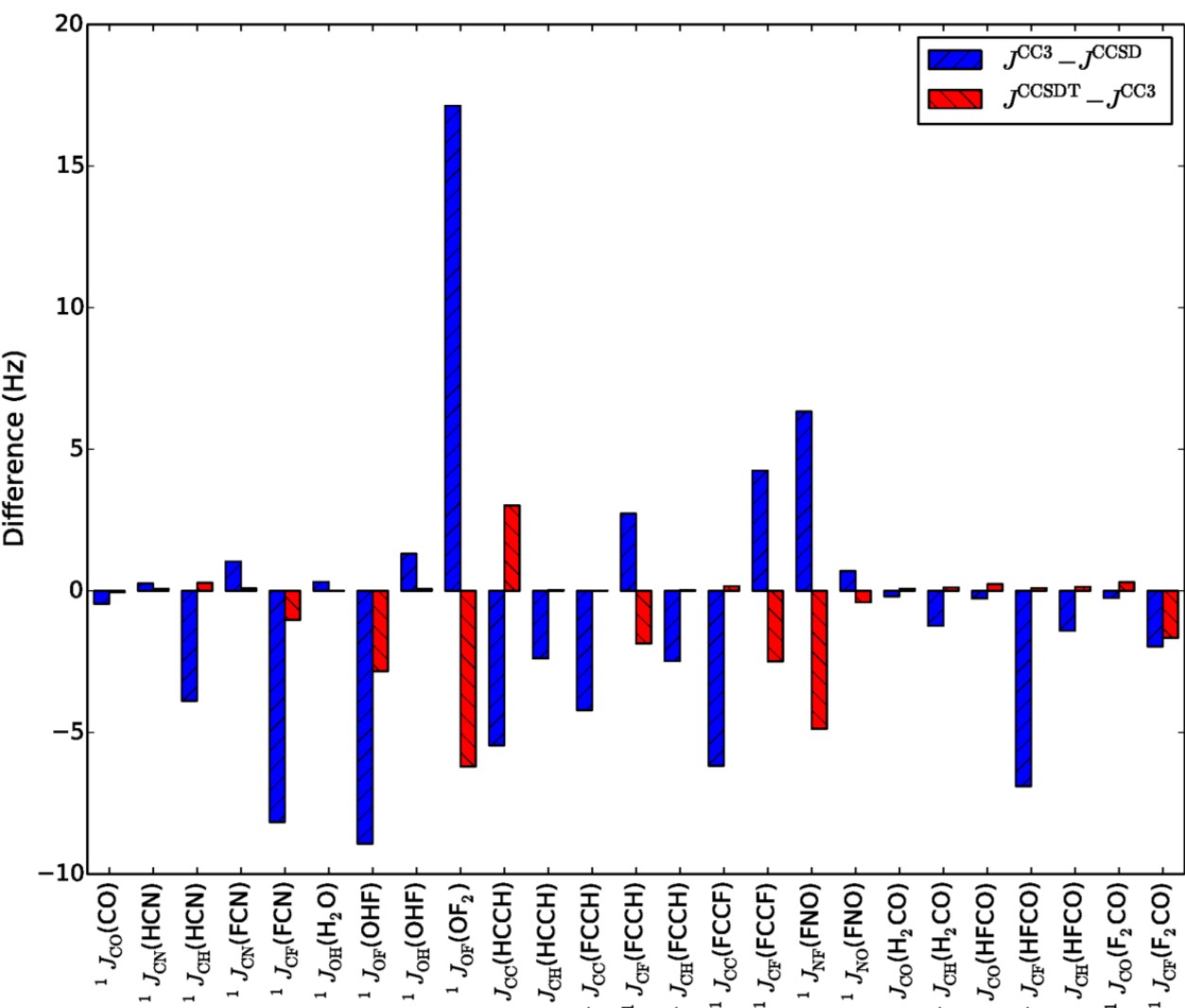

**Figure 1.** The CC3 triples correction (blue bars) and the residual triples correction (red bars) to various one-bond nuclear spin–spin coupling constants (in Hz). The CC3 corrections are calculated using aug-ccJ-pVTZ, whereas the residual corrections are calculated using the ccJ-pVDZ basis set. Reproduced from Ref. [254] with permission from American Chemical Society.

The change in the one-bond SSCCs when going from CCSD to CC3 level was found to be approximately 1–3%, with the exceptions of $^1J_{CN}$ in FCN and $^1J_{OF}$ in $OF_2$, where the correlation corrections to the CCSD results due to the triple excitations (triples) inclusion effect occurred to be as much as 15.7% and 6.4%, correspondingly. The changes in geminal and vicinal SSCCs due to the triples effect were found to be rather more significant as compared to the one-bond SSCC, with the differences of up to 10%, and even more—13.6%—for the $^3J_{FH}$ in fluoroacetylene. In these calculations, it was established that the most important contributions arising from the connected triple excitations in the coupled cluster expansion are accounted for at the CC3 level. Thus, the CC3 method is expected to become a standard approach for the calculation of reference values of the nuclear spin–spin coupling constants.

Jaszuński et al. [270] investigated the NMR shielding and spin–spin coupling constants of dinitrogen difluoride ($N_2F_2$), which represents an extremely challenging test for modern quantum chemical methods, within the CCSDT and CC3 models, respectively. For adequate

comparison with the experiment, vibrational and relativistic corrections were calculated. Coupled cluster methods were used with very large basis sets and complete basis set (CBS) extrapolations. Namely, for the calculation of NMR shielding constants, basis sets as large as aug-cc-pCV7Z were used. Spin–spin coupling constants have been determined with specialized versions of the correlation consistent basis sets ccJ-pVXZ, further augmented with diffuse functions. Calculated values of spin–spin coupling constants turned out to be in very good agreement with the experiment. To be more precise, for the *trans* isomer of dinitrogen difluoride, the final calculated values of $^1J_{NN}$ (−18.25 Hz), $^1J_{NF}$ (172.98 Hz) and $^2J_{NF}$ (−61.97 Hz) differ from the experimental values (−18.5, 172.8, and −62.8 Hz, respectively) by less than 1 Hz, while the calculated three-bond fluorine-fluorine spin–spin coupling constant, $^3J_{FF}$ (−303.61 Hz), was found to deviate from the experimental datum (−316.4 Hz) by only 12.79 Hz (that is 4% of the experimental value). For the *cis* isomer, the differences occurred to be consistently larger: between 2 and 4 Hz for the one- and two-bond couplings, $^1J_{NN}$, $^1J_{NF}$, $^2J_{NF}$, and about 19 Hz for the three-bond coupling, $^3J_{FF}$. The deviation between calculated (gas phase) and experimental (solvated) $^{19}$F shielding constants of the *cis* and *trans* isomers was found to be 15.7 and 11.1 ppm, respectively.

### 3.3. Density Functional Theory

The electron density functional theory (DFT) has become extremely popular in recent years in the application to the calculations of the second-order molecular properties, such as chemical shifts and SSCCs. DFT method takes into account electron correlation effects via the exchange-correlation (XC) potential and has moderate computational requirements at that. In this sense, one can hardly expect to find a more balanced approach than the DFT method.

The main idea of the density functional theory is to use an electron density matrix rather than a many-particle wavefunction when describing an electronic system. This leads to a significant simplification of the problem, since the many-particle wavefunction depends on $3N$ variables, where $N$ is the number of electrons, while the electron density is a function of only three spatial coordinates.

The formalism of the DFT theory is based on two Hohenberg–Kohn theorems [271]. According to the first Hohenberg–Kohn theorem, the ground state of a system of interacting particles is a unique functional of the electron density $\rho(\mathbf{r})$. From this theorem it follows that, for a given many-particle system, the external potential $v(\mathbf{r})$ and, thus, the Hamiltonian and thereby every ground state property of this system are determined only by the electron density $\rho(\mathbf{r})$. The second Hohenberg–Kohn theorem establishes a variational principle of quantum mechanics, which states that the electron density that minimizes the energy of the overall functional $E[\rho(\mathbf{r})]$ is the true electron density. This can be rephrased as follows: the energy of a given $N$-electron system, $E[\rho(\mathbf{r})]$, has a minimum equal to the ground state energy $E_0$, which implies that for any trial electron density function, such that $\int \rho_{\text{trial}}(\mathbf{r})d^3\mathbf{r}$ = $N$, the energy of the system, $E[\rho_{\text{trial}}(\mathbf{r})]$, must satisfy the inequality $E[\rho_{\text{trial}}(\mathbf{r})] \geq E_0$. The Kohn–Sham (KS) computational scheme [272] is based on the Hohenberg–Kohn theorems. In that scheme, the exact function of the ground state electron density of a given many-particle system is replaced by a function of non-interacting particles. In KS theory, the total energy is written as:

$$E(\rho) = -\sum_i^{occ} \int d^3\mathbf{r}\varphi_i^*(\mathbf{r})\frac{\nabla^2}{2}\varphi_i^*(\mathbf{r}) + \int d^3\mathbf{r}v_{ext}(\mathbf{r})\rho(\mathbf{r}) + \frac{1}{2}\int d^3\mathbf{r}\int d^3\mathbf{r}'\frac{\rho(\mathbf{r})\rho(\mathbf{r}')}{|\mathbf{r}-\mathbf{r}'|} + E_{xc}. \quad (62)$$

where the terms are, respectively, the non-interacting Kohn–Sham kinetic energy ($T_s$), the interaction energy with the external field ($E_{ext}$), the Hartree ($E_H$) and the exchange-correlation (XC), $E_{xc}$, energies. Thus, $E_{xc}$ is nothing more but the sum of errors originating from using the approximation of non-interacting particles to describe the kinetic energy term and $E_H$ instead of a real interelectronic interaction energy:

$$E_{xc}[\rho] = (T[\rho] - T_s[\rho]) + (E_{ee}[\rho] - E_H[\rho]) \quad (63)$$

The variational principle applied to the energy functional gives a system of Kohn–Sham equations:

$$\left[ -\frac{1}{2}\nabla^2 + v_{\text{ext}}(\mathbf{r}) + \int \frac{\rho(\mathbf{r}')}{|\mathbf{r} - \mathbf{r}'|} d\mathbf{r}' + V_{\text{xc}}(\mathbf{r}) \right] \varphi_i(\mathbf{r}) = \varepsilon_i \varphi_i(\mathbf{r}), \tag{64}$$

$$\rho(\mathbf{r}) = \sum_i^N |\varphi_i(\mathbf{r})|^2, \tag{65}$$

$$V_{\text{xc}}(\mathbf{r}) = \frac{\delta E_{\text{xc}}[\rho]}{\delta \rho}. \tag{66}$$

This system of nonlinear equations describes the behavior of non-interacting electrons in some effective local potential. For an exact exchange-correlation (XC) functional and, consequently, an exact local potential, the resulting orbitals give the exact energy and electron density function of the ground state. The Kohn–Sham equations have the same structure as the Hartree–Fock equations with the exchange potential replaced with the local exchange-correlation potential. Thus, the Kohn–Sham equations are also solved iteratively, i.e., through the self-consistency procedure. Since the exact exchange-correlation functional is unknown, there are a great number of approximations for it; detailed reviews on this topic can be found in the books [273–277]. Existing XC functionals can roughly be divided into four groups: the local density approximations (LDAs) [278–280], the generalized gradient approximations (GGAs) [281–284], the hybrid functionals [285,286], and meta-GGA functionals [287]. In addition, efforts have been made [288–292] to model the current dependency in the XC functional [293–296].

The simplest approximation is the local density approximation (LDA), within which it is assumed that the electron density is a function that changes slowly in space, in other words, an inhomogeneous system is approximated by a homogeneous density distribution. The GGAs depend not only on density, but also on its gradient. Such functionals provide a higher-order approximation that considers the change in the electron density when going from one point of space to another. The hybrid functionals incorporate a part of the exact Hartree–Fock (HF) exchange. The meta-GGAs depend on the electron density ($\rho$), its gradient ($\nabla \rho$) and on the corresponding Laplacian ($\nabla^2 \rho$).

The first implementation of the Kohn–Sham theory for the calculation of indirect nuclear spin–spin coupling constants was made by Malkin et al. [297–299] at the LDA and GGA levels of theory. They calculated the FC contribution to SSCCs using the finite perturbation theory (FPT), omitting the SD term. The PSO term was approximated by the sum-overstates approach. Dickson and Ziegler [300] also proposed early Kohn–Sham implementation of spin–spin coupling constants within the LDA approximation, neglecting the SD term. Their code exploited the Slater atomic orbitals, and has subsequently been made fully analytical at the GGA level of theory (with the SD term included).

Sychrovsky et al. [301] and Helgaker et al. [302] introduced the first fully analytical Kohn–Sham implementations of indirect spin–spin coupling constants, including four Ramsey contributions. Both implementations included hybrid DFT, in addition to LDA and GGA. A few years later, Watson et al. [303] presented the implementation of SSCCs at the hybrid level of theory using the Slater orbitals, with all four Ramsey terms included. The DFT-based perturbation theory was applied to compute spin–spin coupling tensors in extended systems, subject to periodic boundary conditions [304]. For reviews of the Kohn–Sham theory for the calculation of indirect nuclear spin–spin coupling constants, see the reviews by Malkin et al. [290] and the more recent ones by Alkorta and Elguero [305] and by Helgaker and Pecul [6–8].

Modern applications of the DFT theory to the calculation of NMR chemical shifts have been pioneered by Malkin et al. [288–290,306], using the IGLO method, and by Schreckenbach and Ziegler [69,307–310], using the GIAO approach. A number of other implementations, mostly based on the GIAO method, have also been presented [70,71,291,292].

An assessment of the aptness of various exchange-correlation functionals for calculation of the NMR parameters can be found in several sources, in particular, in Refs. [311–313] for chemical shifts and [314–318] for SSCCs.

The following XC functionals are commonly used for the prediction of the NMR properties in modern quantum chemical DFT calculations:

**GGAs:**
BP86 [281,282], PBE [319,320], BLYP [281,283], PW91 [321], B97-D [322], KT1 [323], KT2 [323,324], KT3 [324], HCTH93 [325], HCTH147 [326], HCTH407 [327], OPTX [328,329], OLYP [329], OPBE [330], OPW91 [328,331];

**Hybrid GGA functionals:**
O3LYP [329], B3P86 [281,282], B3PW91 [281,332], B3LYP [283,286], mPW3PBE [319,320,333], B97-1 [325,334], B97-2 [334,335], B97-3 [334,336], X3LYP [337], PBE0 [338–340], mPW1PW91 [332,333], mPW1LYP [283,333], mPW1PBE [319,320,333], B98 [334,341], B1LYP [342], PBEh1PBE [343], and BHandH [286];

**Meta-GGA functionals:**
TPSS [287], $\tau$HCTH [344], VSXC [345], M06-L [346], M11-L [347];

**Hybrid meta-GGA functionals:**
TPSSh [287,348], $\tau$HCTHhyb [344], M06-2X [349], B1B95 [350];
and

**Long-range corrected hybrid functionals:**
$\omega$B97 [351], $\omega$B97X [351], $\omega$B97X-D [352], CAM-B3LYP [353] and LC-$\omega$PBE [354].

Of the listed functionals, the Keal–Tozer functionals, KT$n$ ($n$ = 1–3), were optimized specifically for the uncoupled isotropic and anisotropic NMR shielding constants. They give the values for a series of challenging molecules involving first- and second-row atoms that are two to three times more accurate than those of commonly used GGAs [323].

The first Keal–Tozer functional, KT1, is expressed in the form:

$$E_{\text{xc}}^{\text{KT1}} = E_{\text{LDA}}^{\text{KT1}} + \gamma \sum_{\sigma} \int \frac{|\nabla \rho_{\sigma}(\mathbf{r})|^2}{\rho_{\sigma}^{4/3}(\mathbf{r}) + \delta} \mathrm{d}^3\mathbf{r} \tag{67}$$

In this expression, the index $\sigma$ designates $\alpha$- and $\beta$-spin densities $\rho_{\sigma}$. The parameters $\gamma$ and $\delta$ were optimized so as to reproduce the NMR shielding constants of a wide range of molecules as accurately as possible. As a result, the final values of the parameters $\gamma$ and $\delta$ are $-0.006$ and $0.1$, respectively. The functionals KT2 and KT3 have more complicated forms and depend on more variational parameters as compared to the KT1. They were created on the basis of KT1 in order to improve the description of other properties, such as ionization potentials, electron affinity, proton affinity, bond angles, bond lengths, electronic polarizability, thermodynamic properties, etc. Extensive testing carried out by Keal and Tozer [324] showed that, despite significant modifications, the KT2 and KT3 functionals are not inferior in accuracy relative to KT1 functional in the calculations of shielding constants.

The performance of the KT1 and KT2 exchange-correlation functionals have also been assessed by Keal, Tozer, and Helgaker [355] as compared to the other well-known popular functionals on the prediction of NMR shielding constants and indirect nuclear spin–spin coupling constants. The authors selected 14 different molecules with significant electron correlation effects, containing light main group nuclei. In line with previous observations, the KT1 and KT2 gave a significant improvement for NMR shielding constants over the conventional functionals. In particular, for isotropic shieldings, the KT1 and KT2 functionals outperformed the functionals BLYP and B3LYP by far, providing results more than twice as accurate as those of BLYP and B3LYP. They proved to be very successful for both charged and neutral species, and for equilibrium and non-equilibrium geometries. The improvement has been traced almost exclusively to the paramagnetic contribution. These functionals were also used to determine SSCCs for 11 molecules. The results occurred to be of variable quality and there was no improvement observed over the conventional functionals.

Zhang and his colleagues [330] have shown that the OLYP and OPBE, which use the OPTX as the exchange functional, exhibit remarkably good performance in the prediction of NMR shielding constants of $^{13}$C, $^{15}$N, $^{17}$O and $^{19}$F nuclei. At that, the OPBE was found to be the best method among the considered functionals, namely the B3LYP, PBE0, BLYP, PBE, OLYP, and OPBE. Moreover, since the parameters of OPTX functional were optimized by fitting to the unrestricted HF energies of the first- and second-row atoms, this functional (and so as the other functionals based on it) can be expected to give reliable excited state triplet properties in the sense of triplet instability issue.

The PBE0 functional appears to be the one of the most robust functionals among the hybrid functionals for the calculation of NMR chemical shifts. This hybrid functional is constructed from the PBE GGA functional, in which all parameters (except those related to the local spin density) are fundamental constants, and of 25% fraction of the Hartree–Fock exchange functional. The accuracy provided by the PBE and PBE0 functionals in the calculations of chemical NMR shifts has been studied in detail on a wide range of compounds by Adamo and Barone [356]. It was concluded that the PBE0 functional provides accuracy comparable to that of the MP2 method in the case of "well-behaving" systems and provides significant advantages over the other known functionals in cases with significant correlation effects.

Thirty-nine different exchange-correlation functionals covering several steps of Jacob's ladder [357] were tested in the DFT calculations of $^{1}$H and $^{13}$C NMR chemical shifts of six neutral and protonated alkylpyrroles by Zahn et al. [312] very recently. The considered functionals included 9 GGA functionals, 16 hybrid GGA functionals, 5 meta-GGA functionals, 4 hybrid meta-GGA, and 5 long-range corrected hybrid functionals. As stated by the authors, the aim of their study consisted in finding a DFT method that can outperform the MP2 and B3LYP methods in accuracy at predicting the chemical shifts for neutral and protonated alkylpyrroles. For $^{13}$C chemical shifts, it was found that most of the functionals perform better than B3LYP, with the hybrid meta-GGA functional TPSSh demonstrating the best performance. At that, there was no single functional found to outperform the MP2 method. For the $^{1}$H chemical shifts, it was found that all the considered functionals outperformed the MP2 and B3LYP methods, with the best performance demonstrated by TPSSh, likewise in the case of $^{13}$C chemical shifts.

An extensive analysis of performance of different functionals for the calculations of SSCCs has been performed recently by Sauer and his colleagues [317,318]. In particular, for the one-bond $J_{FC}$ SSCCs, it was shown that meta-GGA functional M06-L demonstrates surprisingly high accuracy, outperforming any other investigated functional, including the PBE0, otherwise considered one of the most reliable for this type of SSCCs. Although the computation of nuclear magnetic resonance (NMR) parameters involving F is known to be a challenging task [38], even with a rather small basis, such as pcJ-1, M06-L provided the results with a MAD = 11.7 Hz, whereas the MAD for the PBE0 results was assessed as much as 60.0 Hz. Providing that the $J_{FC}$ cover the range of about 300 Hz, the achieved accuracy can be considered as particularly remarkable. It was found that the accuracy of the M06-L/pcJ-1 scheme does not stem from a well-suited exchange or correlation part of the functional. Instead, Sauer et al. assumed that that high accuracy can arise from a fortuitous cancellation of errors, as revealed by investigating the convergence of the basis set. Their findings also indicated that $^{1}J_{FC}$ constants are highly dependent on the amount of exact exchange included in the expression of the functional, with large fractions being critically important to achieving satisfactory results. Sauer et al. have also studied the effects of the geometry on the $^{1}J_{FC}$ and found that optimizing the geometry at the same level of theory as used for the calculation of SSCCs generally improves the quality of the results.

In another work of Jaszuński, Świder, and Sauer [318] the through-space SSCCs involving fluorine atom(s) were investigated using the density functional theory with a special accent placed on the performance of various functionals against the experimental data. Namely, a diverse set functionals, including KT2, PBE, BP86, B97-D, BHandH, B3LYP, CAM-B3LYP, CAM-B3LYPx, and PBE0 were tested in the calculations of the through-

space F–F, F–N, F–C, F–P, and F–Se SSCCs in o-fluorobenzaldehyde oxime, flubenzimine, 2-trifluoromethylphenyl-difluorophosphine, and 8-Fluoro-1-(methylselanyl)naphthalene. The authors used the so-called *J*-oriented basis sets, specifically optimized for the calculations of SSCCS. In that way, they partially excluded one of the plausible sources of error making the choice of the functional more important than the choice of the basis set. As a result, they found noticeable differences between the performance of the studied DFT functionals that have been observed for the through-space and through-bond coupling constants. For the through-space couplings, the hybrid functionals and, in particular, the functionals with the highest amount of Hartree–Fock exchange, namely, the PBE0, CAM-B3LYPx, BHandH, and CAM-B3LYP, were found to perform the best. At that, the most popular B3LYP functional gave the worst results for the through-space coupling constants as compared to the other hybrid functionals. On the other hand, for the set of the considered through-bond couplings, the PBE0, B3LYP, and, to a minor degree, B97-D, functionals gave on average the best agreement with the experimental data, implying that there is no correlation with the amount of Hartree–Fock exchange in the functional. Overall, Jaszuński, Świder, and Sauer concluded that the PBE0 functional appears to be the most robust functional. Moreover, they stated that when using any of the *J*-oriented triple-zeta basis sets the choice of the functional becomes much more important than the choice of the basis set, and the performance of the functional, in its turn, depends on the type of the considered SSCC.

It is well known that indirect nuclear spin–spin coupling constants calculated using the restricted Hartree–Fock theory are unreliable since the usually dominant FC contribution and less significant SD contribution suffered from the triplet instability problem [239,240]. This problem originates from the fact that for some systems the unrestricted HF method yields lower ground-state energy than the ordinary restricted HF method. Any computational model that provides an unbalanced description of the ground state and the most important excited states of a given symmetry, also provides a poor description of the molecular property of interest that depends on these states. In this sense, the restricted Hartree–Fock theory proved to be totally unreliable. Although, in most cases the restricted Kohn–Sham theory produces a fairly accurate prediction of SSCCs, in some distinguished cases, such as near-biradical o-benzyne molecule [358], this theory might show the "symptoms" of triplet instability problem. A very important investigation of this issue has been performed by Lutnæs et al. [316]. They examined the sensitivity of spin–spin coupling constants to triplet instabilities in Kohn–Sham and Hartree–Fock theories by correlating the quality of the spin–spin coupling constants with the quality of the lowest triplet excitation energy for a number of small molecules. In general, it was found that the most stable results for the FC contributions are provided by the LDA approximation. Slightly less stability of the results was reached within the GGA approximation. The hybrid GGA theory was found to give results rather more susceptible to triplet instabilities than that of the pure GGA theory. For the calculations of SSCCs, Lutnæs et al. recommend the Perdew–Burke–Ernzerhof GGA exchange-correlation functional, PBE, which provides a good compromise of accuracy and robustness in the sense of triplet instability problem.

Taking into account the relativistic effects in the DFT theory for NMR properties when dealing with heavy element compounds has become a top priority for today. This is connected with the fact that DFT methods provide a fairly reliable description of the electron correlation effects in the calculations of NMR parameters at the computational cost similar to that of the Hartee–Fock method. As opposed to many computationally demanding ab initio correlated wavefunction-based approaches, the DFT can be regarded as the most promising methodology for the extension to the relativistic domain, as the computational scaling of the relativistic methods exceeds that of the nonrelativistic analogies by times or even dozens of times.

Komorovsky, Repisky, and their colleagues proposed one of the most general relativistic four-component density functional theories (4DFT) for the calculations of NMR shielding constants [359] and spin–spin coupling constants [360]. These theories are based

on the matrix formulation of the Dirac–Kohn–Sham (DKS) method and on the use of the restricted magnetic balance (RMB) condition for the small MO components.

The relativistic theories start from the expressions for the total energy in the presence of external magnetic perturbations $\vec{X}$ and $\vec{Y}$, with $\vec{X} = \vec{B}$, $\vec{Y} = \vec{\mu}^M$ for $\sigma$ and $\vec{X} = \vec{\mu}^M$, $\vec{Y} = \vec{\mu}^N$ for $J$, within the framework of the four-component Dirac–Kohn–Sham approach:

$$E(\vec{X}, \vec{Y}) = \left\langle \varphi_i^{(\vec{X},\vec{Y})} \middle| D_{kin}^{00} + D^{01} + D^{10} \middle| \varphi_i^{(\vec{X},\vec{Y})} \right\rangle + E_{pot}^{(\vec{X},\vec{Y})}. \tag{68}$$

The first term on the right-hand side represents the relativistic kinetic energy of the system in the presence of the magnetic fields.

$$D_{kin}^{00} = (\beta - 1_{4\times4})c2 + c\vec{\alpha}\vec{p}, \tag{69}$$

$$D^{10} = \vec{\alpha}\vec{A}_{\vec{X}}, D^{01} = \vec{\alpha}\vec{A}_{\vec{Y}} \tag{70}$$

The second term in Equation (68) represents the potential energy, which is expressed in terms of large and small components of MOs:

$$E_{pot}^{(\vec{X},\vec{Y})} = \left\langle \varphi_i^{L(\vec{X},\vec{Y})} \middle| E_{2\times2}^{(\vec{X},\vec{Y})} \middle| \varphi_i^{L(\vec{X},\vec{Y})} \right\rangle + \left\langle \varphi_i^{S(\vec{X},\vec{Y})} \middle| E_{2\times2}^{(\vec{X},\vec{Y})} \middle| \varphi_i^{S(\vec{X},\vec{Y})} \right\rangle. \tag{71}$$

The key point in the calculation of second-order properties in 4DFT theory is to obtain the linear response of the four-component molecular orbitals due to the magnetic perturbations. For that, they are expanded in a power series:

$$\varphi_i^{(\vec{X},\vec{Y})} = \varphi_i^{(0,0)} + \varphi_i^{(1,0)_u} X_u + \varphi_i^{(0,1)_v} Y_v + \dots, \tag{72}$$

where

$$\varphi_i^{(1,0)_u} = \partial_{X_u}\left(\varphi_i^{(\vec{X},\vec{Y})}\right)\bigg|_{\vec{X},\vec{Y}=0}, \quad \varphi_i^{(0,1)_v} = \partial_{Y_v}\left(\varphi_i^{(\vec{X},\vec{Y})}\right)\bigg|_{\vec{X},\vec{Y}=0}. \tag{73}$$

To determine the derivatives of the four-spinors in Equation (73), their dependence on the magnetic fields is specified. The large component of the *i*th molecular orbital $\varphi_i^{L(\vec{X},\vec{Y})}$ is expressed as the linear combination of the basis functions $\chi_\lambda$ with the expansion coefficients $\mathbf{C}_{\lambda i}^{L(\vec{X},\vec{Y})}$ depending on two magnetic perturbations, $\vec{X}$ and $\vec{Y}$. The small components can be expressed as the linear combination of the magnetically balanced (restricted condition, in the simplest approximation) basis functions $\chi_\lambda^{S(\vec{X},\vec{Y})}$ with the field-dependent coefficients $\mathbf{C}_{\lambda i}^{S(\vec{X},\vec{Y})}$. Thus, the four-component molecular orbitals depend on the magnetic perturbations via both MO coefficients and basis functions. Due to this fact, the linear responses of MOs in Equation (73) can be expressed as follows:

$$\varphi_i^{(1,0)_u} = \mathbf{C}_{\lambda i}^{(1,0)_u}\chi_\lambda^{(0,0)} + \mathbf{C}_{\lambda i}^{(0,0)_u}\chi_\lambda^{(1,0)_u} \equiv \varphi_i^{r(1,0)_u} + \varphi_i^{m(1,0)_u}, \tag{74}$$

where $\varphi_i^{r(1,0)_u}$ and $\varphi_i^{m(1,0)_u}$ represent the regular and magnetic parts of the four-spinor. The first one depends on the magnetic field only via the MO coefficients, whereas the second part contains unperturbed MO coefficients and the dependence on the magnetic field comes via explicit field-dependent basis functions for the small component. The regular parts are expressed in the basis of the unperturbed molecular orbitals $\{\varphi_p^{(0,0)}\}_p$ through the usual linear-response expansion coefficients:

$$\varphi_i^{r(1,0)_u} = \mathbf{C}_{\lambda i}^{(1,0)_u}\chi_\lambda^{(0,0)} \equiv \beta_{pi}^{X_u}\varphi_p^{(0,0)}, \tag{75}$$

where $X$ is a primary perturbation ($\vec{X} = \vec{B}$ and $\vec{\mu}^M$), and $\beta_{pi}^{X_u}$ are the linear-response expansion coefficients of the perturbed MOs in the basis of the unperturbed MOs. The magnetic term $\varphi_i^{m(1,0)_u}$ arises as a consequence of exploiting the magnetically balanced basis set, and it does not contain any unknown coefficients.

Thus, in accordance with the standard definition of the shielding tensor as the second derivative of energy with respect to the external magnetic field and nuclear magnetic moment (Equation (5)), the main expression for the tensor $\sigma_{uv}$ can be presented as follows:

$$\sigma_{uv} = \sigma_{uv}^D + \sigma_{uv}^{P0} + \sigma_{uv}^{P1}, \tag{76}$$

where

$$\sigma_{uv}^D = \left\langle \varphi_i^{m(1,0)_u} \middle| D^{(0,1)_v^M} \middle| \varphi_i^{(0,0)} \right\rangle + \left\langle \varphi_i^{(0,0)} \middle| D^{(0,1)_v^M} \middle| \varphi_i^{m(1,0)_u} \right\rangle, \tag{77}$$

$$\sigma_{uv}^{P0} = \left(\beta_{ji}^{B_u}\right)^* \left\langle \varphi_j^{(0,0)} \middle| D^{(0,1)_v^M} \middle| \varphi_i^{(0,0)} \right\rangle + \beta_{ji}^{B_u} \left\langle \varphi_i^{(0,0)} \middle| D^{(0,1)_v^M} \middle| \varphi_j^{(0,0)} \right\rangle, \tag{78}$$

$$\sigma_{uv}^{P1} = \left(\beta_{ai}^{B_u}\right)^* \left\langle \varphi_a^{(0,0)} \middle| D^{(0,1)_v^M} \middle| \varphi_i^{(0,0)} \right\rangle + \beta_{ai}^{B_u} \left\langle \varphi_i^{(0,0)} \middle| D^{(0,1)_v^M} \middle| \varphi_a^{(0,0)} \right\rangle. \tag{79}$$

In the nonrelativistic limit $\sigma_D^{uv}$ turns into the classical diamagnetic term, $\sigma_{P1}^{uv}$ vanishes, and $\sigma_{uv}^{P1}$ becomes the standard paramagnetic contribution with summation only over unoccupied positive-energy MOs.

At the same time, the tensor $K_{uv}^{MN}$, defined by Equation (16), can also be presented in the basis of unperturbed MOs:

$$K_{uv}^{(M,N)} = K_{uv}^{D(M,N)} + K_{uv}^{P0(M,N)} + K_{uv}^{P1(M,N)}, \tag{80}$$

$$K_{uv}^{D(M,N)} = \left\langle \varphi_i^{m(1,0)_u^M} \middle| D^{(0,1)_v^N} \middle| \varphi_i^{(0,0)} \right\rangle + \left\langle \varphi_i^{(0,0)} \middle| D^{(0,1)_v^N} \middle| \varphi_i^{m(1,0)_u^M} \right\rangle, \tag{81}$$

$$K_{uv}^{P0(M,N)} = \left(\beta_{ji}^{\mu_u^M}\right)^* \left\langle \varphi_j^{(0,0)} \middle| D^{(0,1)_v^N} \middle| \varphi_i^{(0,0)} \right\rangle + \beta_{ji}^{\mu_u^M} \left\langle \varphi_i^{(0,0)} \middle| D^{(0,1)_v^N} \middle| \varphi_j^{(0,0)} \right\rangle, \tag{82}$$

$$K_{uv}^{P1(M,N)} = \left(\beta_{ai}^{\mu_u^M}\right)^* \left\langle \varphi_a^{(0,0)} \middle| D^{(0,1)_v^N} \middle| \varphi_i^{(0,0)} \right\rangle + \beta_{ai}^{\mu_u^M} \left\langle \varphi_i^{(0,0)} \middle| D^{(0,1)_v^N} \middle| \varphi_a^{(0,0)} \right\rangle. \tag{83}$$

In the non-relativistic limit, $K_{uv}^{D(M,N)}$ becomes the classical diamagnetic term, $K_{uv}^{P0(M,N)}$ vanishes and $K_{uv}^{P1(M,N)}$ becomes the sum of the standard FC, PSO, SD contributions and the FC-SD cross-terms with summation only over unoccupied positive energy MOs.

The Equations (76)–(83) are further transformed into a matrix form, by replacing the unperturbed four-spinors with their bi-spinor form, so that the final expressions contain the unperturbed MO coefficients $\mathbf{C}_{\lambda i}^{L/S(\vec{X},\vec{Y})}$, the linear-response coefficients, $\beta_{pi}^{X_u}$ and the property integrals over the atomic orbitals $\chi_\lambda$. The unperturbed MO coefficients for the large and small components are usually obtained during the self-consistent-field (SCF) procedure. The linear-response coefficients of the occupied molecular orbitals are derived from the normalization condition, and that for the unoccupied molecular orbitals are obtained within the perturbation theory.

The mDKS-RMB methodology of Komorovsky and Repisky et al. has been implemented within the ReSpect program package including the property module MAG-ReSpect [361]. At the present moment, their methodology can be regarded as one of the most efficient approaches to the calculation of the NMR parameters within the four-component DFT method.

Apart from the mDKS-RMB methodology, there is another approach for the relativistic DKS calculations of nuclear shielding tensors, which was proposed by Xiao et al., namely the orbital decomposition approach (ODA) [102,103]. The ODA also goes beyond the kinetic balance and treats the magnetic part of the response of the small components. However, in contrast to mDKS-RMB methodology, the ODA approach starts from the operator form

of the DKS equations and introduces the basis set only at the stage when the response equations are written via the basis for the large component only. This results in more complicated equations due to the occurrence of a nontrivial operator with a denominator depending on the potential and on one-electron energies. An advantage of the ODA approach is that one can use the basis set for the large component only, however, the resulting response equations are very complicated and very time consuming. The ODA method has been implemented within the BDF program package [362,363].

Cheng et al. [104] worked out the magnetically balanced gauge-including atomic orbitals MB-GIAOs, in which each magnetically balanced atomic orbital has its own local gauge origin placed on its center. The MB-GIAOs methodology has been implemented into the BDF package at the coupled-perturbed Dirac–Kohn–Sham level, and can be combined with the ODA, mDKS-RMB, and two–component external field-dependent unitary transformation (EFUT) [364] formalisms.

A simple and efficient scheme (called as sMB) for simulating the magnetic balance between the large and small components of a molecular four-component spinor in the presence of an external magnetic field as applied to the calculation of NMR shielding tensors within the four-component relativistic Kohn–Sham density functional theory including non-collinear spin magnetization and employing London atomic orbitals (LAOs) was proposed recently by Olejniczak et al. [365]. The main idea proposed in the cited article is to optimize the zeroth-order orbitals using the RKB condition and then extend the RKB coefficients by the unrestricted kinetic balance (UKB) complement for use in the subsequent calculations of NMR shieldings in combination with LAOs. The authors have implemented the presented scheme, sMB, into the DIRAC code [366]. They tested the sMB scheme in the DFT and SDFT calculations of isotropic shielding constants of hydrogen halides HX (X = F, Cl, Br, I, At) against various balance schemes, RKB, and UKB. It was found that convergence of results is slow when using the RKB scheme: in order to saturate the space of small-component functions, extensive basis sets for the large component have to be used. On the contrary, the simple scheme for magnetic balance, sMB, has proved to be a useful and computationally economic way to calculate the NMR shielding tensor, yielding results comparable with the mDKS-RMB-GIAO method.

To attain more computational benefit from the DFT methodology in the relativistic domain, a great deal of two-component approaches to the calculation of the NMR parameters have been developed. Sun et al. [107] have published a thorough review, devoted to the two-component schemes for calculating the NMR parameters, surveying in detail various types of two-component Hamiltonians, both the exact two-component (X2C) and the approximate two-component (A2C). It is generally accepted that X2C schemes imply that the eigenvalues of a given two-component Hamiltonian exactly reproduce the solutions of the four-component analogue. According to recent reviews by Autschbach [367] and Peng and Reicher [368], one can distinguish two different types of X2C schemes, namely those which split the Hamiltonian in large and small subblocks in one- and more-than-one-step procedure.

Among the most well-known X2Cs are the following schemes: normalized elimination of the small component (NESC) [369–376], the infinite-order Douglas–Kroll–Hess (DKH) [377], and the Barysz–Sadlej–Snijders (BSS) or the infinite-order two-component (IOTC) approach [378–381].

All other two-component schemes that are theoretically unequal to the original four-component problem are called as quasi-relativistic or approximate two-component schemes, A2C. These include the following most well-known approaches: zero-order regular approximation (ZORA) [101,382–384], second-order regular approximation to normalized elimination of the small component (SORA-NESC) [385], finite-order Douglas-Kroll-Hess approximations (DKH) [386–393], and the infinite-order regular approximation (IORA) [394,395].

As applied within the DFT method, among the two-component approaches mentioned above, it was ZORA which attained the most popularity in the calculations of NMR properties, especially in the solid-state NMR [396–399].

### 3.4. Polarization Propagator Methods

From a physical point of view, the most natural way to determine the spectroscopic NMR parameters is to use linear response theory. Within the linear response theory, NMR parameters are calculated from the response of an electronic system to a perturbation caused by an external electromagnetic field. This can be expressed through the polarization propagator [11,400], which is also called as the linear response function:

$$<< \hat{P}; \hat{Q} >>^r_{E+i\varepsilon} = \lim_{\varepsilon \to 0} \sum_{n \neq 0} \left[ \frac{\langle 0|\hat{P}|n\rangle \langle n|\hat{Q}|0\rangle}{E - E_n + E_0 + i\varepsilon} - \frac{\langle 0|\hat{Q}|n\rangle \langle n|\hat{P}|0\rangle}{E + E_n - E_0 + i\varepsilon} \right]. \tag{84}$$

The response function $<< \hat{P}; \hat{Q} >>^r_{E+i\varepsilon}$ embodies the first-order change in the ground state average value of the quantum operator operator $\hat{P}$ under the action of the perturbation $\hat{Q}$. Accordingly, the physical interpretation of the propagator is determined by the physical meaning of the operators $\hat{P}$ and $\hat{Q}$. In the case of NMR properties, these are one-electron NMR-active hyperfine operators. Thus, for the NMR properties, the polarization propagator describes the propagation of a disturbance in an electronic system caused by the magnetic perturbations generated by the nuclear magnetic moments or an external magnetic field.

As can be seen from expression (84), the exact excitation energies, $E = \pm(E_n - E_0)$, represent the real parts of its poles, while the transition moments $\langle 0|\hat{X}|n\rangle$, $\langle n|\hat{X}|0\rangle$ ($\hat{X} = \hat{P}, \hat{Q}$) are its residues. Obviously, knowing the exact excitation energies and transition moments would make it possible to find the exact values of the NMR parameters by evaluating the function $<< \hat{P}; \hat{Q} >>^r_{E+i\varepsilon}$ at $E = 0$ (this condition means a static case when the perturbations applied to the system do not depend on time, which is the case of NMR operators). However, the polarization propagator approach does not resort to a direct evaluation of $<< \hat{P}; \hat{Q} >>^r_{0+i\varepsilon}$. In general, polarization propagator is found from the equation of motion (EOM). The iterative solution of EOM using the superoperator formalism [401] and the technique of Inner Projection (IP) [402] give rise to the expression for the polarization propagator in a compact form [11]:

$$<< \hat{P}; \hat{Q} >>^r_E = (\hat{P}^+|\hat{\mathbf{h}}) (\hat{\tilde{\mathbf{h}}}|E\hat{I} - \hat{H}_0|\hat{\mathbf{h}})^{-1} (\hat{\tilde{\mathbf{h}}}|\hat{Q}). \tag{85}$$

In Equation (85) the $(\hat{P}|\hat{Q}) = \langle 0|[\hat{P}^+, \hat{Q}]|0\rangle$ designates the superoperator product, and it is accepted that $\hat{H}_0 \hat{Q} \equiv [\hat{H}_0, \hat{Q}]$. The complete excitation operator manifold $\hat{\mathbf{h}}$ consists of an infinite number of sets of elementary (not reduced by spin) operators of single, double, etc. excitations and deexcitations:

$$\hat{\mathbf{h}} = \left\{ \hat{\mathbf{h}}_2, \hat{\mathbf{h}}_4, \hat{\mathbf{h}}_6, \ldots \right\}, \tag{86}$$

where

$$\hat{\mathbf{h}}_2 = \{\hat{\mathbf{q}}^+, \hat{\mathbf{q}}\} = \{a_m^+ a_a, a_a^+ a_m\}, \ \hat{\mathbf{h}}_4 = \{\hat{\mathbf{q}}^+\hat{\mathbf{q}}^+, \hat{\mathbf{q}}\hat{\mathbf{q}}\} = \{a_m^+ a_n^+ a_a a_b, a_b^+ a_a^+ a_n a_m\}, \ \ldots \tag{87}$$

In the polarization propagator formalism, the $<< \hat{P}; \hat{Q} >>^r_E$ is expressed as a product of matrices [11]:

$$<< \hat{P}; \hat{Q} >>^r_E = (\mathbf{P}_a^+, \mathbf{P}_b^+, \ldots) \begin{pmatrix} \mathbf{M}_{aa} & \mathbf{M}_{ab} & \ldots \\ \mathbf{M}_{ba} & \mathbf{M}_{bb} & \ldots \\ \ldots & \ldots & \ldots \end{pmatrix}^{-1} \begin{pmatrix} \mathbf{Q}_a \\ \mathbf{Q}_b \\ \ldots \end{pmatrix}, \tag{88}$$

with

$$\mathbf{P}_a = (\hat{P}|\hat{\mathbf{h}}_a) \text{ and } \mathbf{M}_{ab} = (\hat{\mathbf{h}}_a|E\hat{I} - \hat{H}_0|\hat{\tilde{\mathbf{h}}}_b), \tag{89}$$

Eventually, finding the $<< \hat{P}; \hat{Q} >>^r_E$ is equivalent to the determination of the resolvent matrix $\mathbf{M}^{-1}$ using the spectral theorem, which assumes solving the generalized

eigenvalue problem for the matrix **M**. In general, the calculation of NMR properties within the framework of the polarization propagator approach can briefly be described as a sequence of several steps: (a) solving the unperturbed Hartree–Fock problem in order to find the molecular coefficients, orbital energies, and the ground state energy; (b) calculating the elements of the effective matrices; (c) solving the generalized eigenvalue problem for the matrix **M**; (d) evaluating the polarization propagator through Equation (88).

In order to obtain useable approximations, the operator manifold $\hat{\mathbf{h}}$ is truncated to a subset of operators, $\{\hat{\mathbf{h}}_2\}$ or $\{\hat{\mathbf{h}}_2, \hat{\mathbf{h}}_4\}$ or $\{\hat{\mathbf{h}}_2, \hat{\mathbf{h}}_4, \hat{\mathbf{h}}_6\}$, ... and so on. Thus, narrowing down the manifold $\hat{\mathbf{h}}$ to a set of one-electron excitations/deexcitations, $\{\hat{\mathbf{h}}_2\}$, leads to the random phase approximation (RPA) [113]. This gives results identical to those of the CPHF method (both methods take into account the effects of electron correlation in the first order of fluctuation potential, which is considered to be an "uncorrelated" level). The computational scaling of the RPA equations is of the fourth power of the basis set size ($N^4$).

The next approximation is the method of the second-order polarization propagator approach (SOPPA) [403–409]. In the SOPPA method, the manifold is restricted to the subset $\hat{\mathbf{h}} = \{\hat{\mathbf{h}}_2, \hat{\mathbf{h}}_4\}$. The SOPPA method demonstrates a noticeable improvement in the accuracy of the calculations of both singlet and triplet NMR properties. One of the advantages of the SOPPA method is that the projection manifold and the reference state are uniquely determined, so that the question concerning the choice of the configuration space does not exist. In addition, the SOPPA method assumes a simple systematic extension to higher orders, implying a sequential improvement in the description of the electron correlation effects.

There are two modifications of the SOPPA method—the SOPPA(CC2) [410] and SOPPA(CCSD) [411]. They were created to improve the accuracy of the calculation of the SSCCs relative to the original SOPPA method. Within the framework of these modifications, all coefficients from the Møller–Plesset perturbation theory, which are used to calculate the matrix **M** in the SOPPA approach, are replaced with the coupled cluster amplitudes from the corresponding CC scheme, which makes it imperative to solve the system of cluster equations within the CC-modified SOPPA models. This complicates calculations, and, in the case of the SOPPA(CCSD) scheme, this even increases the scalability of the method by an order of magnitude in terms of the number of basis set functions $N$, i.e., from $N^5$ for SOPPA to $N^6$ for SOPPA(CCSD). The SOPPA method and its modifications have become very popular approaches in the calculations of SSCCs of various types [406,407,412–415].

The RPA model, which corresponds to the coupled Hartree–Fock calculation in the static regime, is feasible for large molecules (it scales as $N^4$), and that is quite an advantage as compared to SOPPA methods and its modifications. However, it neglects the electron correlation effects, which is problematic, especially for the triplet NMR properties resulting in triplet instabilities [416,417]. In this way, there is a need for the correlated wavefunction-based approaches to the calculation of NMR molecular properties, which would have less computational demands as compared to SOPPA or its modifications and yet would be more reliable than the RPA method. Recent developments of Sauer's group pointed at resolving the issue. In particular, the higher RPA (HRPA) model [418–420] could have become one of the plausible extensions of the RPA method. It includes the second-order correction to the matrix **M** of SOPPA, while the contributions of the double excitations found in the SOPPA model are omitted. This turned out to be an obstacle, as the importance of double excitations has already been acknowledged by that moment and methods such as CIS(D) [158,421,422], RPA(D) [420,423], and HRPA(D) [420] have been developed for the excitation energy calculations with promising results [420–427]. Thus, Schnack–Petersen et al. [428] presented two new modifications of the RPA method for the calculations of NMR indirect nuclear spin–spin coupling constants recently, namely the RPA(D) and HRPA(D). In these models, the double excitation contribution is treated noniteratively as a correction to the results of RPA or HRPA levels. The idea of the D-extended approaches consists in the solution of the generalized eigenvalue problem using the pseudoperturbation theory. That means expansion of the matrix **M** and the vectors **P** and **Q** in a kind of perturbation series, based on the deficiency in relation to the corresponding SOPPA matrices and vectors.

Thus, the SOPPA matrices and vectors can be partitioned into the contributions of different (pseudo-)order terms, where the zeroth-order matrices are chosen as those corresponding to a smaller problem, namely RPA or HRPA approximations.

In terms of the number of occupied orbitals, O, the number of basis functions, $N$, and the number of virtual orbitals, $V = N - O$, the RPA(D) and HRPA(D) both have the most demanding term with scaling factor of $N^4O^1$ due to the fact that both methods require transformation with the SOPPA matrices. In $O$-$N$-$V$ terms, the RPA equations scale as $N^4$, while the leading terms in the SOPPA approach scale as a partial two-electron integrals transformation, $N^4O^1$. According to the authors' statement, within the RPA(D) model, only the RPA equations are solved iteratively and the $N^4O$ contribution are calculated once using the converged RPA vectors. Thus, the savings of RPA(D) relative to SOPPA for a large system is proportional to the number of iterations required to converge the SOPPA equations. The HRPA(D) method, on the other hand, requires the same amount of $N^4O$ terms as a SOPPA iteration, though the calculation of some mostly demanding terms can be avoided. The savings of HRPA in terms of the computational cost of an iteration is thus quite small in the typical case of $V >> O$. The test calculations of different types of SSCCs in a number of small inorganic molecules performed in [428] showed that both the RPA(D) and HRPA(D) models yield the results of good accuracy compared to the SOPPA model with noticeable time savings.

The performance of RPA(D) and HRPA(D) models has extensively been examined on the calculation of carbon–carbon spin–spin coupling constants for 39 saturated carbocycles in the next work of Sauer's group [429]. It was found that the HRPA(D) method provides accuracy similar to that of the SOPPA method with 65% reduction in computation time at best (this varies substantially with the investigated molecule, namely in the range 15–65%,), while the RPA(D) demonstrates an essential improvement over the RPA method with approximately 60–85% time saving as compared to the SOPPA method. The RPA(D) model yields the results with ever so slightly lower accuracy than SOPPA. However, the RPA(D) model was found to suffer from the triplet instability problems, which was not observed for the HRPA(D) model. In the author's opinion, the HRPA(D) may prove beneficial in the predictions of the coupling constants of large molecules, while the RPA(D) method, despite less computation times, may not be thought of as a reliable alternative to either SOPPA or HRPA(D) methods.

Other useful models for a deeper understanding of the electronic origin and transmission mechanisms of the spectroscopic NMR parameters, are based on the inner projections within the polarization propagator (IPPP) and contributions from the localized orbitals within the polarization propagator approach (CLOPPA) methods [11,22,430,431]. Originally implemented at the semiempirical levels [432–436] into the RPA method, they are nowadays embedded into the RPA [437] and SOPPA models [438] at the ab initio level. The IPPP and CLOPPA approaches were developed mainly to analyze the NMR SSCCs in terms of the "local" contributions. In particular, CLOPPA is based on the decomposition of SSCC as a summation of contributions from individual coupling pathways involving two virtual excitations $i{\rightarrow}a$ and $j{\rightarrow}b$ with $i$, $j$ ($a$, $b$) occupied (vacant) localized MOs (LMOs) that belong to the local fragment of interest ($L$):

$$J_{MN} = \sum_{ia,jb} J^L_{MN;ia,jb}. \tag{90}$$

This allows one to extract some crucial information on the transmission mechanisms involved in the propagation of a given specific magnetic perturbation, the FC, SD, or PSO.

The implementation of the gauge invariant atomic orbitals (GIAO) approach within the RPA approach (which is equivalent to the CPHF level in the static limit) for the chemical shifts calculations was first carried out by Ditchfield [60], followed by others [30,33,35,36]. The RPA calculations of NMR chemical shifts are rather scarce nowadays. The SOPPA calculations of the NMR shielding constants are practically absent. The problem lies in the inefficiency of solving the gauge origin problem within the SOPPA method. To solve the

gauge origin problem within the SOPPA method, the continuous transformation of origin of the current density whereby the diamagnetic contribution to the current density is set to zero (CTOCD-DZ) [439] was proposed in the application to SOPPA and SOPPA(CCSD) formalisms [440–442]. The CTOCD-DZ method is based on the ideas of the Keith–Bader numerical approach of a continuous set of gauge transformations (CSGT) [443–445], in which the current density induced by an external magnetic field is calculated at each point of space, provided that the origin of the coordinate system is placed at the point under consideration. Then, the magnetic properties are calculated using the well-known relations of classical electrodynamics in the form of three-dimensional integrals involving the current density. In fact, the CTOCD-DZ method represents a transformation of the Keith–Bader numerical approach to an analytical form. The SOPPA method applied in conjunction with the CTOCD-DZ formalism to the calculation of shielding constants is prone to demonstrate very slow convergence over the basis sets. This means that in order to obtain an adequate result within the CTOCD-DZ-SOPPA method, very large basis sets are required.

A new method of algebraic diagrammatic construction (ADC) [446,447] based on the formalism of polarization propagator has been proposed for the calculations of SSCCs recently. In the ADC approach, the polarization propagator is represented in the non-diagonal form:

$$\mathbf{\Pi}(\omega) = \mathbf{f}^+ (\omega - \mathbf{M})^{-1} \mathbf{f}, \tag{91}$$

which is a generalization of the familiar spectral representation of $\mathbf{\Pi}(\omega)$:

$$\mathbf{\Pi}(\omega) = \mathbf{x}^+ (\omega - \mathbf{\Omega})^{-1} \mathbf{x}, \tag{92}$$

where $\mathbf{\Omega}$ is the diagonal matrix of excitation energies:

$$\Omega_n = E_n - E_0, \tag{93}$$

and $\mathbf{x}$ is the matrix of "spectroscopic amplitudes":

$$x_{rs}^n = \langle \Psi_n | c_r^+ c_s | \Psi_0 \rangle. \tag{94}$$

In the ADC approach, the quantities $\mathbf{M}$ and $\mathbf{f}$ are found by comparison of Equation (91) with the Feynman–Goldstone diagrammatic series for $\mathbf{\Pi}(\omega)$ through a given order $n$ of the perturbation theory. For this purpose, the matrices $\mathbf{M}$ and $\mathbf{f}$ are expanded in the power series in the fluctuation potential:

$$\mathbf{M} = \mathbf{M}^{(0)} + \mathbf{M}^{(1)} + \mathbf{M}^{(2)} + \dots , \tag{95}$$

$$\mathbf{f} = \mathbf{f}^{(0)} + \mathbf{f}^{(1)} + \mathbf{f}^{(2)} + \dots \tag{96}$$

Alternatively, the ADC schemes are introduced using a concept of specifically constructed ADC basis states, within the so-called intermediate states representation (ISR) [448] and is expressed in terms of effective values decomposed in a series by the fluctuation potential. A second-order scheme, which takes into account single and double excitations, ADC(2) [447], has been developed and implemented for SSCCs. The scalability of ADC(2) scheme is the same as that of SOPPA method, i.e., $N^5$.

The ADC approach possesses transparent computational procedure operating with Hermitian matrix quantities defined with respect to physical excitations. It is size-consistent and easily extendable to higher orders via the hierarchy of available ADC approximation schemes. The ADC(2) method was tested on the series of small molecules HF, $N_2$, CO, $H_2O$, HCN, $NH_3$, $CH_4$, $C_2H_2$, $PH_3$, $SiH_4$, $CH_3F$, and $C_2H_4$. The calculated indirect nuclear spin–spin coupling constants occurred to be in a good agreement with the experimental data, which means that this method may happen to be promising for applications to larger molecules.

The extension of the RPA method to the four-component relativistic domain was first discussed by Pyykkö [449] who presented the coupled perturbated Dirac–Hartree–Fock (CP-DHF) theory in application to the calculations of the SSCCs, which is equivalent to the four-component RPA (4RPA) method. Then, it was developed further in detail by Aucar and Oddershede [113], whose work was followed by a plethora of calculations of SSCCs within the framework of 4RPA formalism [112,450–452]. For nuclear shielding constants, Iliaš and co-authors developed the theory of relativistic localized atomic orbitals, LAO (London Atomic Orbitals), and implemented it into the CP-DHF (4RPA) approach, demonstrating the efficiency of the presented methodology on the example of calculating the $\sigma(^1\text{H})$ and $\sigma(^{127}\text{I})$ in the HI molecule [130]. It should be noted that this was preceded by the developments of Quiney et al. [453–455], who showed the applicability of the London orbitals to the CP-DHF theory. Further, the developments of the 4RPA method were concentrated on the correct inclusion of the magnetic balance condition in the formalism of relativistic localized atomic orbitals, which is necessary for all relativistic four-component methods for calculating NMR shielding constants. Cheng et al. [104] presented the theory of the so-called magnetically balanced gauge-including atomic orbitals theory (MB-GIAOs), in which each "magnetically balanced" atomic orbital has its own local origin located in its center. Cheng and co-authors refuted an accepted statement that the magnetic balance (MB) condition is incompatible with the four-component relativistic polarization propagator theory 4c-PPT [456]. They believed that the statement is rather counterintuitive as its random phase approximation is, in the static limit, fully equivalent to the CP-DHF theory, and, in its turn, the CP-DHF theory is known to be compatible with the formalism of magnetically balanced localized orbitals. Thus, they showed that the MB-GIAO scheme can be combined with any four-component electronic structure calculation method, in particular, with the 4PPT methods. However, the first demonstration of the efficiency of the theory of relativistic MB-GIAO was carried out on the example of the shielding constants calculated within the framework of the Dirac–Kohn–Sham theory (DKS) or four-component DFT (4DFT) method [457,458]. The calculations of nuclear shielding constants within the 4RPA formalism were presented in a number of papers [112,452,459], however, this method has not received as much popularity as the Dirac–Kohn–Sham approach.

A real breakthrough in the computational NMR was made by the scientific group of Sauer [127] very recently, who presented the second-order-polarization-propagator-approximation (SOPPA) method within the relativistic framework. The equations for relativistic SOPPA were deduced in their most general form, i.e., in a non-canonical spin-orbital basis, which can be reduced to the canonical case. The obtained equations are one-index transformed, giving more compact expressions that correspond to those already available for the four-component RPA. Thereby, a possible scheme has been outlined for the implementation of the presented equations in a program that already contains an RPA code that allows for the evaluation of the non-canonical RPA equations in a spin-orbital basis. This will allow the NMR spectroscopic properties of molecules containing heavy elements to be calculated within the relativistic SOPPA method.

### 3.5. Methods Based on the Many-Body Perturbation Theory

Perturbation theory has been used since the early days of quantum chemistry to obtain electron correlation-corrected descriptions of the electronic structure and properties of molecules [460]. The calculation of NMR chemical shifts at the simplest correlated level of second-order many-body perturbation theory, MBPT(2), or the second-order Møller-Plesset, MP2 theory [461,462], is one of the most attractive methods, since it offers a good compromise between the accuracy and computational costs (scaling as $N^5$, with $N$ being the number of basic functions). The correlation energy in MP2 is determined by the following equation:

$$E(\text{MP2}) = \frac{1}{4}\sum_{ij}\sum_{ab} t_{ij}^{ab*}\langle ab\|ij\rangle, \tag{97}$$

where $\langle pq \| rs \rangle$ are the antisymmetric two-electron spin-orbit integrals and $t_{ij}^{ab}$ are the amplitudes of the two-particle excitation:

$$t_{ij}^{ab} = \langle ab \| ij \rangle / (f_{ii} + f_{jj} - f_{aa} - f_{bb}). \tag{98}$$

Molecular orbitals $|p\rangle$ are the eigenfunctions of the Fock operator with corresponding eigenvalues $f_{pp}$. Indexes $i, j, k, l, \ldots$ refer to occupied spin orbitals, while $a, b, c, d$ designate the vacant (unoccupied) spin-orbitals. The general indices $p, q, r, s, \ldots$ are the spin-orbitals that can be either occupied or vacant. The MP2 expressions for the shielding tensor are obtained by differentiating the Equation (97) with respect to the nuclear magnetic moment, and then with respect to the magnetic field. The resulting expression has the form [463]:

$$\sigma_{N;ji}(\text{MP2}) = \sum_{\mu\nu} D_{\mu\nu} \frac{\partial^2 h_{\mu\nu}}{\partial B_i \partial \mu_{N;j}} + \sum_{\mu\nu} \frac{\partial D_{\mu\nu}}{\partial B_i} \frac{\partial h_{\mu\nu}}{\partial \mu_{N;j}}, \tag{99}$$

where $D_{\mu\nu}$ represents the elements of a one-particle density matrix at the MP2 level in the basis of atomic orbitals, while $h_{\mu\nu}$ are the elements of the one-electron Hamiltonian in the basis of atomic orbitals. Thus, the calculation of $\sigma_{N;ji}(\text{MP2})$ requires the knowledge of the perturbed integrals (which can usually be calculated in a simple way using the standard methods), as well as the perturbed and unperturbed MP2 density matrix, $\frac{\partial D_{\mu\nu}}{\partial B_i}$ and $D_{\mu\nu}$.

The pioneering works on the implementation of the GIAO formalism into the MP2 theory have been commenced by Pulay and his colleagues [65,464]. Initially, the problem consisted in the calculation of the perturbed two-electron integrals in the GIAO formalism, however, Pulay et al. have shown that such integrals can be calculated in a fairly simple way using the similarity of the perturbed GIAO integrals with derivatives of ordinary two-electron integrals. In their work, it was also shown that at the initial stage of calculations, which implies the framework of the GIAO-CPHF method, the storage of the perturbed GIAO integrals is not required. It has also been shown that perturbed integrals are calculated very quickly, which is an important feature for any computational method. The method proposed by Pulay was applicable only in the case of non-contracted Gaussian basis sets. To date, the perturbed GIAO-molecular integrals are easily calculated within the framework of standard methods, such as the Dupuis' polynomial approach [465], the McMurchie–Davidson scheme [466], or within the framework of recursive Obara–Saika methods [467].

Gauss presented the first rigorous scheme for calculating the NMR chemical shifts at the MP2 (or MBPT(2)) level based on the GIAO ansatz [468,469]. Subsequent calculations using the GIAO-MP2 approach demonstrated the importance of electron-correlation effects in the NMR chemical shifts calculations. A number of problems concerning the interpretation of experimental spectra of boranes [470], carboranes [470,471], and carbocations [472–475] were resolved using the MP2 approach. The GIAO-MP2 approach has been subsequently extended to the third- and fourth orders of perturbation theory [476], based on the MP3 [477,478] and MP4 [478–481] theories. All these developments were made using the analytic second-derivative techniques. The GIAO-MP3 and GIAO-MP4 methods [476] are substantially more costly than the GIAO-MP2 approach ($N^6$ and $N^7$, respectively, against $N^5$), but if applied with the approximation of the resolution-of-the-identity (RI) [241,482], the calculations with these models could become more feasible. In practice, Gauss suggested the GIAO-SDQ-MBPT(4) approximation for the GIAO-MP4 approach [476], specifically for the cases when triple correlation effects play a minor role in a given system. The computational cost of the GIAO-SDQ-MBPT(4) approach scales as $N^6$.

It should be noted that the MBPT($n$) methods are not recommended for the calculation of the SSCCs for the same reason as the CCSD(T) method, which consists in resorting to the relaxed Hartree–Fock orbitals.

The relativistic MBPT theory is presented elsewhere [483–486] and most recently in [487]. In the relativistic representation, the original Hamiltonian is generalized as follows [483]:

$$H = H_0 + V_C + B, \tag{100}$$

$$H_0 = \sum_{i=1}^{N} (\vec{\alpha}_i \cdot \vec{p}_i + \beta_i m) - \sum_{i=1}^{N} \frac{Z\alpha}{r_i}, \tag{101}$$

$$V_C = \frac{1}{2} \sum_{i \neq j} \frac{\alpha}{\left| \vec{r}_i - \vec{r}_j \right|}, \tag{102}$$

$$B = -\frac{\alpha}{2} \sum_{i \neq j} \frac{\vec{\alpha}_i \cdot \vec{\alpha}_j + \vec{\alpha}_i \cdot \hat{r}_{ij} \vec{\alpha}_j \cdot \hat{r}_{ij}}{\left| \vec{r}_i - \vec{r}_j \right|}. \tag{103}$$

However, this Hamiltonian is known to result in degeneracy collapse or the Brown–Ravenhall disease [488,489], when an intermediate state with zero excitation energy can appear. The standard way to avoid that is to restrict the excitations to positive-energy orbitals by means of projection operators, suggested by Sucher [490–492]. This approach is usually referred to as the no-(virtual)-pair approximation (NVPA).

The quasirelativistic second-order Møller–Plesset perturbation theory for the magnetic shielding constants was proposed by Fukuda [99]. In that work, the quasirelativistic generalized unrestricted Hartree–Fock method for the magnetic shielding constants [97,98] has been extended to include the electron correlation effects at the MP2 level of theory. The finite-perturbation method was applied to calculate the magnetic shielding constant at the quasirelativistic MP2 level of theory. The method was applied to the calculation of the NMR shielding constants and chemical shifts of $^{125}$Te nucleus in various tellurium compounds. It was shown that the calculated magnetic shielding constants and NMR chemical shifts are well reproduced by the relativistic MP2 theory as compared to the experimental values.

For SSCCs, the development of the relativistic MBPT theory makes no sense for the same reason as in the nonrelativistic case.

## 4. Computational Factors Influencing the Accuracy of NMR Spectrum Modeling

### 4.1. Specialized Basis Sets

One of the most important aspects of any quantum chemical calculation of NMR parameters is the choice of an atomic basis set. The basis set should be flexible enough to fully describe orbitals in an important region in terms of the distance to the nuclei in the case of a specific molecular property. However, the vast majority of commonly used basis sets have been developed by minimizing the parameters of the basis set with respect to atomic or molecular energies. Therefore, the calculation of molecular properties that directly depend on energy, such as dissociation energy, equilibrium geometry parameters (the first derivatives of energy in nuclear coordinates), and harmonic oscillation frequencies (the second derivatives of energy in nuclear coordinates), are described within the framework of energy-optimized basis sets very well. However, other molecular properties that strongly require the exact description of orbitals in specific areas, which are not so important for energy, require either very large standard energy-optimized basis sets or specialized basis sets, optimized for these certain molecular properties. Nuclear shielding constants and spin–spin coupling constants are prominent representatives of such molecular properties that directly depend on the quality of the description of orbitals in specific areas.

Despite the inefficiency of energy-optimized basis sets for calculating NMR properties, they are usually used on atoms that are of no interest in a specific NMR calculation. This is an example of the so-called local dense basis set (LDBS) scheme [414,493,494]. Its original meaning consisted in using large high-quality basis sets on specific atoms, which are important in the calculation, and significantly smaller basis sets on other parts of the molecules. This, to a certain extent, reduces the calculation time, and, with the correct configuration of LDBS, it is possible to achieve an accuracy of the result that is practically

the same as in the case when one type of a large high-quality basis set is set on all atoms of the molecule. With the advent of specialized property-oriented basis sets, the original meaning of the LDBS concept has undergone some changes and become more general. In that way, if a specialized basis set is installed on the atoms of interest, and the rest of the molecule is described with the non-specialized basis sets, then such a scheme is also regarded as LDBS.

### 4.1.1. Specialized Basis Sets for Calculating Spin–Spin Coupling Constants

The so-called *J*-oriented basis sets reflect the features of the SSCCs determined by their simplest nonrelativistic formulation by Ramsey [51] as the sum of four different contributions: Fermi-contact (FC), spin-dipole (SD), paramagnetic spin-orbital (PSO), and diamagnetic spin-orbital (DSO). The Fermi-contact contribution depends on the electron density on the nuclei, and its correct description requires tight *s*-type functions expressed with large exponents. The PSO, DSO, and SD contributions include matrix elements that have maximum values in the inner region of the wave function, which description requires tight functions with nonzero angular momentum, *p*-, *d*-, etc. [265]. Specifically, for the PSO contribution, it is known that tight functions of *p*-type are important. In contrast to the PSO contribution, the SD contribution is sensitive to the addition of *p*, *d*, and *f* functions [265]. On the other hand, the FC, PSO, and SD contributions are the response properties determined by the summation over excited states. The representation of the excited states in the response part, on the other hand, is expected to be sensitive to the quality of the wave function in the region far from the nucleus. Thus, they also depend on both tight and diffuse exponents. It is typical that the DSO term is vanishingly small for the vast majority of SSCCs. Therefore, studies on its dependence on basis sets have not been systematically conducted. However, based on the general considerations, it can be assumed that since the DSO contribution is calculated as the mathematical expectation value of the DSO operator for the undisturbed wave function of the ground state, then the accuracy of its description should be determined by the quality of the description of the ground state, and this requires the same optimal exponents as for the total energy of the ground state. Thus, it is plausible that the usual energy-optimized basis sets will be effective for the DSO contribution as well. In general, the situation with the dependence of the full SSCCs on various types of functions is quite complicated, because it is determined by a superposition of dependencies of four contributions that are different in physical nature, therefore, the development of new *J*-oriented basis sets is a very challenging task.

At the moment, there are a limited number of ways to create *J*-oriented basis sets. The simplest way to obtain a *J*-oriented basis set is to consistently expand the angular spaces of standard, energy-optimized basis sets with additional functions until the convergence of total SSCC or some of its dominant contributions is achieved. Within this method, the exponents of additional functions are usually calculated in the form of subsequent elements of a geometric progression $\zeta_i = \alpha \beta^i$, where $\alpha$ is the last exponent in the original basis set ($\zeta_n$), and $\beta$ is the ratio of the two last exponents $\beta = \zeta_n / \zeta_{n-1}$. Such a way of building additional exponential sets leads to the even-tempered functional series, for which the ratio between two neighboring exponents is a constant: $\zeta_i / \zeta_{i-1} = \beta = $ const. In most cases, dealing with the design of the nonrelativistic *J*-oriented basis sets, the energy-optimized correlation consistent Dunning's basis sets [266,495–498] are usually taken as starting basis sets, though, there are several examples of using medium-size polarized Sadley's basis sets MSP [499–502] or small Huzinaga basis sets [62,503], although the latter two are much less popular.

The first investigations of the influence of the even-tempered expansion of the basis sets on the SSCCs were carried out by Enevoldsen et al. and Helgaker et al. Enevoldsen et al. [406] investigated the influence of expansion of the correlation-consistent basis sets of Dunning aug-cc-pVXZ (X = D, T, Q) on the example of SSCCs $^1J(^{13}C, ^1H)$ and $^2J(^1H, ^1H)$ in the $CH_4$ molecule. It was proposed to use fully uncontracted aug-cc-pVTZ basis set, augmented by four additional tight *s*-type functions, minus one *f*-function, on the carbon and hydrogen

atoms. Helgaker et al. [200] analyzed the basis set dependence of $^1J(H,F)$ and $^1J(O,H)$ in two simple molecules, HF and $H_2O$, respectively, using the complete active space self-consistent field (CASSCF) method with a large active space. The Dunning's basis sets cc-pVXZ (X = D, T, Q, 5, 6) were decontracted completely in the *s*-space, resulting in cc-pVXZ-su0 sets, then a sequence of *n* tight *s* functions with the exponents forming a geometric progression was added, resulting in a series of basis sets cc-pVXZ-su*n*. His group also conducted the investigation [302] on how the expansion of the *s*-space of Huzinaga's basis sets, HIII [503], affects the SSCCs with leading FC contribution calculated at the DFT-B3LYP level. Overall, it was shown that the sequential saturation of the *s* shell provides a gradual improvement of the description of the FC contribution to the coupling constants.

This work was systematically continued by Provasi, Sauer, and their colleagues, who presented the *J*-oriented basis sets named as aug-cc-pVTZ-J for various elements: H, C-O [406,416], B, Al [406,504], F [406,505], Si [412,440,504], P, Cl [504], S [416], Sc-Zn [506], Ga, Ge, As, Se, and Br [507]. Rusakov and Rusakova have also presented *J*-oriented basis sets acvXz-J (X = 2, 3, 4) for Se, Te [508] and Sn [509], designed on the basis of even-tempered approach, starting from relativistic energy-optimized Dyall's basis sets dyall.acvXz (X = 2, 3, 4) [510,511].

Another way of expanding functional sets was presented by Kjær and Sauer [512], who introduced a recurrent formula for generating an exponential sequence in such a way that the ratio between two adjacent exponents increases monotonically, in contrast to the constant ratio of exponents in the case of even-tempered functional series. Kjaer and Sauer used the formula:

$$\zeta_{i-1} = \frac{(\zeta_i/\zeta_{i+1})^2}{\zeta_{i+1}/\zeta_{i+1}}\zeta_i, \tag{104}$$

in order to expand the *s*-angular space of the uncontracted Pople's basis sets 6-31G [513] and 6-311G [514] for elements H, C, N, and O. The resulting contracted basis sets were called 6-31G-J and 6-311G-J. These contain twice the number of contracted *s*-type functions but the same number of contracted *p*-type functions as the original Pople's basis sets. The obtained sets were then augmented with standard diffuse and polarization functions, resulting in basis sets named 6-31+G*-J and 6-311++G**-J. These *J*-oriented basis sets are purposed for the fast and efficient calculations of SSCCs involving $^1H$, $^{13}C$, $^{15}N$, and $^{17}O$ nuclei with leading Fermi-contact contribution within the framework of nonrelativistic DFT approach.

Another way to generate *J*-oriented basis sets involves the optimization process. In particular, Jensen et al. proposed the (aug)pcJ-*n* (*n* = 0–4) [265,515,516] basis sets, which are suitable for the calculation of the spin–spin coupling constants involving 1–3 row nuclei (H-Ar) with the density functional methods, applicable to quite large systems at a favorable computational cost. Jensen's basis sets were developed on the grounds of previously proposed series of energy-optimized polarization-consistent basis sets (aug)pc-*n* (*n* = 0–4) [517–521] by augmenting these with tight *s*, *p*, *d*, and *f* functions and determining the additional optimum exponents with a variational procedure for the sum of the absolute values of all four contributions. It was in Jensen's work where tight *s* and *p* functions were shown to be required for the correct description of the FC and PSO terms, respectively, while tight *p*, *d*, and *f* functions were shown to be necessary to converge the SD contribution.

Another series of *J*-oriented basis sets for H, He, and B-Ne, called ccJ-pVXZ (X = D, T, Q, 5) [264] has been developed by Benedikt et al. for high-quality wave function calculations such as those resorting to the coupled cluster methods. These were made by extension of the uncontracted Dunning's basis sets, cc-pVXZ(uc), with the high-exponent functions, followed by the optimization of the added exponents with a variational procedure for the sum of the absolute values of all four contributions.

Another interesting approach is based on the completeness function [522]. This approach automatically generates a universal, element-independent exponential set spanning the desired range with a completeness profile as close to unity as wanted with as few functions as possible. For magnetic properties, such as SSCCs, this scheme was introduced by Manninen and Vaara [523], who suggested the optimization procedure for a system-

atical approach to the complete basis set (CBS) limit using the completeness profile. This concept was applied to several molecular properties by Lehtola and co-authors in a limited series of works [524–526]. The completeness-based approach is rarely used and is not fully investigated in the problem of generating compact *J*-oriented basis sets.

A completely different approach of generating property-oriented basis sets, named the property-energy consistent (PEC) approach, was presented by Rusakov and Rusakova [527] very recently. The PEC method is based on the consistent optimization of the exponents using the Monte Carlo (MC) simulations [528–530] with respect to the property under consideration and total molecular energy. The approach was introduced on the example of generation of pecJ-*n* (*n* = 1, 2) basis sets for the FC dominating SSCCs involving the nuclei of the most popular $\frac{1}{2}$ spin NMR isotopes of 1–2 row elements: $^1$H, $^{13}$C, $^{15}$N, and $^{19}$F. It was mentioned that the PEC method is aimed at the generation of very efficient small property-oriented basis sets, which provide more accurate results as compared to the other property-oriented basis sets of similar sizes, while, for the larger basis sets, the accuracy of the results is expected to be comparable to that provided by the other property-oriented commensurate basis sets. The first calculations of SSCCs involving the nuclei $^1$H, $^{13}$C, $^{15}$N, and $^{19}$F in a wide row of small molecules, performed with the pecJ-*n* (*n* = 1, 2), ccJ-pVXZ (X = D, T) and pcJ-*n* (*n* = 1, 2) basis sets at the CCSD level of theory with taking into account solvent and vibrational corrections, confirmed the above-mentioned statement. In particular, the accuracy of $^1J$(C,C) SSCCs, calculated with the pecJ-1 and pec-2 basis sets was characterized by the MAPE of 3.1% and 2.6%, respectively, against the experiment. At the same time, the MAPEs of the theoretical corresponding data, obtained with the ccJ-pVDZ and pcJ-1 basis sets were found to be approximately 9.8% and 10.6%, and for the ccJ-pVTZ and pcJ-2 basis sets, these figures occurred to be 3.7% and 2.3%, correspondingly.

4.1.2. Specialized Basic Sets for Calculating NMR Chemical Shifts

For the nuclear shielding constants/chemical shifts, the choice of a basis set is very much determined by the method used to solve the gauge-origin problem. When using different methods for solving the gauge-origin problem (IGLO, LORG, GIAO, CTOCD), the convergence of NMR shielding constants, observed within different families of basis sets, also have to be different. From this side, the issue has not been studied systematically, but rather, more frequent are the studies on the convergence of shielding constants calculated using certain families of basis sets within a particular computational protocol [62,65,71,469,531,532]. For instance, Helgaker et al. [7] presented a systematical study on the convergence of nuclear shielding constants in nine simple molecules $CH_4$, $NH_3$, $H_2O$, $SiH_4$, $PH_3$, $H_2S$, $CO_2$, $C_2H_4$, and $C_2H_2$ calculated at the GIAO-CPHF level with five different families of basis sets, namely, the Pople's basis sets 6-31G, 6-311G with and without polarization and diffuse functions [513,514,533–538], Ahlrich basis sets (Karlsruhe-XZP, X = D, T, Q) [268,539], basis sets of Schindler and Kutzelnigg (IGLO II, III, IV) [61,62], Widmark basis sets (ANO(Lund)) [540–542], and correlation-consistent basis sets of Dunning, (aug)cc-pVXZ (X = D, T, Q) [266,495–498] against the GIAO-CPHF CBS limit. The fastest convergence was demonstrated within the family of IGLO basis sets, with the mean relative error less than 3% for the two largest representatives of this family, IGLO III, IV. However, they are not apt for the medium and large molecules due to their large sizes. The important conclusion made by the authors consists of the following: for an accurate calculation of nuclear shielding constants, a basis set of at least valence triple-$\zeta$ quality and with at least one set of polarization functions is needed. This conclusion is in accordance with Carmichael's observation [543] that, for the reliable calculations of the shielding constants, there is a need for flexibility in the outer-core inner-valence regions. Basis sets with tightly contracted core orbitals, such as the correlation-consistent basis sets and the small ANO sets, have little flexibility in the core region and perform poorly in the calculations of nuclear shielding constants. In general, just as in the case of SSCCs, standard, energy-optimized, single-electron basis sets are ineffective for the nuclear shielding constants, since in order

to approach the CBS limit within the framework of a particular method, it is necessary to use rather large, non-specialized basis sets.

A real breakthrough in the development of basis sets for nuclear shielding constants was provided by Jensen, who presented his famous specialized (σ-oriented) basis sets (aug)-pcS-*n* and pcSseg-*n* (*n* = 0–4) [544,545] for elements of 1–3 periods (H-Ar). He was guided by the rule that the rate of convergence over the basis set directly depends on the saturation of *p*-shell of basis sets. This conclusion was first derived by Jensen [544] on the basis of a general expression for the paramagnetic contribution to the shielding tensor, see Equation (8), which, in addition to the matrix elements of the orbital Zeeman operator, includes the matrix elements of the PSO operator. Therefore, based on the idea that saturation of the tight *p*-region of basis sets results in a faster convergence of the PSO term of the SSCCs, Jensen suggested that the same tendency would also take place in the case of paramagnetic contributions to the nuclear shielding constants. To confirm the assumption, Jensen performed a systematic analysis of the convergence of the nuclear shielding constants calculated within the GIAO-DFT formalism using the B3LYP and KT3 functionals on the standard energetically optimized polarization-consistent uncontracted basis sets pc-*n* (*n* = 0–4), expanded with additional diffuse and tight functions. He showed that only tight *p*-functions have a significant effect on the values of the nuclear magnetic shielding constants. Indeed, this is coherent with his original assumption about the PSO contribution. Diffuse functions in some cases also have a significant effect, which may be due to the Zeeman orbital operator or simply to the fact that polar systems with lone electron pairs require diffuse functions for a correct description. The diamagnetic contributions did not reveal any additional requirements for basis sets. Jensen has found that additional tight *p*-functions also affect the shielding constants of light *s*-block nuclei, though to a much lesser extent than the shielding constants of *p*-block elements. However, when creating pcS-*n* basis sets, he added one tight *p*-type function to the pc-*n* basis sets of all the elements, for the sake of systematic improvement.

To determine the optimal parameters for the additional tight *p*-type functions, Jensen used an optimization procedure similar to that used when creating his *J*-oriented basis sets, which consisted in maximizing the deviation of the NMR shielding constants relative to the value obtained with the pc-*n* basis sets. The contraction of the pcS-*n* basis sets was carried out in accordance with the general contraction scheme [546], using atomic orbital coefficients, and the contraction rate was determined based on the criterion according to which the contraction error should not exceed the error of the decontracted basis set relative to the CBS limit. The pcSseg-*n* series proposed by Jensen [545] also represents a family of σ-oriented, segmented-contracted basis sets for the elements H-Ar and K-Kr optimized for calculating NMR shielding constants. In general, both series are suitable not only for the DFT calculations, but also for the calculations within the framework of correlated wave function methods.

At the moment, there are no more specialized σ-oriented basis sets obtained at the nonrelativistic level, however, a publication has recently appeared in which, for the first time, the segmented-contracted relativistic basis sets x2c-SVPall-s and x2c-TZVPall-s were presented for the calculations of NMR shielding constants [547] of almost all nuclei: H-Rn, La-Lu. These sets were developed on the basis of relativistic Karlsruhe basis sets x2c-XVPall (X = S, TZ) [548]. The property-tailored basis sets, x2c-SVPall-s and x2c-TZVPall-s, were constructed in several steps focusing on the valence and the core region. At first, errors in the valence region of the existing all-electron relativistic Karlsruhe basis sets with respect to the reference even-tempered basis set (with a factor of $10^{1/4}$ between its exponents) were determined for a large set of more than 250 molecules. For the heavy elements, functions with larger exponents were added to flexibilize the description of the density in the outer-core region. The exponents were optimized in several cycles reducing the mean absolute error in the NMR shielding constants. To improve the description of the inner-most shells, tight *p* functions were introduced, namely a single additional *p* function was added for each element. Therefore, the exponent of the inner-most primitive function

of the parent x2c-SVPall or x2c-TZVPall sets was scaled with a factor of 6.5 and the outermost primitive was excluded from the segment and utilized as augmenting function to increase the flexibility. The contraction coefficients of the new segment were re-optimized at the X2C level of theory in atomic calculations resorting to quasi-Newton algorithm based on the variational principle. The x2c-SVPall-s and x2c-TZVPall-s basis sets were further compared to the segmented-contracted Jensen's basis sets pcSseg-*n* (*n* = 0–4) based on the percent-wise error [545] measured against the large reference even-tempered basis set.

### 4.2. Vibrational Corrections

The molecular rotational and vibrational motions determine both the vibrational spectrum and the vibrationally averaged molecular properties. Even at the temperature of absolute zero, the molecules have the so-called zero-point vibrations (ZPV), which affect the NMR properties. Many approaches were developed for taking into account vibrational effects on molecular properties. Among the most important methodologies are those which use the perturbation expansions to obtain vibrational frequencies and vibrationally averaged molecular properties [549–580] and those which resort to a conceptually different variational approach [581–586].

The most effective approach for the calculation of the vibrational wavefunction and zero-point vibrational corrections to molecular properties of polyatomic molecules was presented by Ruud, Åstrand, and Taylor [587,588]. This was implemented into the Dalton program package [589], which provides an efficient automated procedure for calculating the rovibrationally averaged molecular geometries and a large number of second-order molecular properties, including the NMR shieldings and SSCCs, using the SCF and MCSCF wave functions. A brief description of the approach introduced by Ruud, Åstrand, and Taylor, is presented further.

To take into account the effect of vibrations on a specific property $P$, it is necessary to average the property over the vibrational wave function $\Psi$:

$$< P > = \frac{\langle \Psi | P | \Psi \rangle}{\langle \Psi | \Psi \rangle}. \tag{105}$$

The property $P$ is expanded in a Taylor series around an arbitrary expansion point $r_{\text{exp}}$ as follows:

$$P(q_1, q_2, \ldots, q_N) = \sum_m P_m = P_{\text{exp}}^{(0)} + \sum_{i=1}^{N} P_{\text{exp},i}^{(1)} q_i + \frac{1}{2} \sum_{i,j=1}^{N} P_{\text{exp},ij}^{(2)} q_i q_j + \ldots, \tag{106}$$

where $q_i$ is a mass-weighted displacement of the nuclei from the expansion geometry along the normal coordinate $i$ ($q_i = r_i - r_{\text{exp},i}$), and $N$ is the number of normal coordinates, namely, 3*K*–6 in general case, or 3*K*–5 for linear molecules, with *K* being the number of atoms in the molecule. $P_{\text{exp},i_1,i_2,\ldots,i_n}^{(n)}$ represents the *n*th derivative of the property at the expansion point with respect to normal coordinates; $P_{\text{exp}}^{(0)}$ is the property at the expansion point. To find the vibrational wave function $\Psi$, a standard Rayleigh–Schrödinger vibrational perturbation theory is applied. The vibrational perturbation theory uses the harmonic oscillator Hamiltonian as the zeroth-order Hamiltonian. The latter represents the sum of the nuclear kinetic energy operator and the quadratic term of the potential energy surface expansion in terms of deviations $q_i$:

$$H^{(0)} = \frac{1}{2} \sum_{i=1}^{N} \left[ -\frac{\partial^2}{\partial^2 q_i} + V_{\text{exp},ii}^{(2)} q_i^2 \right], \tag{107}$$

where $V_{ii}^{(2)}$ is the second derivative of the potential with respect to the normal coordinates at the expansion point. The remaining terms of the expansion of the potential can be considered as perturbations to $H^{(0)}$:

$$H^{(1)} = \sum_{i=1}^{N} V_{exp,i}^{(1)} q_i + \frac{1}{6} \sum_{ijk} V_{exp,ijk}^{(3)} q_i q_j q_k, \tag{108}$$

$$H^{(2)} = \frac{1}{24} \sum_{ijkl=1}^{N} V_{exp,ijkl}^{(4)} q_i q_j q_k q_l. \tag{109}$$

The Hamiltonian for the harmonic oscillator $H^{(0)}$ has the eigenvalues:

$$E^{(0)} = \sum_{i=1}^{N} \left[ n_i + \frac{1}{2} \right] \omega_i, \quad \omega_i = \sqrt{V_{ii}^{(2)}}, \tag{110}$$

and the corresponding eigenfunctions are the products of Hermite polynomials:

$$\Psi^{(0)} = \prod_{i=1}^{N} \psi_{n_i} = \prod_{i=1}^{N} N_{n_i} H_{n_i}(\xi_i) e^{-\frac{1}{2}\xi_i^2}, \quad \xi_i = \sqrt{\omega_i} q_i. \tag{111}$$

The Rayleigh–Schrödinger perturbation theory gives the corrections $E^{(n)}$ and $\Psi^{(n)}$ to $E^{(0)}$ and $\Psi^{(0)}$. In the first order, $E^{(1)} = <\Psi^{(0)} | H^{(1)} | \Psi^{(0)}>$ equals to zero, as $H^{(1)}$ (see Equation (108)) contains only the odd terms with respect to at least one geometrical displacement $q_i$. The second-order energy can be regarded as an energy functional:

$$\widetilde{E}^{(2)} = \left\langle \Psi^{(0)} \middle| H^{(2)} \middle| \Psi^{(0)} \right\rangle + 2 \left\langle \Psi^{(0)} \middle| H^{(1)} - E^{(1)} \middle| \widetilde{\Psi}^{(1)} \right\rangle + \left\langle \widetilde{\Psi}^{(1)} \middle| H^{(0)} - E^{(0)} \middle| \widetilde{\Psi}^{(1)} \right\rangle, \tag{112}$$

where $\left| \widetilde{\Psi}^{(1)} \right\rangle$ is the trial function, which can be expressed in the harmonic oscillator eigenfunctions as follows:

$$\widetilde{\Psi}^{(1)} = \sum_{i=1}^{N} \sum_{r=1}^{\infty} a_{r,i}^{(1)} \phi_{r,i} + \sum_{i,j=1;i\neq j}^{N} \sum_{r,s=1}^{\infty} b_{rs,ij}^{(1)} \phi_{rs,ij} + \sum_{i,j,k=1;i\neq j\neq k}^{N} \sum_{r,s,t=1}^{\infty} c_{rst,ijk}^{(1)} \phi_{rst,ijk}, \tag{113}$$

with $\phi_{rs...t,ij...k} = \psi_{r,i}\psi_{s,j}\ldots\psi_{t,k}$, $i \neq j \neq k$.

Taking into account the expansions for $P$ and $\Psi$, the averaged molecular property (Equation (105)) is expressed as a power series with formal perturbational parameter $\lambda$:

$$\langle P \rangle = \sum_{mn} \left\langle P_m^{(n)} \right\rangle = \sum_{mn} \left[ \sum_{k=0}^{\infty} \left\langle \lambda^k \Psi^{(k)} \middle| P_m \middle| \lambda^{n-k} \Psi^{(n-k)} \right\rangle \right] \times \left[ 1 + \sum_{m=1}^{\infty} \sum_{l=1}^{\infty} (-1)^m \left\langle \lambda^l \Psi^{(l)} \middle| \lambda^l \Psi^{(l)} \right\rangle^m \right]. \tag{114}$$

The leading term of the normalization factor of the wave function in Equation (114) is noted to contribute to the second order in $\lambda$, thus the zeroth- and first-order terms in property expansion are read as follows:

$$\left\langle P^{(0)} \right\rangle_{exp} = P_{exp}^{(0)} + \frac{1}{4} \sum_{i=1}^{N} \frac{P_{exp,ii}^{(2)}}{\omega_i}, \tag{115}$$

$$\left\langle P^{(1)} \right\rangle_{exp} = \left\langle P_1^{(1)} \right\rangle_{exp} + \left\langle P_3^{(1)} \right\rangle_{exp} + \ldots, \tag{116}$$

where $\left\langle P_1^{(1)} \right\rangle_{exp}$ and $\left\langle P_3^{(1)} \right\rangle_{exp}$ represent the summations of the products of the first and third derivatives of the property $P$, correspondingly, with respect to the normal coordinates with the coefficients from the expansion of Equation (113) and some $\omega_i$-depending multipli-

ers (for the exact equation, see [587]). Thus, in the second-order of vibrational perturbation theory, the ZPV correction to a property can be written as [590,591]:

$$\Delta_{ZPV}P = \frac{1}{4}\sum_{i=1}^{N}\frac{1}{\omega_i}P_{\exp,ii}^{(2)} - \frac{1}{4}\sum_{i=1}^{N}\frac{1}{\omega_i^2}P_{\exp,i}^{(1)}\sum_{j=1}^{N}\frac{V_{\exp,ijj}^{(3)}}{\omega_j},\tag{117}$$

where $V_{\exp,ijj}^{(3)} = F_{\exp,ijj}$ are the semi-diagonal cubic force constants at the expansion point.

Ruud, Åstrand, and Taylor proposed to use the effective geometry as the expansion point. An effective geometry is chosen such as to minimize the energy functional:

$$\widetilde{E}^{(0)} = V_{\exp}^{(0)} + \left\langle \widetilde{\Psi}^{(0)}\middle|H^{(0)}\middle|\widetilde{\Psi}^{(0)}\right\rangle.\tag{118}$$

At the effective geometry the gradient of $\widetilde{E}^{(0)}$ is zero, and the differentiation of the right-hand side of Equation (118) with respect to the expansion point gives the following relationship at the effective geometry:

$$V_{\text{eff},i}^{(0)} + \frac{1}{4}\sum_{j=1}^{N}\frac{V_{\text{eff},ijj}^{(3)}}{\omega_j} = 0.\tag{119}$$

In this case, $\left\langle P_1^{(1)}\right\rangle_{\exp} \equiv \left\langle P_1^{(1)}\right\rangle_{\text{eff}} = 0$, because of the condition (119).

The major contribution to the zero point vibrationally averaged property from the anharmonicity of the potential as calculated at the equilibrium geometry thus vanishes when the effective geometry is used as an expansion point. The vibrational correction to a molecular property $P$ from zero-point vibrational motion (ZPV) can be calculated from the zeroth-order vibrational wave function as follows:

$$\langle P\rangle_{0,0} = \left(P_{\text{eff}}^0 - P_e^0\right) + \frac{1}{4}\sum_{i=1}^{N}\frac{P_{\text{eff},ii}^{(2)}}{\omega_i},\tag{120}$$

where $P_{eff}^0$ and $P_e^0$ designate the property calculated at the effective and equilibrium geometry, respectively. For the nonzero temperatures, the vibrational averaging includes the excited states of the vibrational wave function involving the averaging over the vibrational states of the molecule with the Boltzmann distribution. Equation (120) does not imply that there is no anharmonicity of the potential included in the calculation of vibrationally averaged properties. Instead, the anharmonicity is included through the use of the effective geometry instead of the equilibrium geometry as an expansion point for the vibrational wave function.

In brief, the complete determination of the ZPVCs can be done in a two-step procedure: (1) determining the effective (vibrationally averaged) geometry by the calculation of parts of the cubic force field; (2) at the effective geometry, determine the harmonic force field and calculate the second derivatives of the molecular properties of interest using the numerical differentiation along the normal coordinates.

At non-zero temperatures, there is a portion of molecules that are in the excited vibrational states. In most cases, these are usually neglected. However, temperature effects can be taken into account by calculating the thermal average of the VPT2-corrected properties. In the chapter presented by Faber, Sauer, and Kaminský [17] it is said that the thermal averaging can be performed by applying the Boltzmann distribution. Thus, the expression, which represents an approximation to the vibrational correction (with

taking into account rotational contributions) at a finite temperature, can be presented as follows [17]:

$$\Delta^{VPT2}P = -\frac{1}{2}\sum_i \frac{1}{\omega_i}\frac{\partial P}{\partial q_i}\bigg|_{\mathbf{q}=0}\left(\frac{1}{2}\sum_j k_{ijj}\coth\left(\frac{hc\omega_j}{2kT}\right) - \frac{kT}{2\pi c}\sqrt{\frac{1}{hc\omega_i}}\sum_\alpha \frac{a_i^{\alpha\alpha}}{I_{\alpha\alpha}^e}\right) +$$

$$+ \frac{1}{4}\sum_i \coth\left(\frac{hc\omega_i}{2kT}\right)\frac{\partial^2 P}{\partial q_i^2}\bigg|_{\mathbf{q}=0}, \tag{121}$$

where $I_{\alpha\alpha}^e$ are the diagonal components of the moment of inertia tensor, $a_i^{\alpha\alpha}$ are the linear expansion coefficient of the moment of inertia in the normal coordinates.

### 4.2.1. Vibrational Corrections to Spin–Spin Coupling Constants

In some cases, the vibrational corrections to the SSCCs can reach significant values, since this property has a very strong dependence on the geometry of the molecule. Therefore, the vibrational corrections are among the most important factors of accuracy of quantum chemical calculations of SSCCs. In particular, for the SSCCs involving non-relativistic nuclei, the difference between the theoretical and experimental values is often explained by unaccounted vibrational corrections, which can reach up to 10% of the total values of SSCCs [407,592,593].

Within common vibrational perturbation theory, the coupling constant in the vibrational ground state, $\langle J\rangle_0$ can approximately be calculated from the value at the equilibrium geometry, $J_{eq}$, and zero-point vibrational correction (ZPVC), $\Delta J_{vib}$, as follows [561,590,591]:

$$\langle J\rangle_0 = J_{eq} + \Delta J_{vib} = J_{eq} + \left\{-\frac{1}{4}\sum_{i=1}^N \frac{1}{\omega_i^2}J_{eq,i}^{(1)}\sum_{j=1}^N \frac{F_{eq,ijj}}{\omega_j} + \frac{1}{4}\sum_{i=1}^N \frac{1}{\omega_i}J_{eq,ii}^{(2)}\right\}, \tag{122}$$

where $J_{eq,i}^{(1)}$ and $J_{eq,ii}^{(2)}$ are the first and second derivatives of the coupling constants with respect to displacements $q_i$; $\omega_i$ are the harmonic vibrational frequencies, and $F_{eq,ijj}$ are the semi-diagonal cubic force constants. Thus, the calculation of vibrational corrections requires the calculation of the geometric derivatives of the potential energy surface and the SSCCs by nuclear coordinates. This results in the large number of repeated calculations of the SSCCs for different molecular geometries. In general, this is a very time-consuming procedure, which requires large computational effort. At that, due to the particular form of the of the operators representing the interaction between the nuclear spins and the electronic spin and angular momentum, SSCCs require specialized $J$-oriented basis sets and computational approaches that circumvent the triplet instability problems manifesting in the calculations of the FC and SD contributions. Therefore, the calculation of the vibrational corrections to the NMR spin–spin coupling constants is generally considered to be more challenging task than the calculation of those to any other linear response property.

A lot of efforts have been made to take into account the vibrational effects on SSCCs. In particular, an exceptionally large value of the ZPVC correction of −25 Hz to SSCC of hydrogen fluoride molecule was calculated by Astrand et al. [594] at the MCSCF level. Wigglesworth et al. [407] calculated vibrational corrections to SSCCs in acetylene at the SOPPA(CCSD) level. Stanton and Sneskov [595] performed the calculations of vibrational corrections to SSCCs of various types in acetylene, ethylene, ethane, and cyclopropane at the CCSD level of theory. According to their results, the value of the vibrational corrections ranged from 8 to 32% for the carbon-proton SSCCs and from 1 to 7% for the carbon-carbon SSCCs of the total values. Kirpekar et al. [596] carried out the calculations of vibrational corrections to one-bond SSCCs in XH$_4$ molecules (X = C, Si, Ge, Sn) at different temperatures at the RPA, SOPPA, and CASSCF levels. According to their results, the value of the vibrational corrections to the considered SSCCs, was found to be about 1–3%, in average, of the total values of SSCCs. Yachmenev et al. [597] estimated the vibrational corrections to

nitrogen-proton and proton-proton SSCCs at different temperatures, namely 0 and 300 K, in ammonia isotopomers. For nitrogen-proton and deuterium-proton SSCCs, it was shown that the total vibrational corrections are about 0.6% and 5% of the total values, respectively. The effect of non-zero temperature was found to be insignificant, amounting in hundredths of Hz. Jordan et al. [598] calculated the vibrational corrections for nitrogen–nitrogen SSCC through the hydrogen bond in model hydrogen-bonded complex, CNH:NCH. In that paper, the expectation values of $<^2J(N,N)>_{0,vib}$ were obtained from the two-dimensional potential energy surface for $CN_aH:N_bCH$, generated in the $N_a$-H and $N_b$-H distances at the MP2 level and the global SSCC surface, which was calculated at the EOM-CCSD level of theory. Despite the fact that only two vibrational modes were considered, the property calculations were performed for 108 single-point molecular geometries, which illustrates the extremely high computational cost of the vibrational problem. Del Bene et al. [599] carried out the calculations of the vibrational effects on the F-F SSCCs ($^{2h}J_{F-F}$) for the $FHF^-$ molecule. The coupling constant surface was generated at the EOM-CCSD level, and two-dimensional wavefunctions for the symmetric and asymmetric stretching vibrations were obtained from the potential energy surface evaluated at the CCSD(T) level of theory. The effect of $FHF^-$ bending mode was also investigated using the one-dimensional calculations along the bending normal coordinate. In the ground vibrational state, the expectation value of F-F SSCC, $<^{2h}J(F,F)>_{0,vib}$, was found to be 212.7 Hz, which is significantly less, namely, by 41.7 Hz, than that at the equilibrium geometry (254.4 Hz). At the same time, the effect of the bending mode was found to be unessential, namely, in the ground vibrational state of the bending mode, the average value of $^{2h}J(F,F)$ is 253.2 Hz, which is very similar to the equilibrium value of 254.4 Hz. This small effect was explained by the large difference in masses of the hydrogen and fluorine atoms, which leads to the fact that the bending vibration principally involves the motion of the hydrogen atom, leaving the F-F distance practically unchanged.

A great deal of computations of vibrational effects on SSCCs were performed within the density functional theory by the main developers of VPT approach. In particular, Ruden, Lutnæs, Helgaker, and Ruud [591] investigated the convergence of ZPVCs to various SSCCs in small molecules ($H_2$, HF, CO, $N_2$, $H_2O$, HCN, $NH_3$, $CH_4$, $C_2H_2$) on the basis set at the DFT(B3LYP) level. They considered two series of Huzinaga basis sets. The first sequence consisted of the Huzinaga sets HII, HIII, and HIV [503] with the polarization functions and contraction patterns of van Wüllen and Kutzelnigg et al. [600], and the second series included the Huzinaga's basis sets, possessing enlarged flexibility in the inner core region, namely the HX-su*n* basis sets (X = II, *n* = 2; X = III, *n* =3; X = IV, *n* = 4). They have found that the HIV-su4 basis set gives the vibrational corrections close to the CBS limit, achievable within the DFT model. Therefore, it was recommended to use the HIV-su4 basis set only in very precise calculations of vibrational corrections to SSCCs in very small systems. At the same time, smaller basis sets such as HIII-su3, also give very good accuracy, therefore, it was recommended to use them in the routine calculations of vibrational effects on the SSCCs of larger systems. The ZPVCs obtained in ref. [591] were compared with those calculated in previous works [404,405,407,412,592,594,601–603]. It was found that the DFT vibrational corrections are in good agreement with those calculated using the other correlated non-empirical methods, except for two cases of striking differences—the $^1J_{NN}$ in $N_2$ and $^3J_{HH}$ in $C_2H_2$. Although the obtained B3LYP results turned out to be close to experiment, the authors did not attach much significance to this fact since, for the particular systems, the B3LYP might predict much too low equilibrium coupling constants. In general, it is well known that restricted Kohn–Sham theory is known to manifest the triplet instability problem [316], which leads to an unbalanced description of the ground state and the most important excited states of a given symmetry, thus, providing a poor description of the molecular property of interest that depends on these states. In the case of SSCCs, the triplet instability results in incorrect calculation of the SD and FC terms (which are the triplet second-order properties) and may sometimes give an error of several orders of magnitude.

In the subsequent work, Ruden et al. [604] presented a systematical comparison of the SSCCs in allene, cyclopropane, cyclopropene and cyclobutene, calculated at different levels of electronic theory, namely using the CCSD, SOPPA, MCSCF, and DFT(B3LYP) approaches with taking into account the vibrational corrections, against the experimental data. The ZPV corrections to the spin–spin coupling constants were calculated at the B3LYP level of theory, using the theory of Ruden et al. [591]. In the paper by ref. [604], the Fermi-contact contribution to the ZPV correction was calculated using the HIIsu2 basis set, whereas the HII basis was used for the other three contributions. Based on the idea that the spin–spin vibrational corrections are typically of the same order of magnitude as the differences between theory and experiment, Ruden et al. estimated the accuracies of the considered methods from the comparison of two types of values: the differences between the experimental SSCCs and the vibrational corrections and the theoretical values of those SSCCs computed at different levels of theory. In general, it was found that the effects of electron correlation are underestimated by MCSCF theory, somewhat better described by SOPPA, and well described by CCSD theory. Hybrid B3LYP was found to perform as well as SOPPA for the one-bond coupling constants, while, for the other constants, it provided results of similar quality as CCSD.

### 4.2.2. Vibrational Corrections to NMR Shielding Constants

Besides spin–spin coupling constants, nuclear shielding constants are also dependent on the molecular geometry to a high extent. In general, it has been shown that the effects of nuclear motion can be as important for the shielding constants as the effects of electron correlation [407,588,605–607]. For instance, as was shown by Ruud et al. [588], that the Hartree–Fock value of the nitrogen shielding constant in ammonia is approximately 262 ppm. Taking into account the effects of electron correlation increases this value up to 274 ppm. Taking into account ZPV effects decreases this to 267 ppm. The experimental value is about 265 ppm. Thus, one can see that the effects of electron correlation and vibrational degrees of freedom to large extent compensate each other, so that the Hartree–Fock method applied without taking into account any corrections may seem deceptively sufficient for an adequate prediction of the nitrogen shielding constants and chemical shifts.

Within the second-order VPT approach, the averaged nuclear shielding constant over the vibrational ground state, can approximately be presented as follows [590]:

$$\langle\sigma\rangle_0 = \sigma_{eq} + \Delta\sigma_{vib} = \sigma_{eq} + \left\{ -\frac{1}{4}\sum_{i=1}^{N}\frac{1}{\omega_i^2}\sigma_{eq,i}^{(1)}\sum_{j=1}^{N}\frac{F_{eq,ijj}}{\omega_j} + \frac{1}{4}\sum_{i=1}^{N}\frac{1}{\omega_i}\sigma_{eq,ii}^{(2)} \right\}, \qquad (123)$$

The ZPVCs to the NMR shielding constants of a wide range of NMR-active nuclei have been studied systematically over the past 40 years [608–614].

Fukui et al. [615] calculated NMR shielding constants in the first- and second-row hydrides, taking into account the electron correlation effects within the finite-field Møller–Plesset perturbation theory through the third order (FF-MP2, FF-MP3) and the rovibrational corrections. For the polyatomic molecules, rotational motions were neglected because of the earlier finding that the centrifugal distortion due to molecular rotational motions is usually one order of magnitude smaller than the effect due to anharmonic vibrations [614]. It was shown that the calculated isotropic shielding constants at the experimental geometries are higher than the experimental values, but the vibrational corrections are generally negative and improve the calculated shielding constants.

Sundholm et al. [259] calculated NMR shielding tensors for $H_2$, HF, $N_2$, CO, and $F_2$ at the coupled cluster singles and doubles level augmented by a perturbative correction for triple excitations, CCSD(T). The shielding constants for the lowest rovibrational states of the considered diatomic molecules were obtained by solving the rovibrational Schrödinger equation with the finite-element techniques followed by evaluating appropriate expectation values. Temperature effects have been accounted for by applying the Boltzmann averaging. The total calculated shielding constants were in good agreement with the available exper-

imental values, except for the $F_2$ molecule. The deviations observed for the $F_2$ molecule turned out to be about 4 ppm smaller than the experimental value.

Lonila et al. [616] investigated the vibrational and temperature effects on the carbon and selenium shielding constants in carbon diselenide, $CSe_2$. The shielding tensors of $^{13}C$ and $^{77}Se$ nuclei, with and without taking into account the vibrational and rotational degrees of freedom, were calculated using several ab initio methods in conjunction with the GIAO formalism. The results obtained at the CHF, MCSCF (RAS, CAS), and DFT (LDA, BLYP and BPW91) levels were compared with theoretical data taken from other different sources. Thus, according to the results, obtained within the CAS and DFT(BPW91) methods, the vibrational-rotational corrections (T = 300 K) to $\sigma(^{77}Se)$ in the $^{77}SE=^{13}C=^{80}Se$ isotopomer is about 2–2.5% relative to the full values. The vibrational-rotational corrections to the $\sigma(^{13}C)$ were found to be about 5.5–6%. Overall, it was found that the effect of rotational degrees of freedom does not exceed hundredths of a percent of the total value of both shielding constants. In the particular case of the $CSe_2$ molecule, the authors have come to a conclusion that taking into account vibrational and rotational degrees of freedom worsens the agreement of theoretical values with the experiment. This observation has been explained by the lack of relativistic and solvent corrections.

The effect of hydrogen binding and vibrational motions on the oxygen and hydrogen nuclear magnetic shielding constants of the $OH^-$ and $H_3O_2^-$ were investigated by means of ab initio calculations at the RPA and SOPPA levels by Sauer et al. [617]. The effective shielding constants were obtained by averaging of the property over the rovibrational wavefunctions, which are the solutions of the one-dimensional radial Schrödinger equation, solved numerically [618]. The dependence of the nuclear magnetic shielding constants in $H_3O_2^-$ on the anharmonic symmetric and antisymmetric $O \cdots H \cdots O$ stretching motions and on the internal rotation motion of the outer hydrogens was studied with the non-rigid bender model Hamiltonian [619] at the RPA level. The dependence of the shielding constants in $OH^-$ on the bond length was investigated at RPA and SOPPA levels. For all atoms of $H_3O_2^-$, with the exception of the outer hydrogen atoms, a strong dependence on the vibrational quantum number was found for the nuclear magnetic shielding constants. Namely, the NMR shielding constant of the oxygen atom in $H_3O_2^-$ and $OH^-$ were found to increase and decrease, respectively, with the vibrational quantum number. For the hydrogen shielding constants, the opposite behavior was found.

Minaev et al. [620] examined the dependence of the electronic spin–orbit coupling contribution to the proton shielding constant of hydrogen iodide on the internuclear distance using the quadratic response theory. The calculations of proton shielding constant were performed using a complete active space self-consistent field (CASSCF) wavefunction at different internuclear distances. It was shown that spin–orbit coupling correction to the $^1H$ shielding constant manifests strong dependence on the internuclear separation. The zero-point vibrationally averaged shielding constant was calculated as the averaged Taylor series expansion:

$$\sigma(H) = \sigma_e + \sigma_r < r - r_e >^{0\,K} + \frac{1}{2}\sigma_{rr} < (r - r_e)^2 >^{0\,K}, \tag{124}$$

in terms of the deviation of internuclear distance (bond length, $r$) from its equilibrium value $r_e$. The first and second derivatives of the shielding constant, $\sigma_r$ and $\sigma_{rr}$ were determined by fitting the fourth-order polynomial to the shielding constants calculated in the bond-distance interval 1.4–1.8 Å, with points separated by 0.05 Å. The averages of the bond length extension and its square were calculated using the harmonic and cubic force constants, $F_{rr}$ and $F_{rrr}$ by fitting the fourth-order polynomial to the total SOC corrected energies. It was found that the total nonrelativistic vibrationally averaged $^1H$ shielding constant for HI is slightly increased compared to the value calculated at the equilibrium distance (tenth of ppm). At that, a heavy atom SOC induced effect was shown to reverse the sign of the vibrational contribution to the $^1H$ shielding.

For the $H_2O$ molecule, the vibrational effects on nuclear shielding constants were studied thoroughly [561,574,588,615,621–624]. A pioneering study was performed by Fowler and Raynes [574], using an empirical force-field with HF shielding surfaces to obtain a ZPVC of $-13.1$ ppm to $\sigma(^{17}O)$ of $H_2^{17}O$. Then, the correlated study of rovibrational effects on the nuclear shielding constants in the water molecule was performed by Vaara et al. [625]. The restricted active space self-consistent field (RASSCF) wave function with large RAS and large basis sets were used to calculate the rovibrational corrections and the related temperature and isotope dependencies with high accuracy. It was shown that the rovibrational effects are as important as those of electron correlation. In particular, the rovibrational corrections were found as 3.7% and 1.8% for the isotropic oxygen and hydrogen shielding constants, respectively, in the $^1H_2^{17}O$ isotopomer at 300 K. On the basis of the calculations presented in ref. [625] and the CCSD(T) results of Gauss et al. [259,626], a new absolute shielding scale for the $^{17}O$ nucleus was proposed, namely the value of the oxygen shielding constant of $H_2^{17}O$ isotopomer in the gas phase at 300 K was established as $324.0 \pm 1.5$ ppm. Wigglesworth et al. [621] have also performed ab initio calculations of hydrogen and oxygen shielding surfaces for the water molecule at the MCSCF level. The rovibrationally averaged shielding constants of the various isotopomers of water and their temperature dependences were obtained. To determine the relevant coefficients for the expansions of the $\sigma(H)$ and $\sigma(O)$ around the equilibrium geometry in terms of the symmetry coordinates, describing the displacements from equilibrium geometry, the calculations of shielding constants at 49 distinct locations on the proton surface and 37 distinct locations on the oxygen surface were carried out. Wigglesworth et al. obtained ZPVC to $\sigma(^{17}O)$ of $H_2^{17}O$ as $-9.9$ ppm.

The temperature dependences for $\sigma(H)$ in $H_2^{16}O$ and $\sigma(O)$ in $H_2^{17}O$ were simulated as follows:

$$\sigma(H) = 30.232 - 9.7 \times 10^{-5}(T - 300) + 4 \times 10^{-8}(T - 300)^2, \tag{125}$$

$$\sigma(O) = 333.723 - 1.18 \times 10^{-3}(T - 300) + 1.7 \times 10^{-7}(T - 300)^2. \tag{126}$$

An excellent agreement was observed between the shielding surfaces of totally independent studies of Vaara et al. and Wigglesworth et al. However, the $\sigma_e(^{17}O)$ for equilibrium value occurred to be substantially different. The value of $\sigma_e(^{17}O)$ proposed by Wigglesworth et al. was 333.723 ppm, while that of Vaara et al. was $324.0 \pm 1.5$ ppm. Wigglesworth et al. [605] continued the investigation of the rovibrational effects on carbon and hydrogen NMR shielding constants on example of acetylene molecule. The calculations were performed at the correlated MCSCF level of theory using gauge-including atomic orbitals and a large basis set.

Later studies, such as those of Ruud et al. [588], Auer [622], and Kupka et al. [623], used general approaches based on normal coordinate expansions of the surfaces. The most thorough study of the nuclear shielding of water was performed by Puzzarini et al. [624] as an attempt to redefine the absolute shielding scale of $^{17}O$. They used the CCSD(T) method with a large basis set and an accurate numerical description of the vibrational problem to compute the ZPVC of $-11.7$ ppm. Komorovsky et al. [627] combined previously published high-quality experimental spin–rotation data, accurate coupled cluster calculations of Puzzarini and personal relativistic four-component Kohn–Sham density functional calculations of the shielding and spin–rotation constants of $H_2^{17}O$, and revised the absolute shielding value for the $^{17}O$ nucleus of $H_2^{17}O$ as 328.4(3) ppm at 300 K.

Faber et al. [628] have investigated the question of the convergence of zero-point vibrational corrections to nuclear shielding constants and shielding anisotropies towards the complete basis set limit on the example of water. The calculations of ZPVCs at the HF, MP2, CCSD, CCSD(T), and DFT(B3LYP) levels with the cc-pVXZ, aug-cc-pVXZ, cc-pCVXZ, aug-cc-pCVXZ (X = D, T, Q, 5, 6), and aug-pc-$n$ and aug-pcS-$n$ ($n = 1, 2, 3, 4$) series of basis sets were carried out. None of the basis sets exhibited a monotonic convergence for the ZPVC. Four of the five calculations using varying basis sets series agreed on the final value

of the ZPV correction to $\sigma(^1H)$ and $\sigma(^{17}O)$, whereas aug-pc-*n* predicted a slightly smaller absolute value.

For both, the equilibrium geometry and the vibrationally averaged values of shielding constants, the basis set convergence, observed when using the non-augmented basis sets, was rather slow (it was not achieved until the sextuple zeta level). Adding the diffuse functions significantly accelerated the convergence, so that it was reached already at the quintuple zeta level. For ZPVC to oxygen shielding constants, it was found that the convergence is not monotonic in most cases, which means that one cannot accurately extrapolate this ZPVC to the CBS limit from calculations with smaller basis sets.

The vibrational corrections to $^{125}$Te NMR chemical shifts were recently calculated by Rusakova et al. [629]. In that paper, the main factors affecting the accuracy and computational cost of the calculation of $^{125}$Te NMR chemical shifts in medium-size organotellurium compounds were analyzed at the GIAO−DFT level. The LDBS schemes, relativistic corrections, solvent effects and vibrational corrections were considered as the primary accuracy factors.

For the benchmark series of six tellurium compounds, the average relativistic corrections to tellurium chemical shifts were found to be about 22% in relation to their total values. The solvent and vibrational corrections amounted to 8% and 6%, in average. As can be seen from Figure 2, the MAPE calculated within the series of six tellurium compounds, gradually decreases from 24% for the values obtained at the nonrelativistic GIAO-DFT(PBE0) level to 4% for the values obtained at the full four-component relativistic level, with solvent and vibrational corrections taken into account.

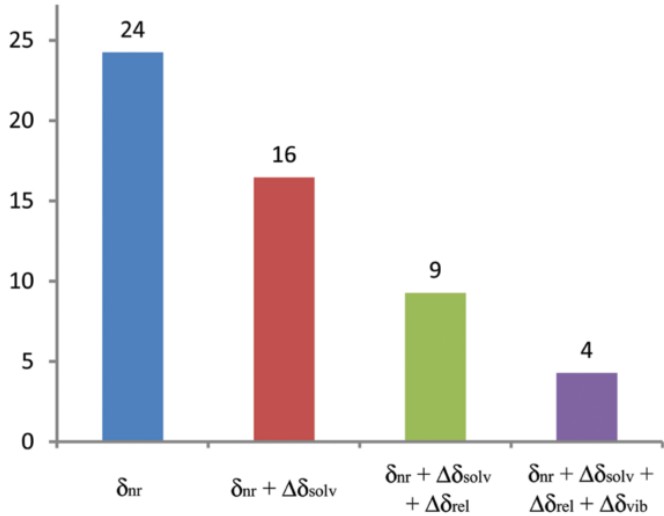

**Figure 2.** MAPEs of $^{125}$Te NMR chemical shifts of six organotellurium compounds calculated at the GIAO-PBE0 level, taking into account solvent, relativistic, and vibrational corrections in comparison with the experiment. Reproduced from Ref. [629] with permission from the American Chemical Society.

*4.3. Solvation Models*

The nuclear shielding and spin–spin coupling constants are sensitive to the electronic structure of a molecule. At the same time, these parameters are also sensitive to the intermolecular interactions and solvent effects. It is generally known that spin–spin coupling constants are less influenced by solvent effects than the nuclear shielding constants. The solvent effects on SSCCs are usually small, as they hardly exceed a few per cent of the total value in most cases, while the nuclear shielding constants are extremely sensitive to the molecular environment, and especially to hydrogen-bonding effects. For producing high-quality predictions of NMR parameters, it is important to take into account the media

effects during both the geometry optimization and the quantum chemical calculations of NMR properties. Overall, this increases the precision of modelling of the NMR spectra and provides a valuable improvement in determining chemical structures, since most experimental NMR studies take place in solution.

In general, one can discern four distinct contributions to the changes of SSCCs due to solvent effects. This includes [630]: (1) magnetic shielding of the nuclear spin–spin coupling by magnetic moments induced in the neighboring molecules by the nuclear dipoles [631,632], which is proportional to the volume magnetic susceptibility of the solution; (2) contributions arising from intermolecular dispersion effects; (3) the effect of intermolecular electrostatic interactions, which will have its smallest value for the case of nonpolar solutes in nonpolar solvents, and have its greatest value for the case in which both the solute and the solvent are electrolytes; (4) specific interactions between solute and solvent molecules, such as charge transfer or hydrogen bonding.

The effects of a solvent on nuclear shieldings can also be divided into four different types [633]: (1) the change in the local magnetic field experienced by the nucleus due to the isotropic magnetizability of the solvent molecules, which is proportional to the magnetizability of the solvent; (2) a change in the local magnetic field due to the magnetizability anisotropy of solvent molecules in the close vicinity; (3) the change in the electronic structure of the solute due to van der Waals interactions with solvent molecules; (4) the contribution from the electrostatic polarization of the solute's charge distribution.

In ab initio calculations, different contributions to solvent corrections to NMR parameters cannot be easily separated and defined, though, some models were presented to study some of the contributions [634]. Overall, there are no models that ensure the absence of the overlap between different contributions for now.

The influence of the solvent effects on NMR parameters can be implemented within the framework of two conceptually different classes of models, according to the microscopic description of the solvent [635]. The first class comprises the continuum (or implicit) models, which explicitly treat only the degrees of freedom associated with the solute, while replacing the solvent with a structureless continuum characterized by its bulk properties. The second class is the discrete (or explicit) models, which treat degrees of freedom associated with the solvent molecules and the solute explicitly.

The most popular continuum model is the polarizable continuum model (PCM) [636–638] and its new version, the integral equation formalism PCM, IEF-PCM [639–643]. In the polarizable continuum model, the solvent is represented by a homogeneous continuum medium, which is polarized by the solute placed in a cavity built in the dielectric medium. From basic electrostatics, it is known that the response of a homogeneous dielectric continuum to any charge distribution of the solute produces the charge distribution on the cavity surface, arising from the polarization of the dielectric medium. The main problem consists in the calculation of the screening charge density on the cavity surface. For the arbitrarily shaped surfaces, this cannot be determined by analytical means, and different numerical approaches are needed. For spherical and ellipsoidal cavities, the screening charge density can be found analytically, in particular, within the Onsager model [644]. The PCM method assumes that the solute charge density is entirely encapsulated in the cavity, however, this is often not the case, and the electron distributions often extend beyond the cavity. The IEF-PCM performs well in this respect, in contrast to the PCM model. Computational modeling of the solvent–solute effect on NMR molecular parameters by polarizable continuum model was thoroughly reviewed by Sadlej and Pecul [645].

The conductor-like screening model, COSMO [646–649], is another popular representative of the class of continuum models. This is an approximate, but very accurate, non-iterative approach for the solution of equation on the screening charge density for arbitrarily shaped cavities. This calculates the dielectric screening charges and energies on the van der Waals-like molecular surface in the approximation of a conductor (the dielectric permittivity ($\varepsilon$) is set to infinity), which is highly accurate and much more efficient compared to the solution of the dielectric boundary conditions. This formalism is to some

extent similar to the approach of Hoshi et al. [650], which is based on the Green's function and allows one to express the screening charge distribution as a linear function of the solute charge distribution. However, in contrast to Hoshi model, the COSMO algorithm leads to rather simple explicit expressions for the screening energy and its analytic gradient with respect to the solute coordinates. The COSMO algorithm was modified to the COSMO-RS (COSMO for realistic solvents) [651] model, which takes into account the ability of the solvent to screen the surface charge on the cavity of the solute. Moreover, it has become possible to take into account the solvent effects via the cluster approach in the COSMO model, i.e., to treat the first solvation shell explicitly at the same level of theory as the solute on the ad hoc basis.

Another continuum-like model is the reaction-field method [652–655], also known as the multipole-expansion MPE method [656,657]. In MPE method, the atom, molecule, or supermolecule is assumed to be surrounded by a linear, homogeneous, cand ontinuous medium described by its macroscopic dielectric constant. A spherical or ellipsoidal [658] cavity is used for the solute and the interaction with the medium is calculated by a multipolar expansion. In general, the reaction-field method performs worse than the IEF-PCM method, most likely because of the less realistic cavity shape than what is adopted in the IEF-PCM.

A purely continuum description might fail in some cases, especially when specific solute–solvent interactions take place. In general, it is believed that reaction field (or continuum) methods provide an effective description of the long-range electrostatic interactions, while specific short-range interactions can be effectively described by the discrete models [635]. A supermolecule approach belongs to the discrete-type models. It treats the solute molecule in the surrounding of a number of explicitly treated solvent molecules. Within the supermolecule approach, it is possible to perform the thermodynamic averaging by a molecular dynamical (MD) or random Monte Carlo (MC) sampling of the relevant states. However, such calculations are still very time-consuming, even on the fastest computers, but they benefit considerably from the present trend to parallel computing.

Alternatively, quantum mechanics/molecular mechanics (QM/MM) methodologies [659–679] treat the chemically important part of the system with a quantum mechanical approach, while the rest part is treated with standard molecular mechanics, using molecular force field. Most QM/MM methods describe interactions between the QM molecular system and the environment using either the simple mechanical embedding scheme or the more accurate electrostatic embedding [680]. The regular QM/MM methods aim to derive effective operators which include the environmental effects in the QM region as opposed to a full quantum mechanical treatment of the whole system.

For spin–spin coupling constants, most simulations of solvation are carried out by means of polarizable continuum models. It is believed that the continuum method is, in principle, more suited for the calculations of solvent effects on the spin–spin coupling constants than on the shielding constants, since the spin–spin coupling constants depend primarily on the electron density at the nuclei, which makes them less susceptible to the specific solvent–solute interactions. Among the most popular models that have been adopted for taking into account solvent effects on spin–spin coupling constants are the following: the reaction-field method MPE, the IEF-PCM (integral equation formalism polarizable continuum model) [640,681], and the COSMO. The implementation of the QM/MM model to the calculation of the NMR indirect spin–spin coupling constants was presented by Møgelhøj and co-workers [679] as the extension of the explicitly polarizable QM/MM model within the density functional theory, although the elaborated theory is applicable to an arbitrary formalism. However, this complicated model received less popularity in the calculations of SSCCs by taking into account solvent effects as compared to the polarizable continuum model.

The PCM model has also become very popular in simulating solvent effects on nuclear magnetic shielding constants, despite the fact that the nuclear magnetic shielding tensors are much more sensitive to the local environment than SSCCs, especially when there are

specific intermolecular interactions such as hydrogen bonding or charge transfer. The theory of polarizable continuum for the gauge invariant atomic orbital (GIAO) calculation of nuclear magnetic shieldings for solvated molecules was presented by Cammi [682].

The QM/MM approaches have also received much popularity in calculations of nuclear magnetic shielding constants accounting for the solvent effects. One of the examples of the application of the QM/MM approach to the calculation of NMR shielding tensor of any nuclei was presented by Cui and Karplus [683]. In their approach, the solute and a number of solvent molecules ("solute-solvent cluster") were described with QM/MM, while the bulk solvent was treated with the IEF-PCM model. Both the MM atoms, represented by fixed partial charges, and the QM atoms, were contained in a generalized cavity embedded in the continuum. It was concluded that an appropriate QM/MM partition is capable of giving good descriptions of the environmental effects on chemical shift tensors. Kongsted et al. [678] also presented a gauge-origin independent hybrid quantum mechanics/molecular mechanics model for the calculation of nuclear magnetic shielding tensors of molecules placed in a structured and polarizable environment. The method is based on a combination of DFT or Hartree–Fock wave functions with molecular mechanics. The proposed method complies with the main requirements for an accurate calculation of the nuclear magnetic shielding constants: (1) it includes electron correlation effects, (2) uses gauge-including atomic orbitals to give gauge-origin independent results, and (3) the effect of the environment is treated self-consistently using a discrete reaction-field methodology.

The formulation of the four-component relativistic Hartree–Fock and Kohn–Sham theories for a molecular solute described within the framework of the polarizable continuum model has been presented by Di Remigio et al. [684]. The linear response function for the four-component PCM-SCF state was derived, allowing the four-component calculations of the NMR parameters, by taking into account the solvent effects within the PCM model. The algorithm was implemented into the DIRAC program package [366].

**Funding:** This research received no external funding.

**Institutional Review Board Statement:** Not applicable.

**Informed Consent Statement:** Not applicable.

**Data Availability Statement:** Not applicable.

**Conflicts of Interest:** The author declares no conflict of interest.

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
