# Peer review of "Quantum Chemical Approaches to the Calculation of NMR Parameters: From Fundamentals to Recent Advances"

_magnetochemistry, doi:10.3390/magnetochemistry8050050_

Round 1
Reviewer 1 Report
see attached

Author Response
This article is a thorough review on the state of the art of calculating NMR parameters. The review is nicely structured and well written. I have to admit that my NMR expertise is on the experimental side of NMR spectroscopy and not the NMR theory subject matter of this reviewer. That said, if I were interested to get into this theoretical field, I would be highly appreciative of having available such a nice review article. However, as a reviewer, I am falling short to be able to detect potential errors for example in most of the 126 presented equations. Furthermore, I cannot comment if some important work in the area has been overlooked. However, this seems unlike given that the review article well over 600 articles. Therefore, I have only very minor, mostly typographical corrections.
Author response 1:
I’m very delighted to hear that nice opinion on my review. I wish to thank the reviewer for the appreciation of this work and for a great job that the reviewer surely has done to pinpoint such subtle details as, for example, the error in eq. 10. Moreover, I am not a native English speaker, and I am so glad that the reviewer has lent me a helping hand to improve the language. Thank you very much for all!
I have fixed all the issues arisen. The numeration of lines in the revised manuscript has been changed, so I give new line numbers from the left when answering the issues below.
Line:
33: omit “the” to “of medium-size molecules”
33: Done.
122: equation 10 appears to have an incorrect 1 in the numerator
122: Indeed, the eq. 10 is erroneous, that was my misprint. I’ve replaced it with the equation from the book by Becker, E.D. “High Resolution NMR. Theory and Chemical Applications”, 3rd ed., p. 88, eq. (4.6).
177: need to correct “there is no an explicit ensure”
179: I’ve replaced it with “... there is no an explicit reformulation of the problem in the gauge invariant form”
269: the correct noun is introduction not introducing. Please check throughout manuscript
271,176: Done.
279: similar here construction not constructing
281: Done.
444: unclear “increasing it by several times”
446: “it” was replaced with the “the speed of light”. Indeed, increasing the speed of light by the factor of 6 to 10 is already enough to approximate the “nonrelativistic regime.”
517: Within these
552: Oh, what an awful misprint! Accept my apologies. Fixed.
698: triplets? (instead of triples). If so, then correction at other places will be needed
741: triples = triple excitations. I’ve enclosed “triples” in parentheses and clarified the notion.
716: and following: consider using commonly accepted symbols such as 2JFF to clarify which coupling constants are referred to. Also, I am surprised that the longer ranged coupling constants are stated as having larger values. Usually coupling constants become smaller with the number of bonds the coupling nuclei are separated
760-770: Thank you for this remark. Now I see that my writing about JaszuÅ„ski’s work was unclear, moreover, it was so unclear that you have understood me wrong. I totally rewrote this passage using common notations for SSCCs, 1JNN, 1JNF, 2JNF and 3JFF.
747: positively defined electron density? Unclear, seems contradictory
795: “positively defined” was removed.
1144: correct name
1193: Corrected.
1191: something wrong with “ of singles size”
1240-1242: Done. The sentence was reduced, for there is no reason to entangle the reader on the matter that is far beyond the common comprehension.
1224: omit “the”
1274: Done.
1242: basis set
1292: Done.
1404: properties that are …
1454: Done.
1593: Indeed, this is…
1644: Done.
1597: omit one of the “also” in this sentence
1648: Done.
1840: Besides spin-spin coupling…
1892: Done.
1854: over the past
1907: Done.
1970: Four of the five calculations using varying basis sets…
2023: Done.
2031: omit allow to: …that ensure the absence…
2084: Done.
2087: sentence reads strangely
2140: The supermolecular approach can be classified as the discrete-type method for media simulation. So, I’ve paraphrased the sentence as follows: A supermolecule approach belongs to the discrete-type models.

Reviewer 2 Report
It is an excellent review summarizing the various computation models for calculating NMR parameters, such as nuclear spin-spin coupling constants (SSCC) and chemical shifts. When NMR data are applied for structural identification the high-quality quantum chemical calculations are necessary in order to avoid misinterpretation in particular for the case of large biological molecules. This review can offer help to find the most suitable computational methods.
This review gives a very comprehensive report on the route started by the pioneering work of Ramsey to the most recent computations and refers to a great number of works (670!).
There are three major chapters in this review:
- Theoretical background,
- Quantum chemical methods for calculating NMR parameters
- Computational factors influencing the accuracy of NMR spectrum modeling.
The second chapter compares the formalisms of nonrelativistic and relativistic representations of NMR parameters. Here the importance of gauge invariance principle is emphasized. Direct and indirect effects of the relativistic corrections are also taking into account for the diamagnetic spin-orbit contribution (DSO), the paramagnetic spin-orbit contribution (PSO), the Fermi-contact (FC) and spin-dipole (SD) interactions.
In the third chapter first the configuration interaction methods are surveyed. It is compared how the CI can be computed in different approaches namely when singly and doubly excited configurations as well as triples and quadruples are also included. The question arises, however, whether the application of more sophisticated models can also improve the precision of NMR parameters, since due to the lack of size consistency there are considerable errors in the calculation of SSCCs and chemical shifts for large molecules. This point should be critically commented by the author.
In the case of coupled clusters methods the review emphasizes the importance of applying diffuse functions, eg. quadruple zeta basis set, which allows obtaining excellent SSCC values. The most important part of this chapter deals with the results of the density functional theory. Here the electron correlation effects can be taken into account via the exchange-correlation (XC) potential. There is a significant computation advantage that the electron correlation effects are taken into account via the exchange-correlation (XC) potential. The review emphasizes that the critical point is the triplet instability problem for calculating the SSCC values. The top priority of future studies for NMR properties should be evaluating the relativistic effects for heavy element compounds.
The chapter also deals with polarization propagator methods. Here the linear response theory and the second-order polarization propagator approach are compared. The latter proved to be a breakthrough for deriving good NMR parameters.
The last point in this chapter deals with the applications of many-body perturbation theory. It looks like a good compromise between the accuracy and computational costs for calculating the NMR chemical shifts.
Chapter 4 can be considered as the most important part of the whole review, where the computational factors influencing the accuracy of NMR parameters are discussed. It is of primary importance choosing flexible enough specialized basis sets to fully describe orbitals in an important region. The problem is that the magnetic terms represent only a very small part of the total molecular energy and thus the energy-optimized basis sets can often produce false NMR parameters. Different basis sets should be chosen when the SSCC values or the nuclear shielding constants are calculated. Even less precise overall functions can give better results if in the critical domain good and flexible orbitals are chosen. This choice can be different whether the contributions of Fermi-contact, spin-dipole, paramagnetic spin-orbital or diamagnetic spin-orbital terms are calculated.
A special part of this effort is to take into account the vibrational effects, where the anharmonicity of the potential can play an important role. Further correction should be done for the solvation effects. E.g. the nuclear shielding constants are extremely sensitive to the molecular environment, and especially to hydrogen-bonding effects.
As a summary, this review can be very useful for the researchers dealing with the computations of NMR parameters. The mathematical background is extensive and important remarks are given for the molecular backgrounds. This review could be even more useful by adding a new chapter at the end orienting the researchers which mathematical model is the most promising, when a special class of molecules is studied.

Author Response
It is an excellent review summarizing the various computation models for calculating NMR parameters, such as nuclear spin-spin coupling constants (SSCC) and chemical shifts. When NMR data are applied for structural identification the high-quality quantum chemical calculations are necessary in order to avoid misinterpretation in particular for the case of large biological molecules. This review can offer help to find the most suitable computational methods.
This review gives a very comprehensive report on the route started by the pioneering work of Ramsey to the most recent computations and refers to a great number of works (670!).
There are three major chapters in this review:
- Theoretical background,
- Quantum chemical methods for calculating NMR parameters
- Computational factors influencing the accuracy of NMR spectrum modeling.
The second chapter compares the formalisms of nonrelativistic and relativistic representations of NMR parameters. Here the importance of gauge invariance principle is emphasized. Direct and indirect effects of the relativistic corrections are also taking into account for the diamagnetic spin-orbit contribution (DSO), the paramagnetic spin-orbit contribution (PSO), the Fermi-contact (FC) and spin-dipole (SD) interactions.
In the third chapter first the configuration interaction methods are surveyed. It is compared how the CI can be computed in different approaches namely when singly and doubly excited configurations as well as triples and quadruples are also included. The question arises, however, whether the application of more sophisticated models can also improve the precision of NMR parameters, since due to the lack of size consistency there are considerable errors in the calculation of SSCCs and chemical shifts for large molecules. This point should be critically commented by the author.
Author response 1:
First of all, I would like to thank the reviewer for the detailed revision of my manuscript and for kind gratitude to this work.
Indeed, I’ve missed the discussion of the size consistency problem for the truncated CI methods and the CI advanced models. The size consistency problem is, indeed, the one of the key issues hindering a widespread usage of the truncated CI models as compared to the commensurate (in the sense of computational cost) CC models. There were a great number of attempts to introduce the size consistency corrections to CI models, especially, to CISD method. The size consistency feature is not the Forte of the CI models, and I totally agree that this point must have been discussed in more details in the review. To fill this gap, I’ve added the discussion of size consistency problem in lines 502-534, 558-559, 564-568, 572-573. A good many of new references were also added.
Continue:
In the case of coupled clusters methods the review emphasizes the importance of applying diffuse functions, eg. quadruple zeta basis set, which allows obtaining excellent SSCC values. The most important part of this chapter deals with the results of the density functional theory. Here the electron correlation effects can be taken into account via the exchange-correlation (XC) potential. There is a significant computation advantage that the electron correlation effects are taken into account via the exchange-correlation (XC) potential. The review emphasizes that the critical point is the triplet instability problem for calculating the SSCC values. The top priority of future studies for NMR properties should be evaluating the relativistic effects for heavy element compounds.
The chapter also deals with polarization propagator methods. Here the linear response theory and the second-order polarization propagator approach are compared. The latter proved to be a breakthrough for deriving good NMR parameters. The last point in this chapter deals with the applications of many-body perturbation theory. It looks like a good compromise between the accuracy and computational costs for calculating the NMR chemical shifts.
Chapter 4 can be considered as the most important part of the whole review, where the computational factors influencing the accuracy of NMR parameters are discussed. It is of primary importance choosing flexible enough specialized basis sets to fully describe orbitals in an important region. The problem is that the magnetic terms represent only a very small part of the total molecular energy and thus the energy-optimized basis sets can often produce false NMR parameters. Different basis sets should be chosen when the SSCC values or the nuclear shielding constants are calculated. Even less precise overall functions can give better results if in the critical domain good and flexible orbitals are chosen. This choice can be different whether the contributions of Fermi-contact, spin-dipole, paramagnetic spin-orbital or diamagnetic spin-orbital terms are calculated.
A special part of this effort is to take into account the vibrational effects, where the anharmonicity of the potential can play an important role. Further correction should be done for the solvation effects. E.g. the nuclear shielding constants are extremely sensitive to the molecular environment, and especially to hydrogen-bonding effects.
As a summary, this review can be very useful for the researchers dealing with the computations of NMR parameters. The mathematical background is extensive and important remarks are given for the molecular backgrounds. This review could be even more useful by adding a new chapter at the end orienting the researchers which mathematical model is the most promising, when a special class of molecules is studied.
Author response 2:
I must admit that the idea about surveying “mathematical models” for different classes of compounds looks great! Unfortunately, as to my opinion, gathering and analyzing that volume of information surely would require at least half a year. And, this topic is worth a particular review. In the current time frames, I think I can’t manage with this issue this time, but I definitely will take your idea into consideration when the time comes to write a new review. Thank you for the great idea!
